# Unleashing the Denoising Capability of Diffusion Prior for Solving Inverse Problems

**Jiawei Zhang**
Tsinghua University
jiawei-z23@mails.tsinghua.edu.cn

**Jiaxin Zhuang**
Tsinghua University
zhuangjx23@mails.tsinghua.edu.cn

**Cheng Jin**
Tsinghua University
jinc21@mails.tsinghua.edu.cn

**Gen Li**
CUHK
genli@cuhk.edu.hk

**Yuantao Gu**
Tsinghua University
gyt@tsinghua.edu.cn

## Abstract

The recent emergence of diffusion models has significantly advanced the precision of learnable priors, presenting innovative avenues for addressing inverse problems. Previous works have endeavored to integrate diffusion priors into the maximum a posteriori estimation (MAP) framework and design optimization methods to solve the inverse problem. However, prevailing optimization-based rithms primarily exploit the prior information within the diffusion models while neglecting their denoising capability. To bridge this gap, this work leverages the diffusion process to reframe noisy inverse problems as a two-variable constrained optimization task by introducing an auxiliary optimization variable that represents a 'noisy' sample at an equivalent denoising step. The projection gradient descent method is efficiently utilized to solve the corresponding optimization problem by truncating the gradient through the $\mu$-predictor. The proposed algorithm, termed ProjDiff, effectively harnesses the prior information and the denoising capability of a pre-trained diffusion model within the optimization framework. Extensive experiments on the image restoration tasks and source separation and partial generation tasks demonstrate that ProjDiff exhibits superior performance across various linear and nonlinear inverse problems, highlighting its potential for practical applications. Code is available at https://github.com/weigerzan/ProjDiff/.

## 1 Introduction

Denoising diffusion models have achieved tremendous success in the field of generative modeling [1–5]. Their remarkable ability to capture data priors provides promising avenues for solving inverse problems [6], which are widely exploited in image restoration [7–12], medical image processing [13, 14], 3D vision [15], audio processing [16, 17] and beyond.

Numerous endeavors have sought to harness diffusion models to address inverse problems [18–27]. Since the sampling process of diffusion models is a reverse Markov chain, most approaches attempt to integrate the guidance provided by the observation equation into the sampling chain. For instance, DDRM [20] achieves favorable results for linear inverse problems with low complexity by introducing a new variational distribution. DDNM [21] capitalizes on the concept of null-range decomposition, effectively rectifying the range-space component of the intermediate steps by

Jiawei Zhang, Jiaxin Zhuang, Cheng Jin, and Yuantao Gu are with the Department of Electronic Engineering, Beijing National Research Center for Information Science and Technology, Tsinghua University, Beijing 100084, China. Gen Li is with the Department of Statistics, The Chinese University of Hong Kong, Hong Kong, China. The corresponding author is Yuantao Gu.

38th Conference on Neural Information Processing Systems (NeurIPS 2024).

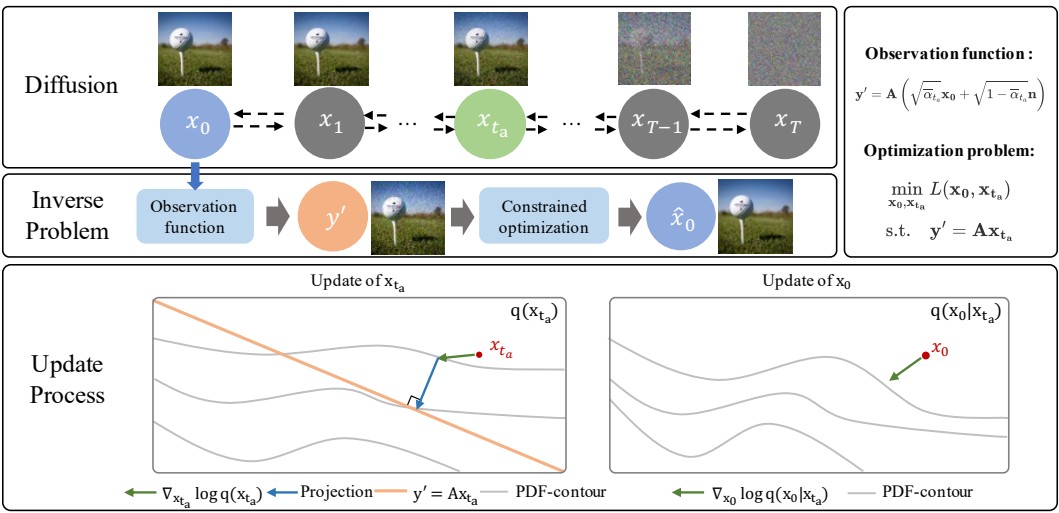

Figure 1: Framework of ProjDiff. We introduce an auxiliary variable $\mathbf{x}_{t_a}$ and transform the inverse problem into a two-variable constrained optimization problem which can be solved using the projection gradient method.

leveraging the observation information. DPS [23] guides the generation process using the gradients of intermediate steps with respect to the observation error. Moreover, methods based on Monte Carlo Particle Filtering [25–27] are employed to approximate the true posterior.

Since inverse problems are often modeled as maximum a posteriori (MAP) estimations, recent efforts have been made to explicitly integrate diffusion models as the prior term in the optimization framework [28, 24]. [28] proposed to employ the Evidence Lower Bound (ELBO) to approximate the prior, facilitating the application of diffusion models in inverse problems in a plug-and-play manner. [24] achieved analogous results from a variational perspective. In these optimization-based methods, due to the presence of the observation noise, a common practice is to adopt a Gaussian likelihood. However, it's worth noting that, since diffusion models are inherently effective at denoising, considering the observation noise in the likelihood term fails to fully leverage diffusion models' denoising capability.

To fully utilize both the prior information and the denoising capability inherent in diffusion models within the optimization framework, we introduce an auxiliary optimization variable to accommodate the influence of the observation noise. We derive a novel two-variable objective based on the properties of the diffusion process and transform the inverse problem into a constrained optimization task. Through gradient truncation, we obtain a more practical approximation of the stochastic gradient of the objective which sidesteps significant computational overhead. The proposed algorithm, termed ProjDiff, tackles the inverse problem by employing the concept of projection gradient descent to solve the corresponding optimization task. We also discuss the noise-free version of ProjDiff as a special case, which, compared to other optimization-based methods, ensures superior consistency between the generated results and observations. ProjDiff's applicability also extends beyond linear observations to encompass nonlinear functions, which enhances its competitiveness and versatility.

We demonstrate the outstanding performance of ProjDiff through comprehensive experiments on various benchmarks. In both linear and nonlinear image restoration tasks, ProjDiff exhibites superior performance among existing state-of-the-art (SOTA) algorithms. In the music separation task, ProjDiff shows for the first time, to the best of our knowledge, that a diffusion-based separation algorithm can surpass previous SOTA, which further demonstrates the powerful potential of diffusion models and provides insights to better harness their capabilities in inverse problems for future research.

## 2 Backgrounds

### 2.1 Denoising diffusion models

Denoising diffusion models [1–3] are a class of latent variable generative models tailored to capture a targeted data distribution $q(\mathbf{x})$. Diffusion models typically predefine a forward process characterized as a Markov chain, wherein the transition probability is stipulated as Gaussian distribution:

$$q(\mathbf{x}_{0:T}) = q(\mathbf{x}_0) \prod_{t=1}^{T} q(\mathbf{x}_t | \mathbf{x}_{t-1}), \quad q(\mathbf{x}_t | \mathbf{x}_{t-1}) = \mathcal{N}(\mathbf{x}_t; a_t \mathbf{x}_{t-1}, b_t \mathbf{I}), \tag{1}$$

where $a_t$ and $b_t$ represent the scale and variance parameters, respectively. The sampling process of diffusion models aims to invert the forward chain via a parameterized reverse process, which is also modeled as a Markov chain with learnable parameter $\boldsymbol{\theta}$:

$$p_{\boldsymbol{\theta}}(\mathbf{x}_{0:T}) = p_{\boldsymbol{\theta}}(\mathbf{x}_T) \prod_{t=1}^{T} p_{\boldsymbol{\theta}}(\mathbf{x}_{t-1} | \mathbf{x}_t), \quad p_{\boldsymbol{\theta}}(\mathbf{x}_{t-1} | \mathbf{x}_t) = \mathcal{N}(\mathbf{x}_{t-1}; \mathbf{f}_{\boldsymbol{\theta}}(\mathbf{x}_t, t), \sigma_t^2 \mathbf{I}), \tag{2}$$

where $\mathbf{f}_{\boldsymbol{\theta}}$ denotes the mean function and the variance $\sigma_t^2$ is predefined. In this work, we primarily focus on the Variance Preservation (VP) diffusion [19][1], characterized by parameters $a_t = \sqrt{\alpha_t}, b_t = 1 - \alpha_t$ with $\alpha_0 \approx 1$ and $\prod_{t=0}^{T} \alpha_t \approx 0$. With these parameters, we have $q(\mathbf{x}_t | \mathbf{x}_0) = \mathcal{N}(\mathbf{x}_t; \sqrt{\overline{\alpha}_t} \mathbf{x}_0, (1 - \overline{\alpha}_t) \mathbf{I})$, where $\overline{\alpha}_t = \prod_{i=0}^{t} \alpha_t$. The initial distribution of the reverse process is set to match the forward process, i.e., $p_{\boldsymbol{\theta}}(\mathbf{x}_T) = \mathcal{N}(\mathbf{0}, \mathbf{I})$, while the mean function is chosen as

$$\mathbf{f}_{\boldsymbol{\theta}}(\mathbf{x}_t, t) = \sqrt{\overline{\alpha}_{t-1}} \boldsymbol{\mu}_{\boldsymbol{\theta}}(\mathbf{x}_t, t) + \sqrt{1 - \overline{\alpha}_{t-1} - \sigma_t^2} \frac{\mathbf{x}_t - \sqrt{\overline{\alpha}_t} \boldsymbol{\mu}_{\boldsymbol{\theta}}(\mathbf{x}_t, t)}{\sqrt{1 - \overline{\alpha}_t}}. \tag{3}$$

Here $\boldsymbol{\mu}_{\boldsymbol{\theta}}$ serves as an estimation of $\mathbf{x}_0$ at time $t$, typically implemented through a parameterized neural network referred to as a "$\boldsymbol{\mu}$-predictor"[2]. The variance $\sigma_t^2 \in [0, 1 - \overline{\alpha}_{t-1}]$ can take various values. When $\sigma_t = \sqrt{\frac{1 - \overline{\alpha}_{t-1}}{1 - \overline{\alpha}_t}} \sqrt{1 - \frac{\overline{\alpha}_t}{\overline{\alpha}_{t-1}}}$, the reverse process aligns with DDPM [2]; while when retaining $\sigma_t = \tilde{\sigma}_t$ as an adjustable parameter, one obtains DDIM [3]. We refer to [19, 29–31] for more theoretical guarantees of diffusion models.

## 2.2 Diffusion models as data prior

As stated in [32, 28], for a well-trained diffusion model, the Evidence Lower Bound (ELBO) can effectively approximate the log-likelihood of the samples, i.e.

$$\log q(\mathbf{x}_0) \approx C - \sum_{t=1}^{T} \lambda(t) \mathbb{E}_{q(\mathbf{x}_t | \mathbf{x}_0)} ||\mathbf{x}_0 - \boldsymbol{\mu}_{\boldsymbol{\theta}}(\mathbf{x}_t, t)||^2, \tag{4}$$

where $\lambda(t)$ represents the coefficient within the ELBO and $C$ is a constant. Utilizing the ELBO as the log-prior term enables diffusion models to serve as a plug-and-play prior for general inverse problems. [28] handled the ELBO via stochastic gradient descent and the reparameterization method. By sampling $t \sim \mathcal{U}\{1, 2, \dots, T\}$ and $\boldsymbol{\epsilon} \sim \mathcal{N}(\mathbf{0}, \mathbf{I})$, a stochastic gradient of the ELBO can be derived as

$$\mathbf{d}_{t, \mathbf{x}_0} = \left( \mathbf{D}_{\mathbf{x}_0} \boldsymbol{\mu}_{\boldsymbol{\theta}} \left( \sqrt{\overline{\alpha}_t} \mathbf{x}_0 + \sqrt{1 - \overline{\alpha}_t} \boldsymbol{\epsilon}, t \right) - \mathbf{I} \right) \left( \mathbf{x}_0 - \boldsymbol{\mu}_{\boldsymbol{\theta}}(\sqrt{\overline{\alpha}_t} \mathbf{x}_0 + \sqrt{1 - \overline{\alpha}_t} \boldsymbol{\epsilon}, t) \right), \tag{5}$$

where $\mathbf{D}_{\mathbf{x}_0} \boldsymbol{\mu}_{\boldsymbol{\theta}}(\cdot)$ denotes the Jocabian of the $\boldsymbol{\mu}$-predictor. [24] derived a similar proxy objective from the perspective of variational inference, and proposed a more computationally efficient approximation of the stochastic gradient by reweighting the objective. Notably, when the variational distribution is a delta function, variational inference aligns with the MAP estimation. In this work, we introduce a novel two-variable ELBO by constructing an auxiliary variable that accounts for the observation noise, thereby utilizing both the prior information and the denoising capability in diffusion models simultaneously.

## 2.3 Equivalent noisy samples

As stated in [26], the noisy observation of a sample can be regarded as a noise-free observation of an equivalent noisy sample at certain steps in the diffusion process. For linear observations $\mathbf{y} = \mathbf{A}\mathbf{x}_0 + \sigma \mathbf{n}$, where $\mathbf{x}_0 \in \mathbb{R}^{m_x}, \mathbf{y} \in \mathbb{R}^{m_y}$, and $\mathbf{n} \in \mathcal{N}(\mathbf{0}, \mathbf{I}_{m_y})$, a common practice [20, 21, 26] involves the Singular Value Decomposition (SVD) to attain a decoupled form. Without loss of generality, assume $\mathbf{A}$ has full row rank. Utilizing the SVD $\mathbf{A} = \mathbf{U}\mathbf{S}\mathbf{V}^T$ yields $\mathbf{U}^T \mathbf{y} = \mathbf{S}(\mathbf{V}^T \mathbf{x}_0) + \sigma \mathbf{n}$, thus

---

[1] This work is also effective for Variance Exploding (VE) diffusion. We defer the discussion to Appendix B.
[2] The case of $\boldsymbol{\epsilon}$-predictor is similar, with the conversion given by $\boldsymbol{\mu}_{\boldsymbol{\theta}}(\mathbf{x}_t) = \left( \mathbf{x}_t - \sqrt{1 - \overline{\alpha}_t} \boldsymbol{\epsilon}_{\boldsymbol{\theta}}(\mathbf{x}_t) \right) / \sqrt{\overline{\alpha}_t}$.

transforming the observation equation to $\overline{\mathbf{y}} = \mathbf{S}\overline{\mathbf{x}}_0 + \sigma\mathbf{n}$, where $\overline{\mathbf{y}} = \mathbf{U}^T\mathbf{y}$ and $\overline{\mathbf{x}}_0 = \mathbf{V}^T\mathbf{x}_0$, which can be further written element-wise as

$$\frac{\overline{\mathbf{y}}_i}{s_i} = \overline{\mathbf{x}}_{0,i} + \frac{\sigma}{s_i}\mathbf{n}_i, 1 \leq i \leq m_{\mathbf{y}}, \tag{6}$$

where $\overline{\mathbf{x}}_{0,i}$ and $\overline{\mathbf{y}}_i$ represent the $i$th component of $\overline{\mathbf{x}}_0$ and $\overline{\mathbf{y}}$, respectively, and $s_i$ denotes the $i$th singular value of $\mathbf{A}$. Assuming there exist $t_i \in \{1, 2, \ldots, T\}, 1 \leq i \leq m_{\mathbf{y}}$ such that $\overline{\alpha}_{t_i} = 1/(1 + (\sigma/s_i)^2)$, let $\overline{\mathbf{y}}'_i = \sqrt{\overline{\alpha}_{t_i}}\overline{\mathbf{y}}_i/s_i$ and then the observation function can be rewritten as

$$\overline{\mathbf{y}}'_i = \sqrt{\overline{\alpha}_{t_i}}\overline{\mathbf{x}}_{0,i} + \sqrt{1 - \overline{\alpha}_{t_i}}\mathbf{n}_i. \tag{7}$$

$\sqrt{\overline{\alpha}_{t_i}}\overline{\mathbf{x}}_{0,i} + \sqrt{1 - \overline{\alpha}_{t_i}}\mathbf{n}_i$ can be interpreted as the $i$th component of a sample at time step $t_i$, namely $\overline{\mathbf{x}}_{t_i,i}$. Thus [26] proposed to employ Monte Carlo Particle Filtering to restore the samples up to the equivalent noise level, and then apply the backward transition to map these noisy samples back to the clean samples. In this work, we introduce this equivalent noisy sample as an auxiliary variable, thus better handling the observation noise in the optimization framework.

# 3   Method

In this section, we introduce the ProjDiff algorithm. The main idea behind ProjDiff lies in constructing an auxiliary variable to align the noisy observations to specific steps of the diffusion process, thereby forming a two-variable constrained optimization task, which can be tackled by approximating the stochastic gradients and employing the projection gradient descent method. We begin by deriving the ProjDiff algorithm for noisy linear observations and then discuss its noise-free version as a special case and the extension of ProjDiff to accommodate nonlinear observations.

## 3.1   ProjDiff

Consider the Gaussian observation equation $\mathbf{y} = \mathbf{A}\mathbf{x}_0 + \sigma\mathbf{n}$. The goal of the inverse problem is to recover the original data $\mathbf{x}_0$ given $\mathbf{y}$, $\mathbf{A}$, and $\sigma$. The decoupled observation equation in (7) indicates that one can apply SVD to reduce a linear observation into the simple form with $\mathbf{A} = [\mathbf{I}_{m_{\mathbf{y}}}, \mathbf{0}]$ in the spectral domain, which implies the observation $\mathbf{y}$ represents the first $m_{\mathbf{y}}$ components of $\mathbf{x}$. Consequently, we can focus on this simplest observation function to declare our algorithm. Under this condition, assuming there exists some $t_a \in [0, 1, \ldots, T]$[3] such that $\overline{\alpha}_{t_a} = 1/(1 + \sigma^2)$, we rewrite the observation function as

$$\mathbf{y}' = [\mathbf{I}_{m_{\mathbf{y}}}, \mathbf{0}] \left(\sqrt{\overline{\alpha}_{t_a}}\mathbf{x}_0 + \sqrt{1 - \overline{\alpha}_{t_a}}\mathbf{n}'\right), \tag{8}$$

where $\mathbf{y}' = \mathbf{y}/\sqrt{1 + \sigma^2}$ and $\mathbf{n}' \sim \mathcal{N}(\mathbf{0}, \mathbf{I}_{m_{\mathbf{x}}})$. The noisy observation $\mathbf{y}'$ can be interpreted as a noise-free observation of $\sqrt{\overline{\alpha}_{t_a}}\mathbf{x}_0 + \sqrt{1 - \overline{\alpha}_{t_a}}\mathbf{n}'$, which in turn can be viewed as a sample in the manifold of $q(\mathbf{x}_{t_a})$. Thus, we introduce an auxiliary variable $\mathbf{x}_{t_a}$ which denotes a noisy sample at time step $t_a$, and propose to optimize the log-posterior term of the two variables as

$$\max_{\mathbf{x}_0, \mathbf{x}_{t_a}} \log q(\mathbf{x}_0, \mathbf{x}_{t_a}|\mathbf{y}) = \max_{\mathbf{x}_0, \mathbf{x}_{t_a}} \log q(\mathbf{y}|\mathbf{x}_0, \mathbf{x}_{t_a}) + \log q(\mathbf{x}_0, \mathbf{x}_{t_a}) - \log q(\mathbf{y}) \tag{9}$$

$$\approx \max_{\mathbf{x}_0, \mathbf{x}_{t_a}} \log q(\mathbf{y}|\mathbf{x}_0, \mathbf{x}_{t_a}) + \log p_{\boldsymbol{\theta}}(\mathbf{x}_0, \mathbf{x}_{t_a}) - \log q(\mathbf{y}), \tag{10}$$

where in (10) we use $p_{\boldsymbol{\theta}}$ to approximate the true prior $q$. By the construction of the auxiliary variable $\mathbf{x}_{t_a}$, the likelihood term embodies a stringent consistency between this auxiliary variable and the noisy observation, which serves as a constraint. Note that $\log q(\mathbf{y})$ is independent of the optimization variables, thus (10) is transformed into the following constrained optimization task:

$$\max_{\mathbf{x}_0, \mathbf{x}_{t_a}} \log p_{\boldsymbol{\theta}}(\mathbf{x}_0, \mathbf{x}_{t_a}) \quad \text{s.t.} \quad \mathbf{y}' = [\mathbf{I}_{m_{\mathbf{y}}}, 0]\mathbf{x}_{t_a}. \tag{11}$$

Now we seek the ELBO as a proxy of the joint log-prior term of $\mathbf{x}_0$ and $\mathbf{x}_{t_a}$ to render it tractable. Leveraging Jensen's inequality and the transition probability of the diffusion model, we reach the following proposition.

---

[3]In practice, $t_a$ may not always be an integer, while the derivation remains similar. We concentrate on the scenario where $t_a$ assumes an integer in the main text. More discussions are deferred to the Appendix.

**Proposition 1.** *Considering the DDIM reference distribution $q_\sigma$, we have the variational lower bound of the log-prior term as*

$$\log p_{\boldsymbol{\theta}}(\mathbf{x}_0, \mathbf{x}_{t_a}) \geq C - \underbrace{\sum_{t=1}^{t_a} \mathbb{E}_{q_\sigma(\mathbf{x}_t|\mathbf{x}_0,\mathbf{x}_{t_a})} \left(g(t)||\mathbf{x}_0 - \boldsymbol{\mu}_{\boldsymbol{\theta}}(\mathbf{x}_t,t)||^2\right)}_{\text{(1) denoising matching term}}$$

$$-\underbrace{\sum_{t=t_a+1}^{T} \mathbb{E}_{q_\sigma(\mathbf{x}_t|\mathbf{x}_{t_a}),\boldsymbol{\epsilon}'\sim\mathcal{N}(\mathbf{0},\mathbf{I})} \left(g(t)\left|\left|\mathbf{x}_{t_a} - \left(\sqrt{\overline{\alpha}_{t_a}}\boldsymbol{\mu}_{\boldsymbol{\theta}}(\mathbf{x}_t,t) + \sqrt{1-\overline{\alpha}_{t_a}}\boldsymbol{\epsilon}'\right)\right|\right|^2 + w(t)\langle\boldsymbol{\mu}_{\boldsymbol{\theta}}(\mathbf{x}_t,t),\boldsymbol{\nu}\rangle\right)}_{\text{(2) noisy prior term}},$$

$$(12)$$

*where $C$ is a constant independent of $\mathbf{x}_0$ and $\mathbf{x}_{t_a}$, the weight $g(t), w(t)$ are functions of $\overline{\alpha}_t$ and the DDIM variance $\tilde{\sigma}_t$, and $\boldsymbol{\nu} = \left(\mathbf{x}_t - \sqrt{\frac{\overline{\alpha}_t}{\overline{\alpha}_{t_a}}}\mathbf{x}_{t_a}\right)/\sqrt{1-\frac{\overline{\alpha}_t}{\overline{\alpha}_{t_a}}}$.*

The complete derivation of the lower bound is provided in Appendix A. Note that the summation on the right-hand side of (12) is partitioned into two parts. The first part, the **denoising matching term**, pertains to the approximation of $\log p_{\boldsymbol{\theta}}(\mathbf{x}_0|\mathbf{x}_{t_a})$, signifying the consistency between the noise-free variable $\mathbf{x}_0$ and the auxiliary variable $\mathbf{x}_{t_a}$. The second part, the **noisy prior term**, corresponds to the approximation of the log-prior of the noisy auxiliary variable $\mathbf{x}_{t_a}$. Conceptually, the workflow of utilizing this ELBO for inverse problems can be delineated as follows: the noisy prior term, coupled with the constraint on the auxiliary variable, exploits the prior information of the diffusion model and the observation information to recover a noisy sample $\mathbf{x}_{t_a}$ that satisfies the observation equation, while the denoising matching term leverages the denoising capability of the diffusion model to restore the clean sample $\mathbf{x}_0$ from the noisy sample $\mathbf{x}_{t_a}$.

Now, we address the ELBO outlined in Proposition 1 utilizing the principle of the stochastic gradient method. The constant $C$ is independent of the optimization variables and can therefore be disregarded. The following proposition is obtained through reparameterization.

**Proposition 2.** *Randomly sampling $t \sim \mathcal{U}\{1, 2, \ldots, T\}, \boldsymbol{\epsilon}, \boldsymbol{\epsilon}' \sim \mathcal{N}(\mathbf{0}, \mathbf{I})$, the stochastic gradients of the proxy objective (for minimization) with respect to $\mathbf{x}_0$ and $\mathbf{x}_{t_a}$ are, respectively,*

$$\mathbf{d}_{\mathbf{x}_0,t} = \begin{cases} \nabla_{\mathbf{x}_0} ||\mathbf{x}_0 - \boldsymbol{\mu}_{\boldsymbol{\theta}}(\mathbf{h}_t,t)||^2 & t \leq t_a; \\ \mathbf{0} & t > t_a, \end{cases} \tag{13}$$

$$\mathbf{d}_{\mathbf{x}_{t_a},t} = \begin{cases} \nabla_{\mathbf{x}_{t_a}} ||\mathbf{x}_0 - \boldsymbol{\mu}_{\boldsymbol{\theta}}(\mathbf{h}_t,t)||^2 & t \leq t_a; \\ \nabla_{\mathbf{x}_{t_a}} ||\mathbf{x}_{t_a} - (\sqrt{\overline{\alpha}_{t_a}}\boldsymbol{\mu}_{\boldsymbol{\theta}}(\mathbf{h}_t,t) + \sqrt{1-\overline{\alpha}_{t_a}}\boldsymbol{\epsilon}')||^2 + w(t)/g(t)\nabla_{\mathbf{x}_{t_a}}\langle\boldsymbol{\mu}_{\boldsymbol{\theta}}(\mathbf{h}_t,t),\boldsymbol{\epsilon}\rangle & t > t_a, \end{cases} \tag{14}$$

*where*

$$\mathbf{h}_t = \begin{cases} \sqrt{\overline{\alpha}_t}\mathbf{x}_0 + \gamma_t(\mathbf{x}_{t_a} - \sqrt{\overline{\alpha}_{t_a}}\mathbf{x}_0)/\sqrt{1-\overline{\alpha}_{t_a}} + \zeta_t\boldsymbol{\epsilon} & t \leq t_a; \\ \sqrt{\overline{\alpha}_t/\overline{\alpha}_{t_a}}\mathbf{x}_{t_a} + \sqrt{1-\overline{\alpha}_t/\overline{\alpha}_{t_a}}\boldsymbol{\epsilon} & t > t_a, \end{cases} \tag{15}$$

*and $\gamma_t, \zeta_t$ are determined by $\overline{\alpha}_t$ and the DDIM variance $\tilde{\sigma}_t$.*

The coefficients are omitted as they can be scaled into the step sizes. Note that in such stochastic gradients, the Jacobian of $\boldsymbol{\mu}_{\boldsymbol{\theta}}(\cdot)$ is involved, which necessitates backpropagation through the neural network and incurs significant computational and storage overhead. Therefore, we use an approximate yet effective and practical method by truncating the gradients in $\boldsymbol{\mu}_{\boldsymbol{\theta}}(\cdot)$, i.e. $\mathbf{D}_{\mathbf{x}_0}\boldsymbol{\mu}_{\boldsymbol{\theta}}(\cdot) \approx \mathbf{0}$ and $\mathbf{D}_{\mathbf{x}_{t_a}}\boldsymbol{\mu}_{\boldsymbol{\theta}}(\cdot) \approx \mathbf{0}$. Intuitively, the $\boldsymbol{\mu}$-predictor of the diffusion model should be resilient to small perturbations in the input. For instance, when we feed a noisy image into a pre-trained diffusion model and introduce minor perturbations to the original image, we anticipate the predicted image to maintain consistency with the original one. Thus the approximation for $\mathbf{x}_0$ is acceptable, and similarly for $\mathbf{x}_{t_a}$ if the noise level of $\mathbf{x}_{t_a}$ is not too large. By such gradient truncation, the approximation of the stochastic gradients for $\mathbf{x}_0$ and $\mathbf{x}_{t_a}$ are, respectively,

$$\tilde{\mathbf{d}}_{\mathbf{x}_0,t} = \begin{cases} \mathbf{x}_0 - \boldsymbol{\mu}_{\boldsymbol{\theta}}(\mathbf{h}_t,t) & t \leq t_a; \\ \mathbf{0} & t > t_a, \end{cases} \tag{16}$$

$$\tilde{\mathbf{d}}_{\mathbf{x}_{t_a},t} = \begin{cases} \mathbf{0} & t \leq t_a; \\ \mathbf{x}_{t_a} - \sqrt{\overline{\alpha}_{t_a}}\boldsymbol{\mu}_{\boldsymbol{\theta}}(\mathbf{h}_t,t) - \sqrt{1-\overline{\alpha}_{t_a}}\boldsymbol{\epsilon}' & t > t_a. \end{cases} \tag{17}$$

Note that $\mathbf{x}_0$ is not subject to the constraint and can be updated directly by stochastic gradient descent. $\mathbf{x}_{t_a}$ is involved in the equality constraint, so we apply the projection gradient descent method. Considering the projection operator of the observation in (11), an iteration step of the proposed algorithm is outlined as follows

$$\mathbf{x}_0 \leftarrow \mathbf{x}_0 - \eta_1 \tilde{\mathbf{d}}_{\mathbf{x}_0,t}, \tag{18}$$

$$\mathbf{x}_{t_a} \leftarrow \left[ \mathbf{I}_{m_{\mathbf{y}}}, \mathbf{0}_{m_{\mathbf{y}} \times (m_{\mathbf{x}} - m_{\mathbf{y}})} \right]^T \mathbf{y}' + \mathrm{diag}\left( \mathbf{0}_{m_{\mathbf{y}} \times m_{\mathbf{y}}}, \mathbf{I}_{m_{\mathbf{x}} - m_{\mathbf{y}}} \right) \left( \mathbf{x}_{t_a} - \eta_2 \tilde{\mathbf{d}}_{\mathbf{x}_{t_a},t} \right), \tag{19}$$

where $\eta_1$ and $\eta_2$ are selected step sizes and $\mathrm{diag}(\cdot)$ denotes the operation of arranging entries into a diagonal matrix. We term this algorithm ProjDiff.

---

**Algorithm 1** ProjDiff for VP diffusion (noisy observation).

---

**Require:** Observation $\mathbf{y}'$, pre-trained diffusion model $\boldsymbol{\mu}_{\boldsymbol{\theta}}$, step sizes $\eta_1, \eta_2$, total steps $T$, noise schedule $\overline{\alpha}_1 \dots \overline{\alpha}_T$, equivalent noise level $\overline{\alpha}_{t_a}$.
1: Sample $\boldsymbol{\epsilon}_T, \boldsymbol{\epsilon}'_T \sim \mathcal{N}(\mathbf{0}, \mathbf{I})$;
2: Initialize $\mathbf{x}_{t_a} \leftarrow \sqrt{\overline{\alpha}_{t_a}} \boldsymbol{\mu}_{\boldsymbol{\theta}}(\boldsymbol{\epsilon}_T, T) + \sqrt{1 - \overline{\alpha}_{t_a}} \boldsymbol{\epsilon}'_T$;
3: **for** $t = T$ to $t_a + 1$ **do**
4:     Sample $\boldsymbol{\epsilon}, \boldsymbol{\epsilon}' \sim \mathcal{N}(\mathbf{0}, \mathbf{I})$;
5:     Calculate the approximate stochastic gradient $\tilde{\mathbf{d}}_{\mathbf{x}_{t_a},t}$ as (17);
6:     Update $\mathbf{x}_{t_a}$ as (19);
7: **end for**
8: Initialize $\mathbf{x}_0 \leftarrow \boldsymbol{\mu}_{\boldsymbol{\theta}}(\mathbf{x}_{t_a}, t_a)$;
9: **for** $t = t_a$ to $1$ **do**
10:     Sample $\boldsymbol{\epsilon} \sim \mathcal{N}(\mathbf{0}, \mathbf{I})$;
11:     Calculate the approximate stochastic gradient $\tilde{\mathbf{d}}_{\mathbf{x}_0,t}$ as (16);
12:     Update $\mathbf{x}_0$ as (18);
13: **end for**
14: **return** $\mathbf{x}_0$

---

## 3.2 Noise-free observations: a special case

In the noise-free scenario, the observation equation becomes $\mathbf{y} = \mathbf{A}\mathbf{x}_0$ and the auxiliary variable $\mathbf{x}_{t_a}$ degrades to $t_a = 0$, thus the constraint is directly applied on $\mathbf{x}_0$. The corresponding optimization problem can be expressed as

$$\min_{\mathbf{x}_0} \sum_{t=1}^{T} \lambda(t) \mathbb{E}_{q(\mathbf{x}_t|\mathbf{x}_0)} ||\mathbf{x}_0 - \boldsymbol{\mu}_{\boldsymbol{\theta}}(\mathbf{x}_t, t)||^2 \qquad \text{s.t.} \quad \mathbf{y} = \mathbf{A}\mathbf{x}_0. \tag{20}$$

Utilizing reparameterization and gradient truncation, the approximate stochastic gradient of the objective is

$$\tilde{\mathbf{d}}_{t,\mathbf{x}_0} = \mathbf{x}_0 - \boldsymbol{\mu}_{\boldsymbol{\theta}} \left( \sqrt{\overline{\alpha}_t} \mathbf{x}_0 + \sqrt{1 - \overline{\alpha}_t} \boldsymbol{\epsilon}, t \right). \tag{21}$$

Then the iteration of ProjDiff in the noise-free scenario is given by:

$$\mathbf{x}_0 \leftarrow \mathbf{A}^{\dagger} \mathbf{y} + \left( \mathbf{I} - \mathbf{A}^{\dagger} \mathbf{A} \right) (\mathbf{x}_0 - \eta \tilde{\mathbf{d}}_{t,\mathbf{x}_0}), \tag{22}$$

where $\mathbf{A}^{\dagger}$ is the Moore-Penrose pseudo-inverse of matrix $\mathbf{A}$ and $\eta$ is the selected step size. The principle of the ProjDiff algorithm in noise-free scenarios resides in treating the observation as a constraint. In contrast to [24] where the authors used Gaussian likelihood for noise-free observations, ProjDiff guarantees better consistency between the generated results and the observations.

## 3.3 Extension to nonlinear observation

Nonlinear inverse problems are inherently more ill-posed than linear ones. For instance, in our experiments, we consider the phase retrieval task, which aims to recover the original image given only the Fourier magnitude spectrum. The observation equation is:

$$\mathbf{y} = |\mathrm{DFT}(\mathbf{P}\mathbf{x}_0)|, \tag{23}$$

where $\mathbf{P}$ is the zero-padding matrix. Effectively addressing nonlinear inverse problems can significantly enhance the algorithm's versatility. Here, we discuss about the ProjDiff algorithm for nonlinear observations concerning noise-free and noisy scenarios.

For noise-free nonlinear observation functions, the key point of ProjDiff lies in identifying an appropriate projection operator for the nonlinear constraint $\mathbf{y} = \mathcal{A}(\mathbf{x}_0)$. For some nonlinear functions $\mathcal{A}(\cdot)$, such a projection operator is analytic or can be approximated. For more intricate functions, we can resort to gradient descent or sub-gradient methods to minimize $||\mathbf{y} - \mathcal{A}(\mathbf{x})||^2$ starting from the initial point $\mathbf{x}_0$ to approximate the projection operator. Thus ProjDiff is applicable to such class of nonlinear observations.

For noisy inverse problems, ProjDiff necessitates mapping the noise applied on the observation to the noise applied on the original data $\mathbf{x}_0$. Such a mapping is strict in the linear case. For nonlinear observations, the exact equivalent noise might not be accessible but may be estimated through some appropriate approach. In cases where the observation function is overly complex, one may leave the determination of the equivalent noise variance as a hyperparameter to be manually tuned. This renders ProjDiff applicable to noisy nonlinear observations.

### 3.4 Time steps, initialization, and Restricted Encoding

Here we elaborate on crucial details and address the weak observation problem.

**Time steps.** In practice, we observe that selecting time steps in a decreasing manner from $T$ to $1$ yields the optimal performance and provides more stable results with reduced oscillations, which aligns with the findings reported in [28, 24]. Therefore, we adopt this time step schedule as the default setting in this work. Furthermore, we note that in certain scenarios, repeating the same time step multiple times (say $N$ times) leads to enhanced performance, which bears a resemblance to the correction step proposed in [19] and the time-travel technique in [21].

**Initialization.** We propose to use random initialization in the ProjDiff algorithm. The core concept entails initializing $\mathbf{x}_0$ and $\mathbf{x}_{t_a}$ to be close to the manifold of $q(\mathbf{x}_0)$ and $q(\mathbf{x}_{t_a})$, respectively, which is achievable via the $\boldsymbol{\mu}$-predictor (or the Tweedie's formula [33]) and the forward process. Randomly sampling a Gaussian noise $\boldsymbol{\epsilon}_T \sim \mathcal{N}(\mathbf{0}, \mathbf{I})$, we initialize the optimization variables by $\mathbf{x}_0 \leftarrow \boldsymbol{\mu_\theta}(\boldsymbol{\epsilon}_T, T)$ and $\mathbf{x}_{t_a} \leftarrow \sqrt{\overline{\alpha}_{t_a}}\mathbf{x}_0 + \sqrt{1 - \overline{\alpha}_{t_a}}\boldsymbol{\epsilon}'_T$ for some $\boldsymbol{\epsilon}'_T \sim \mathcal{N}(\mathbf{0}, \mathbf{I})$. Nonetheless, given the time step schedule and the form of the stochastic gradients discussed above, we note that the $\mathbf{x}_0$ remains unaltered until $\mathbf{x}_{t_a}$ is fully optimized. Therefore, for noisy observations, it would be more efficient to re-initialize $\mathbf{x}_0$ as $\boldsymbol{\mu_\theta}(\mathbf{x}_{t_a}, t_a)$ once $\mathbf{x}_{t_a}$ has been optimized.

**Weak observation problem.** In most image restoration tasks, the constraints imposed by observations are typically strong enough to lead to a unique original data point, rendering ProjDiff effective in yielding satisfactory results. However, in scenarios where observations are weak, particularly in situations with significant degrees of freedom in the inverse problem, the performance of ProjDiff tends to decline. We term these scenarios the weak observation problem. Drawing inspiration from [34], where the authors found that including a larger variance when remapping $\mathbf{x}_0$ to $\mathbf{x}_t$ leads to noisy results, we attempt to adjust the noise schedule to diminish the variance of the sampled $\mathbf{x}_t$. We propose Restricted Encoding to tackle the weak observation problem, which entails fixing an initial noise $\boldsymbol{\epsilon}_0 \sim \mathcal{N}(\mathbf{0}, \mathbf{I})$ and reparameterizing the sampling of $\mathbf{x}_t$ as

$$\mathbf{x}_t = \sqrt{\overline{\alpha}_t}\mathbf{x}_0 + \xi\boldsymbol{\epsilon}_0 + \sqrt{1 - \overline{\alpha}_t - \xi^2}\boldsymbol{\epsilon}, \tag{24}$$

where $\boldsymbol{\epsilon} \sim \mathcal{N}(\mathbf{0}, \mathbf{I})$, and $\xi \in [0, \sqrt{1 - \overline{\alpha}_t}]$ regulates the level of randomness.

The complete implementation of ProjDiff for noisy observations is outlined in Algorithm 1. More variations of ProjDiff used in our experiments can be found in Appendix H.

## 4 Experiments.

We present the comparison of the proposed ProjDiff algorithm with several SOTA algorithms in two primary scenarios: (1) image restoration, and (2) source (music) separation and partial generation. All experiments are performed on a single NVIDIA 3090ti GPU.

Table 1: Noisy restoration on ImageNet with $\sigma = 0.05$. The LPIPS metrics are multiplied by 100.

| NFEs | Method | Super-Resolution | Inpainting | Gaussian Deblurring |
|---|---|---|---|---|
| | | PSNR↑ SSIM↑ LPIPS↓ FID↓ | PSNR↑ SSIM↑ LPIPS↓ FID↓ | PSNR↑ SSIM↑ LPIPS↓ FID↓ |
| - | $A^\dagger y$ | 21.85 / 0.45 / 65.34 / 183.32 | 14.21 / 0.24 / 67.42 / 176.52 | 17.78 / 0.29 / 60.25 / 100.05 |
| 1000 | DPS | 24.44 / 0.67 / **31.81** / **36.17** | 30.15 / 0.86 / 17.76 / 22.03 | 24.26 / 0.64 / 37.03 / 50.17 |
| 100 | DDRM | 25.66 / 0.72 / 34.88 / 55.71 | 29.99 / 0.87 / 17.11 / 19.88 | 27.82 / 0.79 / 28.62 / 45.96 |
| 100 | DDNM+ | 25.61 / 0.72 / 34.45 / 54.10 | 30.00 / 0.87 / 17.09 / 20.08 | 27.90 / **0.79** / 28.63 / 46.54 |
| 100 | RED-diff | 22.74 / 0.49 / 53.24 / 96.26 | 9.85 / 0.17 / 83.92 / 281.65 | 23.74 / 0.50 / 48.12 / 68.23 |
| 100 | **ProjDiff** | **25.73** / **0.73** / 34.12 / 55.03 | **31.09** / **0.89** / **13.65** / **13.69** | **27.91** / **0.79** / **25.11** / **32.27** |

Table 2: Phase retrieval results. The LPIPS metrics are multiplied by 100.

| NFEs | Method | Phase Retrieval $\sigma = 0$ | Phase Retrieval $\sigma = 0.1$ |
|---|---|---|---|
| | | PSNR↑ SSIM↑ LPIPS↓ FID↓ | PSNR↑ SSIM↑ LPIPS↓ FID↓ |
| - | ER | 11.15 / 0.19 / 84.48 / 409.91 | 11.16 / 0.19 / 84.43 / 412.59 |
| - | HIO | 11.97 / 0.25 / 82.06 / 342.52 | 11.87 / 0.24 / 82.18 / 339.13 |
| - | OSS | 12.57 / 0.35 / 81.08 / 360.98 | 12.55 / 0.25 / 81.20 / 364.52 |
| 1000 | DPS | 13.19 / 0.20 / 67.14 / 172.56 | 12.58 / 0.18 / 67.56 / 134.64 |
| 1000 | RED-diff | 14.19 / 0.28 / 63.42 / 173.52 | 12.36 / 0.14 / 75.98 / 221.11 |
| 1000 | **ProjDiff** | **33.39 / 0.74 / 20.42 / 35.91** | **23.84 / 0.60 / 38.55 / 76.20** |

## 4.1 Image restoration

**Linear tasks.** We demonstrate the performance of ProjDiff across three linear image restoration tasks on ImageNet [35]: $4\times$ super-resolution (with average pooling), 50% random inpainting, and Gaussian deblurring. Comparisons are conducted against prominent diffusion-based image restoration algorithms in recent years, including DDRM [20], DDNM+ [21], DPS [23], and RED-diff [24]. The performance of the least square solution ($\hat{\mathbf{x}}_0 = \mathbf{A}^\dagger \mathbf{y}$) is reported as a baseline. For a fair comparison, DDRM, DDNM, RED-diff, and ProjDiff utilize 100 function evaluations, while DPS uses 1000 function evaluations as its performance significantly declines with 100 steps. The pre-trained model on ImageNet is sourced from [36], specifically the $256 \times 256$ model without classifier guidance. Testing is conducted on a 1k sub-testset of ImageNet consistent with [21]. Metrics include PSNR, SSIM [37], LPIPS [38], and FID [39], with all LPIPS values multiplied by 100 for clarity.

Table 1 presents the restoration metrics for noisy observations with standard deviation $\sigma = 0.05$ (doubled when pixels are rescaled to $[-1, 1]$) on ImageNet. ProjDiff demonstrated highly competitive performance compared to other algorithms. Some visualization results are shown in Figure 2. More experiments on ImageNet and CelebA [40] alongside ablation studies are shown in Appendix C.

**Nonlinear tasks.** We evaluate the effectiveness of ProjDiff for nonlinear observations through the challenging phase retrieval task on the FFHQ dataset [41]. We compare ProjDiff against DPS [23] and RED-diff [24] as they are applicable to nonlinear observations. Traditional baseline algorithms are also reported, including the Error Reduction (ER) algorithm [42], Hybrid Input-Output (HIO) [43], and Oversampling Smoothness (OSS) [44]. Table 2 presents the results for both noise-free and noisy scenarios. One sample is generated per image from each algorithm. Given the inherently ill-posedness of phase retrieval, ProjDiff requires more steps to achieve superior results. We set the number of function evaluations to 1000 for all three algorithms. The results demonstrate that ProjDiff excels in addressing nonlinear inverse problems, regardless of whether the observation is noise-free or noisy. Some visualization results are shown in Figure 3. Further experiments on the high dynamic range (HDR) task can be found in Appendix C.

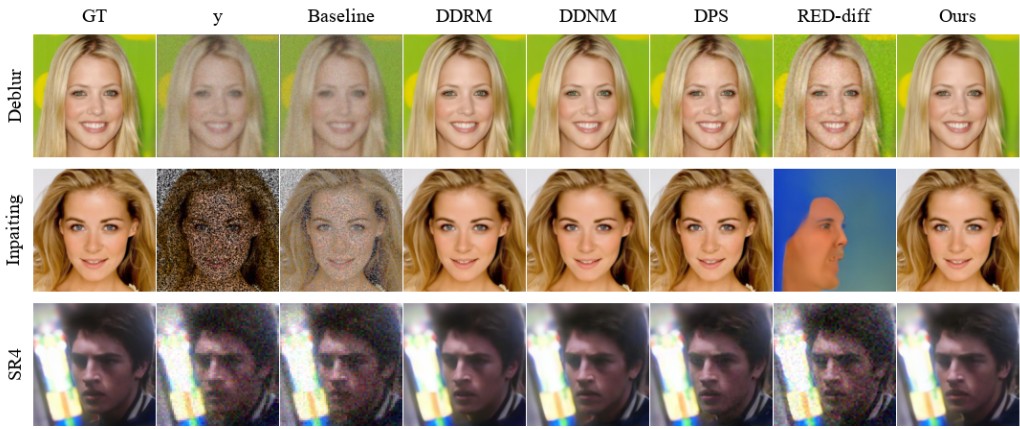

Figure 2: Linear restoration on CelebA ($\sigma = 0.05$). Baseline means $\hat{\mathbf{x}}_0 = \mathbf{A}^\dagger \mathbf{y}$.

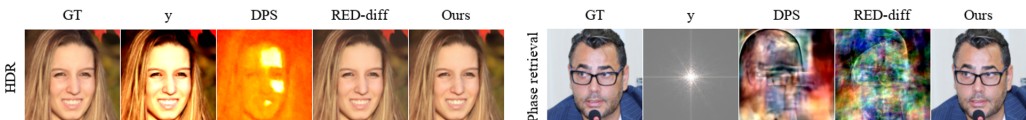

Figure 3: Nonlinear restoration on FFHQ (noise-free).

## 4.2 Source separation and partial generation

We evaluate ProjDiff on source separation and partial generation tasks following [45, 46]. The SLACK2100 dataset [47] is employed and the pre-trained diffusion models are sourced from [45].

The objective of the source separation task is to separate the audio tracks of four instruments (piano, bass, guitar, and drums) from a mixed audio sequence. We compare ProjDiff against the methods reported in [45], along with RED-diff [24] due to its relevance to this work. Performance is assessed using the scale-invariant SDR improvement (SI-SDR$_i$) [48] for each instrument as well as their average. Tabel 3 presents the performance on the ISDM model [45] of different algorithms. ProjDiff demonstrates significant performance improvements compared to other diffusion-based algorithms. The particularly noteworthy result is that ProjDiff surpasses the Demucs method [49], marking the first instance, to the best of our knowledge, where a diffusion-based separation algorithm significantly outperforms previous SOTA, not to mention that the diffusion model is not specifically designed or trained for the separation task.

The partial generation task aims to generate tracks of other instruments given partial of the instruments tracks while ensuring harmony. We employ the sub-FAD metrics [50, 45] for different partial generation tasks. Note that partial generation poses a weak observation problem as discussed in Section 3.4, thus the proposed Restricted Encoding method is applied for ProjDiff. The results are shown in Table 4. ProjDiff outperforms other algorithms across different partial generation tasks. Additional results and in-depth analysis including ablation studies can be found in Appendix D.

## 5 Conclusion

In this work, we introduce ProjDiff, a versatile inverse problem solver that harnesses the power of diffusion models to capture intricate data prior and denoise simultaneously. By deriving a novel two-variable ELBO as a proxy for the log-prior, we reframe the inverse problems as constrained optimization tasks and address them via the projection gradient method. We meticulously explore the implementation details of ProjDiff and propose refined initialization methods and optional noise schedules to enhance its performance. Through extensive experiments across image restoration tasks and source separation and partial generation tasks, we demonstrate the competitive performance and versatility of ProjDiff in linear, nonlinear, noise-free, and noisy problems. ProjDiff provides insights into better leveraging diffusion priors for inverse problems for future work.

Table 3: SI-SDR$_i$ for source separation task (higher is better).

| Method | Bass | Drums | Guitar | Piano | Average |
|--------|------|-------|--------|-------|---------|
| Demucs+Gibbs* | 17.16 | 19.61 | **17.82** | 16.32 | 17.73 |
| ISDM-Gaussian | 14.27 | 19.10 | 12.74 | 12.20 | 14.58 |
| ISDM-Dirac | 19.36 | 20.90 | 14.70 | 14.13 | 17.27 |
| RED-diff | 17.96 | 21.60 | 15.99 | 15.33 | 17.72 |
| **ProjDiff** | **20.09** | **22.91** | 17.29 | **16.62** | **19.23** |

\* Previous SOTA.

Table 4: The sub-FAD metrics on different partial generation tasks (lower is better). The capital letters in the table header represent the initial letters of the generated instruments. For example, BD represents the task of generating bass and drums given piano and guitar.

| Method | B | D | G | P | BD | BG | BP | DG | DP | GP | BDG | BDP | BGP | DGP |
|--------|------|------|------|------|------|------|------|------|------|------|------|------|------|------|
| MSDM | 0.43 | 1.30 | **0.12** | 0.73 | 1.79 | 0.79 | 1.60 | 1.59 | 2.00 | 1.47 | 2.19 | 2.63 | 3.02 | 3.02 |
| RED-diff | 0.44 | 2.26 | 0.18 | 0.71 | 3.66 | 1.14 | 2.56 | 3.10 | 3.26 | 1.74 | 5.65 | 7.73 | 6.14 | 4.87 |
| **ProjDiff** | **0.42** | **1.15** | 0.31 | **0.60** | **1.37** | **0.69** | **1.06** | **1.41** | **1.60** | **1.17** | **1.66** | **1.79** | **1.85** | **2.25** |

**Limitations and future work.** One of the possible limitations of ProjDiff is the potential challenge of effectively handling noisy observations with highly intricate functions. Also, its reliance on Gaussian observation noise may limit its applicability to other noise types like Poisson or multiplicative noise. Additionally, ProjDiff needs manual tuning for the step size. Future research could focus on extending ProjDiff to tackle intricate noisy nonlinear observations more effectively, as well as developing adaptive step size strategies to further enhance the performance while reducing the need for manual intervention.

## Acknowledgement

This work was supported by NSAF (Grant No. U2230201) and a Grant from the Guoqiang Institute, Tsinghua University.

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

# Appendix

In the appendix, we provide further details and additional experimental results omitted in the main text. First, we present the complete derivations of the propositions for the ProjDiff algorithm (Appendix A), and then provide the derivations for ProjDiff in the variance exploding setting (Appendix B). Next, we present additional numerical results. For the image restoration tasks, we report the results of noise-free scenarios on the ImageNet dataset, both noise-free and noisy scenarios on the CelebA dataset, the nonlinear HDR task on the FFHQ dataset, and the ablation studies (Appendix C). For source separation and partial generation tasks, results on the MSDM model, and ablation experiments with waveform visualizations are shown in Appendix D. Then we provide the details of the experiments, including the mathematical form of the inverse problems, and the details for calculating metrics (Appendix E). The implementation details of ProjDiff for various inverse problems are discussed in Appendix F, especially the details for the phase retrieval task and high dynamic range task. Discussion about the performance gap between ProjDiff and the MAP framework is presented in Appendix G. Algorithm blocks for all kinds of ProjDiff used in this work are outlined in Appendix H. Finally, we include more visualizations of the restoration results to present the performance of ProjDiff intuitively in Appendix I.

## A  Derivation for ProjDiff in VP diffusion

### A.1  Preliminaries and lemmas

Assume $\mathbf{y} = \mathbf{A}\mathbf{x}_0 + \mathbf{n} = [\mathbf{I}_{m_{\mathbf{y}}}, \mathbf{0}]\mathbf{x}_0 + \mathbf{n}$. For general linear observations, SVD can be used to reach this decoupled form in the spectral domain. We rewrite the observation equation as

$$\mathbf{y}' = \left[\mathbf{I}_{m_{\mathbf{y}}}, \mathbf{0}\right]\left(\sqrt{\overline{\alpha}_{t_a}}\mathbf{x}_0 + \sqrt{1 - \overline{\alpha}_{t_a}}\mathbf{n}'\right), \tag{25}$$

where $\overline{\alpha}_{t_a} = 1/(1 + \sigma^2)$ for some $t_a \in [0, 1, \ldots, T]$, $\mathbf{y}' = \sqrt{\overline{\alpha}_{t_a}}\mathbf{y}$ and $\mathbf{n}' \sim \mathcal{N}(\mathbf{0}, \mathbf{I}_{m_{\mathbf{x}}})$. This makes the $\mathbf{y}'$ a noise-free observation for a sample at time step $t_a$, namely the auxiliary variable $\mathbf{x}_{t_a}$.

We consider the DDIM reference distribution as

$$q_\sigma(\mathbf{x}_{1:T}|\mathbf{x}_0) = q_\sigma(\mathbf{x}_T|\mathbf{x}_0)\prod_{t=2}^{T} q_\sigma(\mathbf{x}_{t-1}|\mathbf{x}_t, \mathbf{x}_0), \tag{26}$$

where the transition probability follows

$$q_\sigma(\mathbf{x}_T|\mathbf{x}_0) = \mathcal{N}(\mathbf{0}, \mathbf{I}),$$
$$q_\sigma(\mathbf{x}_{t-1}|\mathbf{x}_t, \mathbf{x}_0) = \mathcal{N}\left(\mathbf{x}_{t-1}; \sqrt{\overline{\alpha}_{t-1}}\mathbf{x}_0 + \sqrt{1 - \overline{\alpha}_{t-1} - \tilde{\sigma}_t^2}\frac{\mathbf{x}_t - \sqrt{\overline{\alpha}_t}\mathbf{x}_0}{\sqrt{1 - \overline{\alpha}_t}}, \tilde{\sigma}_t^2\mathbf{I}\right), \tag{27}$$

where $\tilde{\sigma}_t$ is the variance hyperparameter in DDIM. We further choose $\tilde{\sigma}_t = \sigma_t = \sqrt{\frac{1 - \overline{\alpha}_{t-1}}{1 - \overline{\alpha}_t}}\sqrt{1 - \frac{\overline{\alpha}_t}{\overline{\alpha}_{t-1}}}$ for $t > t_a$, which makes the stochastic process between $t_a$ and $T$ Markov. For $t \leq t_a$, we leave $\tilde{\sigma}_t$ a hyperparameter. Thus the reference distribution can be rewritten as

$$q_\sigma(\mathbf{x}_{1:T}|\mathbf{x}_0) = q_\sigma(\mathbf{x}_T|\mathbf{x}_{t_a})q_\sigma(\mathbf{x}_{t_a}|\mathbf{x}_0)\prod_{t=2}^{t_a} q_\sigma(\mathbf{x}_{t-1}|\mathbf{x}_t, \mathbf{x}_0)\prod_{t=t_a+2}^{T} q_\sigma(\mathbf{x}_{t-1}|\mathbf{x}_{t_a}, \mathbf{x}_t). \tag{28}$$

Thus

$$q_\sigma(\mathbf{x}_{1:T\backslash t_a}|\mathbf{x}_{t_a}, \mathbf{x}_0) = q_\sigma(\mathbf{x}_T|\mathbf{x}_{t_a})\prod_{t=2}^{t_a} q_\sigma(\mathbf{x}_{t-1}|\mathbf{x}_t, \mathbf{x}_0)\prod_{t=t_a+2}^{T} q_\sigma(\mathbf{x}_{t-1}|\mathbf{x}_{t_a}, \mathbf{x}_t). \tag{29}$$

The following Lemma regarding transition probabilities can be proved by induction.

**Lemma 1.** *The conditional distribution of $\mathbf{x}_t$ conditioning on $\mathbf{x}_{t_a}$ and $\mathbf{x}_0$ is*

$$q_\sigma(\mathbf{x}_t|\mathbf{x}_{t_a}, \mathbf{x}_0) = \begin{cases} \mathcal{N}\left(\mathbf{x}_t; \sqrt{\overline{\alpha}_t/\overline{\alpha}_{t_a}}\mathbf{x}_{t_a}, (1 - \overline{\alpha}_t/\overline{\alpha}_{t_a})\mathbf{I}\right) & t > t_a; \\ \mathcal{N}\left(\mathbf{x}_t; \sqrt{\overline{\alpha}_t}\mathbf{x}_0 + \gamma_t\left(\mathbf{x}_{t_a} - \sqrt{\overline{\alpha}_{t_a}}\mathbf{x}_0\right)/\sqrt{1 - \overline{\alpha}_{t_a}}, \zeta_t^2\mathbf{I}\right) & t \leq t_a, \end{cases} \tag{30}$$

*where*

$$\gamma_t = \sqrt{1 - \overline{\alpha}_{t_a} - \tilde{\sigma}_{t_a}^2} \prod_{k=t}^{t_a-1} \frac{\sqrt{1 - \overline{\alpha}_k - \tilde{\sigma}_{k+1}^2}}{\sqrt{1 - \overline{\alpha}_{k+1}}}, \zeta_t = \sqrt{\sum_{k=t}^{t_a-1} \tilde{\sigma}_{k+1}^2}. \tag{31}$$

*Proof.* When $t > t_a$, (30) holds as the chain between $t_a$ and $T$ Markov.

When $t < t_a$, (30) and (31) holds when $t = t_a$ as it is equal to the DDIM reference distribution. We assume (30) and (31) holds when $t = t_0 \in \{1, 2, \ldots, t_a\}$, then when $t = t_0 - 1$, we have

$$q_\sigma(\mathbf{x}_{t_0-1}|\mathbf{x}_{t_a}, \mathbf{x}_0) = \int \mathbf{dx}_{t_0} q_\sigma(\mathbf{x}_{t_0-1}, \mathbf{x}_{t_0}|\mathbf{x}_{t_a}, \mathbf{x}_0)$$

$$= \int \mathbf{dx}_{t_0} q_\sigma(\mathbf{x}_{t_0-1}|\mathbf{x}_{t_0}, \mathbf{x}_0) q_\sigma(\mathbf{x}_{t_0}|\mathbf{x}_{t_a}, \mathbf{x}_0). \tag{32}$$

$q_\sigma(\mathbf{x}_{t_0-1}|\mathbf{x}_{t_0}, \mathbf{x}_0)$ is a Gaussian distribution according to the DDIM refernece distribution, and $q_\sigma(\mathbf{x}_{t_0}|\mathbf{x}_{t_a}, \mathbf{x}_0)$ is a Gaussian distribution due to the inductive hypothesis. Then by the convolution law of two Gaussian distributions, (30) and (31) hold when $t = t_0 - 1$. By induction, Lemma 1 concludes. $\square$

The following lemma can be proved using Bayes's formula.

**Lemma 2.** *The conditional distribution of $\mathbf{x}_{t-1}$ conditioned on $\mathbf{x}_{t_a}$ and $\mathbf{x}_t$ for $t \geq t_a + 2$ is*

$$q_\sigma(\mathbf{x}_{t-1}|\mathbf{x}_t, \mathbf{x}_{t_a}) = \mathcal{N}\left(\mathbf{x}_{t-1}; \sqrt{\overline{\alpha}_{t-1}/\overline{\alpha}_{t_a}}\mathbf{x}_{t_a} + \sqrt{1 - \overline{\alpha}_{t-1}/\overline{\alpha}_{t_a} - \hat{\sigma}_t^2}\frac{\mathbf{x}_t - \sqrt{\overline{\alpha}_t/\overline{\alpha}_{t_a}}\mathbf{x}_{t_a}}{\sqrt{1 - \overline{\alpha}_t/\overline{\alpha}_{t_a}}}, \hat{\sigma}_t^2\mathbf{I}\right), \tag{33}$$

*where*

$$\hat{\sigma}_t^2 = (1 - \overline{\alpha}_t/\overline{\alpha}_{t-1})(1 - \overline{\alpha}_{t-1}/\overline{\alpha}_{t_a})/(1 - \overline{\alpha}_t/\overline{\alpha}_{t_a}). \tag{34}$$

*Proof.* By Bayes's formula,

$$q_\sigma(\mathbf{x}_{t-1}|\mathbf{x}_t, \mathbf{x}_{t_a}) = \frac{q_\sigma(\mathbf{x}_t|\mathbf{x}_{t-1}, \mathbf{x}_{t_a})q_\sigma(\mathbf{x}_{t-1}|\mathbf{x}_{t_a})}{q_\sigma(\mathbf{x}_t|\mathbf{x}_{t_a})} = \frac{q_\sigma(\mathbf{x}_t|\mathbf{x}_{t-1})q_\sigma(\mathbf{x}_{t-1}|\mathbf{x}_{t_a})}{q_\sigma(\mathbf{x}_t|\mathbf{x}_{t_a})}. \tag{35}$$

Substituting the transition probability for $t \geq t_a$, we obtain the conclusion of Lemma 2. $\square$

The transition probability of $p_{\boldsymbol{\theta}}$ follows $q_\sigma$ as $p_{\boldsymbol{\theta}}(\mathbf{x}_{t-1}|\mathbf{x}_t) = q_\sigma(\mathbf{x}_{t-1}|\boldsymbol{\mu}_{\boldsymbol{\theta}}(\mathbf{x}_t, t), \mathbf{x}_t)$ for $t \geq 2$, and $p_{\boldsymbol{\theta}}(\mathbf{x}_0|\mathbf{x}_1) = \mathcal{N}(\mathbf{x}_0; \boldsymbol{\mu}_{\boldsymbol{\theta}}(\mathbf{x}_1, 1), \tilde{\sigma}_1^2\mathbf{I})$.

### A.2   Proof of Proposition 1

*Proof.* Based on the above preliminaries, we derive the ELBO of $\log p_{\boldsymbol{\theta}}(\mathbf{x}_0, \mathbf{x}_{t_a})$ as follows:

$$\log p_{\boldsymbol{\theta}}(\mathbf{x}_0, \mathbf{x}_{t_a}) = \log \int \mathbf{dx}_{1:t\backslash t_a} \log p_{\boldsymbol{\theta}}(\mathbf{x}_{0:T}) \frac{q_\sigma(\mathbf{x}_{1:T\backslash t_a}|\mathbf{x}_0, \mathbf{x}_{t_a})}{q_\sigma(\mathbf{x}_{1:T\backslash t_a}|\mathbf{x}_0, \mathbf{x}_{t_a})}$$

$$\geq \int \mathbf{dx}_{1:t\backslash t_a} \log \left(p_{\boldsymbol{\theta}}(\mathbf{x}_T)\prod_{t=1}^T p_{\boldsymbol{\theta}}(\mathbf{x}_{t-1}|\mathbf{x}_t)\frac{q_\sigma(\mathbf{x}_{1:T\backslash t_a}|\mathbf{x}_0, \mathbf{x}_{t_a})}{q_\sigma(\mathbf{x}_T|\mathbf{x}_{t_a})\prod_{t=2}^{t_a} q_\sigma(\mathbf{x}_{t-1}|\mathbf{x}_t, \mathbf{x}_0)\prod_{t=t_a+2}^T q_\sigma(\mathbf{x}_{t-1}|\mathbf{x}_{t_a}, \mathbf{x}_t)}\right)$$

$$= \underbrace{\mathbb{E}_{q_\sigma(\mathbf{x}_1|\mathbf{x}_0, \mathbf{x}_{t_a})}\log p_{\boldsymbol{\theta}}(\mathbf{x}_0|\mathbf{x}_1) - \sum_{t=2}^{t_a}\mathbb{E}_{q_\sigma(\mathbf{x}_t|\mathbf{x}_{t_a}, \mathbf{x}_0)}D_{KL}\left(q_\sigma(\mathbf{x}_{t-1}|\mathbf{x}_t, \mathbf{x}_0)||p_{\boldsymbol{\theta}}(\mathbf{x}_{t-1}|\mathbf{x}_t)\right)}_{(1)}$$

$$\underbrace{+ \mathbb{E}_{q_\sigma(\mathbf{x}_{t_a+1}|\mathbf{x}_{t_a})}\log p_{\boldsymbol{\theta}}(\mathbf{x}_{t_a}|\mathbf{x}_{t_a+1}) - \sum_{t=t_a+2}^T D_{KL}(q_\sigma(\mathbf{x}_{t-1}|\mathbf{x}_t, \mathbf{x}_{t_a})||p_{\boldsymbol{\theta}}(\mathbf{x}_{t-1}|\mathbf{x}_t))}_{(2)}$$

$$- D_{KL}(q_\sigma(\mathbf{x}_T|\mathbf{x}_{t_a})||p_{\boldsymbol{\theta}}(\mathbf{x}_T)). \tag{36}$$

The KL divergence between $q_\sigma(\mathbf{x}_T|\mathbf{x}_{t_a})$ and $p_\theta(\mathbf{x}_T)$ is zero as they are both $\mathcal{N}(\mathbf{0}, \mathbf{I})$ by the setting of diffusion models. Now we handle (1) and (2) respectively.

(1) $p_\theta(\mathbf{x}_0|\mathbf{x}_1)$ is Gaussian and thus we have

$$\mathbb{E}_{q_\sigma(\mathbf{x}_1|\mathbf{x}_0,\mathbf{x}_{t_a})} \log p_\theta(\mathbf{x}_0|\mathbf{x}_1) = c_1 - \frac{1}{2\tilde{\sigma}_1^2} \mathbb{E}_{q_\sigma(\mathbf{x}_1|\mathbf{x}_0,\mathbf{x}_{t_a})} \|\mathbf{x}_0 - \boldsymbol{\mu}_\theta(\mathbf{x}_1, 1)\|^2, \qquad (37)$$

where by writing $c_i$ we denote the term independent of $\mathbf{x}_0$ and $\mathbf{x}_{t_a}$. $q_\sigma(\mathbf{x}_{t-1}|\mathbf{x}_t, \mathbf{x}_0)$ and $p_\theta(\mathbf{x}_{t-1}|\mathbf{x}_t)$ are also Gaussian, so by the KL divergence of the Gaussian distributions, we have (for $t \leq t_a$)

$$\mathbb{E}_{q_\sigma(\mathbf{x}_t|\mathbf{x}_0,\mathbf{x}_{t_a})} D_{KL}(q_\sigma(\mathbf{x}_{t-1}|\mathbf{x}_t, \mathbf{x}_{t_a})\|p_\theta(\mathbf{x}_{t-1}|\mathbf{x}_t))$$

$$= c_2 + \frac{1}{2\tilde{\sigma}_t^2} \mathbb{E}_{q_\sigma(\mathbf{x}_t|\mathbf{x}_0,\mathbf{x}_{t_a})} \left\| \sqrt{\overline{\alpha}_{t-1}} \mathbf{x}_0 + \sqrt{1 - \overline{\alpha}_{t-1} - \tilde{\sigma}_t^2} \frac{\mathbf{x}_t - \sqrt{\overline{\alpha}_t}\mathbf{x}_0}{\sqrt{1 - \overline{\alpha}_t}} - \sqrt{\overline{\alpha}_{t-1}} \boldsymbol{\mu}_\theta(\mathbf{x}_t, t) \right.$$

$$\left. - \sqrt{1 - \overline{\alpha}_{t-1} - \tilde{\sigma}_t^2} \frac{\mathbf{x}_t - \sqrt{\overline{\alpha}_t}\boldsymbol{\mu}_\theta(\mathbf{x}_t, t)}{\sqrt{1 - \overline{\alpha}_t}} \right\|^2 \qquad (38)$$

$$= c_2 + \frac{1}{2\tilde{\sigma}_t^2} \left( \sqrt{\overline{\alpha}_{t-1}} - \frac{\sqrt{1 - \overline{\alpha}_{t-1} - \tilde{\sigma}_t^2}}{\sqrt{1 - \overline{\alpha}_t}} \sqrt{\overline{\alpha}_t} \right)^2 \mathbb{E}_{q_\sigma(\mathbf{x}_t|\mathbf{x}_0,\mathbf{x}_{t_a})} \|\mathbf{x}_0 - \boldsymbol{\mu}_\theta(\mathbf{x}_t, t)\|^2.$$

(2) Similarly,

$$\mathbb{E}_{q_\sigma(\mathbf{x}_{t_a+1}|\mathbf{x}_{t_a})} \log p_\theta(\mathbf{x}_{t_a}|\mathbf{x}_{t_a+1})$$

$$= c_3 - \frac{1}{2\sigma_{t_a+1}^2} \mathbb{E}_{q_\sigma(\mathbf{x}_{t_a+1}|\mathbf{x}_{t_a})} \left\| \mathbf{x}_{t_a} - \sqrt{\overline{\alpha}_{t_a}} \boldsymbol{\mu}_\theta(\mathbf{x}_{t_a+1}, t_a + 1) \right.$$

$$\left. - \sqrt{1 - \overline{\alpha}_{t_a} - \sigma_{t_a+1}^2} \frac{\mathbf{x}_{t_a+1} - \sqrt{\overline{\alpha}_{t_a+1}}\boldsymbol{\mu}_\theta(\mathbf{x}_{t_a+1}, t_a + 1)}{\sqrt{1 - \overline{\alpha}_{t_a+1}}} \right\|^2, \qquad (39)$$

and for $t \geq t_a + 2$

$$\mathbb{E}_{q_\sigma(\mathbf{x}_t|\mathbf{x}_{t_a})} D_{KL}\left(q_\sigma(\mathbf{x}_{t-1}|\mathbf{x}_t, \mathbf{x}_{t_a})\|p_\theta(\mathbf{x}_{t-1}|\mathbf{x}_t)\right)$$

$$= c_4 + \frac{1}{2\sigma_t^2} \mathbb{E}_{q_\sigma(\mathbf{x}_t|\mathbf{x}_{t_a})} \left\| \sqrt{\overline{\alpha}_{t-1}/\overline{\alpha}_{t_a}} \mathbf{x}_{t_a} + \sqrt{1 - \overline{\alpha}_{t-1}/\overline{\alpha}_{t_a} - \hat{\sigma}_t^2} \frac{\mathbf{x}_t - \sqrt{\overline{\alpha}_t/\overline{\alpha}_{t_a}}\mathbf{x}_{t_a}}{\sqrt{1 - \overline{\alpha}_t/\overline{\alpha}_{t_a}}} \right.$$

$$\left. - \sqrt{\overline{\alpha}_{t-1}} \boldsymbol{\mu}_\theta(\mathbf{x}_t, t) - \sqrt{1 - \overline{\alpha}_{t-1} - \sigma_t^2} \frac{\mathbf{x}_t - \sqrt{\overline{\alpha}_t}\boldsymbol{\mu}_\theta(\mathbf{x}_t, t)}{\sqrt{1 - \overline{\alpha}_t}} \right\|^2. \qquad (40)$$

Here we apply the reparameterization trick in advance. We write

$$\mathbf{x}_t = \mathbf{x}_t(\boldsymbol{\epsilon}) = \sqrt{\overline{\alpha}_t/\overline{\alpha}_{t_a}} \mathbf{x}_{t_a} + \sqrt{1 - \overline{\alpha}_t/\overline{\alpha}_{t_a}} \boldsymbol{\epsilon}, \qquad (41)$$

where $\boldsymbol{\epsilon} \sim \mathcal{N}(\mathbf{0}, \mathbf{I})$, since $q_\sigma(\mathbf{x}_t|\mathbf{x}_{t_a}) = \mathcal{N}(\mathbf{x}_t; \sqrt{\overline{\alpha}_t/\overline{\alpha}_{t_a}}\mathbf{x}_{t_a}, (1 - \overline{\alpha}_t/\overline{\alpha}_{t_a})\mathbf{I})$ for $t \geq t_a + 1$. Substitute this into (39) and we have

$$\mathbb{E}_{q_\sigma(\mathbf{x}_{t_a+1}|\mathbf{x}_{t_a})} \log p_\theta(\mathbf{x}_{t_a}|\mathbf{x}_{t_a+1})$$

$$= c_3 + \mathbb{E}_{\boldsymbol{\epsilon} \sim \mathcal{N}(\mathbf{0},\mathbf{I})} \frac{1}{2\sigma_{t_a+1}^2} \left\| \left( 1 - \sqrt{\frac{\overline{\alpha}_{t_a+1}}{\overline{\alpha}_{t_a}}} \frac{\sqrt{1 - \overline{\alpha}_{t_a} - \sigma_{t_a+1}^2}}{\sqrt{1 - \overline{\alpha}_{t_a+1}}} \right) \left( \mathbf{x}_{t_a} - \sqrt{\overline{\alpha}_{t_a}} \boldsymbol{\mu}_\theta(\mathbf{x}_{t_a+1}(\boldsymbol{\epsilon}), t_a + 1) \right) \right.$$

$$\left. - \frac{\sqrt{1 - \overline{\alpha}_{t_a} - \sigma_{t_a+1}^2}}{\sqrt{1 - \overline{\alpha}_{t_a+1}}} \sqrt{1 - \frac{\overline{\alpha}_{t_a+1}}{\overline{\alpha}_{t_a}}} \boldsymbol{\epsilon} \right\|^2$$

$$= c_3' + \mathbb{E}_{q_\sigma(\mathbf{x}_{t_a+1}|\mathbf{x}_{t_a}),\boldsymbol{\epsilon}' \sim \mathcal{N}(\mathbf{0},\mathbf{I})} g(t_a + 1) \|\mathbf{x}_{t_a} - \left( \sqrt{\overline{\alpha}_{t_a}} \boldsymbol{\mu}_\theta(\mathbf{x}_{t_a+1}, t_a + 1) + \sqrt{1 - \overline{\alpha}_{t_a}} \boldsymbol{\epsilon}' \right) \|^2$$

$$+ \mathbb{E}_{q_\sigma(\mathbf{x}_{t_a+1}|\mathbf{x}_{t_a})} w(t_a + 1) \left\langle \boldsymbol{\mu}_\theta(\mathbf{x}_{t_a+1}, t_a + 1), \frac{\mathbf{x}_{t_a+1} - \sqrt{\overline{\alpha}_{t_a+1}/\overline{\alpha}_{t_a}}\mathbf{x}_{t_a}}{\sqrt{1 - \overline{\alpha}_{t_a+1}/\overline{\alpha}_{t_a}}} \right\rangle,$$

$$(42)$$

where

$$g(t_a + 1) = \frac{1}{2\sigma_{t_a+1}^2} \left( 1 - \sqrt{\frac{\overline{\alpha}_{t_a+1}}{\overline{\alpha}_{t_a}}} \sqrt{\frac{1 - \overline{\alpha}_{t_a} - \sigma_{t_a+1}^2}{1 - \overline{\alpha}_{t_a+1}}} \right)^2, \tag{43}$$

$$w(t_a + 1) = \frac{1}{\sigma_{t_a+1}^2} \left( 1 - \sqrt{\frac{\overline{\alpha}_{t_a+1}}{\overline{\alpha}_{t_a}}} \sqrt{\frac{1 - \overline{\alpha}_{t_a} - \sigma_{t_a+1}^2}{1 - \overline{\alpha}_{t_a+1}}} \right) \sqrt{\frac{1 - \overline{\alpha}_{t_a} - \sigma_{t_a+1}^2}{1 - \overline{\alpha}_{t_a+1}}} \sqrt{1 - \frac{\overline{\alpha}_{t_a+1}}{\overline{\alpha}_{t_a}}}. \tag{44}$$

And also (40) is (for $t \geq t_a + 2$)

$$\mathbb{E}_{q_\sigma(\mathbf{x}_t|\mathbf{x}_{t_a})} D_{KL}(q(\mathbf{x}_{t-1}|\mathbf{x}_t, \mathbf{x}_{t_a}) || p_{\boldsymbol\theta}(\mathbf{x}_{t-1}|\mathbf{x}_t))$$
$$= c_4' + g(t)\mathbb{E}_{q_\sigma(\mathbf{x}_t|\mathbf{x}_{t_a}), \boldsymbol\epsilon' \sim \mathcal{N}(\mathbf{0}, \mathbf{I})} || \mathbf{x}_{t_a} - \left( \sqrt{\overline{\alpha}_{t_a}} \boldsymbol\mu_{\boldsymbol\theta}(\mathbf{x}_t, t) + \sqrt{1 - \overline{\alpha}_{t_a}} \boldsymbol\epsilon' \right) ||^2$$
$$+ w(t)\mathbb{E}_{q_\sigma(\mathbf{x}_t|\mathbf{x}_{t_a})} \left\langle \boldsymbol\mu_{\boldsymbol\theta}(\mathbf{x}_t, t), \frac{\mathbf{x}_t - \sqrt{\overline{\alpha}_t/\overline{\alpha}_{t_a}} \mathbf{x}_{t_a}}{\sqrt{1 - \overline{\alpha}_t/\overline{\alpha}_{t_a}}} \right\rangle, \tag{45}$$

with

$$g(t) = \frac{1}{2\sigma_t^2} \left( \sqrt{\frac{\overline{\alpha}_{t-1}}{\overline{\alpha}_{t_a}}} - \sqrt{\frac{1 - \overline{\alpha}_{t-1} - \sigma_t^2}{1 - \overline{\alpha}_t}} \sqrt{\frac{\overline{\alpha}_t}{\overline{\alpha}_{t_a}}} \right)^2, \tag{46}$$

$$w(t) = \frac{1}{\sigma_t^2} \left( \sqrt{\overline{\alpha}_{t-1}} - \sqrt{\frac{1 - \overline{\alpha}_{t-1} - \sigma_t^2}{1 - \overline{\alpha}_t}} \sqrt{\overline{\alpha}_t} \right) \left( \sqrt{\frac{1 - \overline{\alpha}_{t-1} - \sigma_t^2}{1 - \overline{\alpha}_t}} \sqrt{1 - \frac{\overline{\alpha}_t}{\overline{\alpha}_{t_a}}} - \sqrt{1 - \frac{\overline{\alpha}_{t-1}}{\overline{\alpha}_{t_a}} - \hat{\sigma}_t^2} \right).$$
$$\tag{47}$$

Thus by integrate (37), (38), (42), and (45), we obtain the Variational Lower bound of $p_{\boldsymbol\theta}(\mathbf{x}_0, \mathbf{x}_{t_a})$ as follows:

$$\log p_{\boldsymbol\theta}(\mathbf{x}_0, \mathbf{x}_{t_a})$$
$$\geq C - \sum_{t=1}^{t_a} \mathbb{E}_{q_\sigma(\mathbf{x}_t|\mathbf{x}_0, \mathbf{x}_{t_a})} \left( g(t) || \mathbf{x}_0 - \boldsymbol\mu_{\boldsymbol\theta}(\mathbf{x}_t, t) ||^2 \right)$$
$$- \sum_{t=t_a+1}^{T} \mathbb{E}_{q_\sigma(\mathbf{x}_t|\mathbf{x}_{t_a}), \boldsymbol\epsilon' \sim \mathcal{N}(\mathbf{0}, \mathbf{I})} \left( g(t) \left\| \mathbf{x}_{t_a} - \left( \sqrt{\overline{\alpha}_{t_a}} \boldsymbol\mu_{\boldsymbol\theta}(\mathbf{x}_t, t) + \sqrt{1 - \overline{\alpha}_{t_a}} \boldsymbol\epsilon' \right) \right\|^2 \right. \tag{48}$$
$$\left. + w(t) \left\langle \boldsymbol\mu_{\boldsymbol\theta}(\mathbf{x}_t, t), \frac{\mathbf{x}_t - \sqrt{\overline{\alpha}_t/\overline{\alpha}_{t_a}} \mathbf{x}_{t_a}}{\sqrt{1 - \overline{\alpha}_t/\overline{\alpha}_{t_a}}} \right\rangle \right).$$

Here we reach the result of **Proposition 1.**, with the coefficient $g(t)$ is given by

$$g(t) = \begin{cases} \frac{1}{2\tilde{\sigma}_1^2} & t = 1; \\ \frac{1}{2\tilde{\sigma}_t^2} \left( \sqrt{\overline{\alpha}_{t-1}} - \sqrt{1 - \overline{\alpha}_{t-1} - \tilde{\sigma}_t^2} \frac{\sqrt{\overline{\alpha}_t}}{\sqrt{1 - \overline{\alpha}_t}} \right)^2 & 2 \leq t \leq t_a; \\ \frac{1}{2\sigma_{t_a+1}^2} \left( 1 - \sqrt{\frac{\overline{\alpha}_{t_a+1}}{\overline{\alpha}_{t_a}}} \sqrt{\frac{1 - \overline{\alpha}_{t_a} - \sigma_{t_a+1}^2}{1 - \overline{\alpha}_{t_a+1}}} \right)^2 & t = t_a + 1; \\ \frac{1}{2\sigma_t^2} \left( \sqrt{\frac{\overline{\alpha}_{t-1}}{\overline{\alpha}_{t_a}}} - \sqrt{\frac{1 - \overline{\alpha}_{t-1} - \sigma_t^2}{1 - \overline{\alpha}_t}} \sqrt{\frac{\overline{\alpha}_t}{\overline{\alpha}_{t_a}}} \right)^2 & t_a + 2 \leq t \leq T, \end{cases} \tag{49}$$

and $w(t)$ is give by

$$w(t) = \begin{cases} \frac{1}{\sigma_{t_a+1}^2} \left( 1 - \sqrt{\frac{\overline{\alpha}_{t_a+1}}{\overline{\alpha}_{t_a}}} \sqrt{\frac{1 - \overline{\alpha}_{t_a} - \sigma_{t_a+1}^2}{1 - \overline{\alpha}_{t_a+1}}} \right) \sqrt{\frac{1 - \overline{\alpha}_{t_a} - \sigma_{t_a+1}^2}{1 - \overline{\alpha}_{t_a+1}}} \sqrt{1 - \frac{\overline{\alpha}_{t_a+1}}{\overline{\alpha}_{t_a}}} & t = t_a + 1; \\ \frac{1}{\sigma_t^2} \left( \sqrt{\overline{\alpha}_{t-1}} - \sqrt{\frac{1 - \overline{\alpha}_{t_a} - \sigma_t^2}{1 - \overline{\alpha}_t}} \sqrt{\overline{\alpha}_t} \right) \left( \sqrt{\frac{1 - \overline{\alpha}_{t-1} - \sigma_t^2}{1 - \overline{\alpha}_t}} \sqrt{1 - \frac{\overline{\alpha}_t}{\overline{\alpha}_{t_a}}} - \sqrt{1 - \frac{\overline{\alpha}_{t-1}}{\overline{\alpha}_{t_a}} - \hat{\sigma}_t^2} \right) & t \geq t_a + 2. \end{cases} \tag{50}$$

$\square$

## A.3 Proof of Proposition 2

*Proof.* We continue the derivation by using the reparameterization trick. The reparameterization of (42) and (45) has been given above, we focus on (37) and (38) now. Applying Lemma 1, we can reparameterize them as

$$
\mathbb{E}_{q_\sigma(\mathbf{x}_t|\mathbf{x}_0,\mathbf{x}_{t_a})}||\mathbf{x}_0 - \boldsymbol{\mu}_{\boldsymbol{\theta}}(\mathbf{x}_t,t)||^2
$$
$$
= \mathbb{E}_{\boldsymbol{\epsilon}\sim\mathcal{N}(\mathbf{0},\mathbf{I})}\left\|\mathbf{x}_0 - \boldsymbol{\mu}_{\boldsymbol{\theta}}\left(\sqrt{\overline{\alpha}_t}\mathbf{x}_0 + \gamma_t\left(\mathbf{x}_{t_a} - \sqrt{\overline{\alpha}_{t_a}}\mathbf{x}_0\right)/\sqrt{1-\overline{\alpha}_{t_a}} + \zeta_t\boldsymbol{\epsilon},t\right)\right\|^2. \tag{51}
$$

We further disregard all the constants. The proxy objective for minimization can be rewritten as follows

$$
L(\mathbf{x}_0,\mathbf{x}_{t_a}) = \sum_{t=1}^{t_a}\mathbb{E}_{\boldsymbol{\epsilon}\sim\mathcal{N}(\mathbf{0},\mathbf{I})}g(t)\left\|\mathbf{x}_0 - \boldsymbol{\mu}_{\boldsymbol{\theta}}\left(\sqrt{\overline{\alpha}_t}\mathbf{x}_0 + \gamma_t\frac{\mathbf{x}_{t_a} - \sqrt{\overline{\alpha}_{t_a}}\mathbf{x}_0}{\sqrt{1-\overline{\alpha}_{t_a}}} + \zeta_t\boldsymbol{\epsilon},t\right)\right\|^2
$$
$$
+ \sum_{t=t_a+1}^{T}\mathbb{E}_{\boldsymbol{\epsilon},\boldsymbol{\epsilon}'\sim\mathcal{N}(\mathbf{0},\mathbf{I})}g(t)\left(\left\|\mathbf{x}_b - \left(\sqrt{\overline{\alpha}_{t_a}}\boldsymbol{\mu}_{\boldsymbol{\theta}}\left(\sqrt{\frac{\overline{\alpha}_t}{\overline{\alpha}_{t_a}}}\mathbf{x}_{t_a} + \sqrt{1-\frac{\overline{\alpha}_t}{\overline{\alpha}_{t_a}}}\boldsymbol{\epsilon},t\right) + \sqrt{1-\overline{\alpha}_{t_a}}\boldsymbol{\epsilon}'\right)\right\|^2\right.
$$
$$
\left.+ \frac{w(t)}{g(t)}\left\langle\boldsymbol{\mu}_{\boldsymbol{\theta}}\left(\sqrt{\overline{\alpha}_t/\overline{\alpha}_{t_a}}\mathbf{x}_{t_a} + \sqrt{1-\overline{\alpha}_t/\overline{\alpha}_{t_a}}\boldsymbol{\epsilon},t\right),\boldsymbol{\epsilon}\right\rangle\right). \tag{52}
$$

Now we seek for stochastic gradients of $L$. Sampling $t\sim\mathcal{U}\{1,2,\ldots,T\}$ and $\boldsymbol{\epsilon},\boldsymbol{\epsilon}'\in\mathcal{N}(\mathbf{0},\mathbf{I})$, the stochastic gradient with respect to $\mathbf{x}_0$ and $\mathbf{x}_{t_a}$ are, respectively,

$$
\mathbf{d}_{\mathbf{x}_0,t} = \begin{cases}\nabla_{\mathbf{x}_0}\left\|\mathbf{x}_0 - \boldsymbol{\mu}_{\boldsymbol{\theta}}\left(\sqrt{\overline{\alpha}_t}\mathbf{x}_0 + \gamma_t(\mathbf{x}_{t_a} - \sqrt{\overline{\alpha}_{t_a}}\mathbf{x}_0)/\sqrt{1-\overline{\alpha}_{t_a}} + \zeta_t\boldsymbol{\epsilon},t\right)\right\|^2 & t\le t_a; \\ \mathbf{0} & t>t_a,\end{cases} \tag{53}
$$

and

$$
\mathbf{d}_{\mathbf{x}_b,t} = \begin{cases}\nabla_{\mathbf{x}_{t_a}}\left\|\mathbf{x}_0 - \boldsymbol{\mu}_{\boldsymbol{\theta}}\left(\sqrt{\overline{\alpha}_t}\mathbf{x}_0 + \gamma_t(\mathbf{x}_{t_a} - \sqrt{\overline{\alpha}_{t_a}}\mathbf{x}_0)/\sqrt{1-\overline{\alpha}_{t_a}} + \zeta_t\boldsymbol{\epsilon},t\right)\right\|^2 & t\le t_a; \\ \nabla_{\mathbf{x}_{t_a}}\left\|\mathbf{x}_{t_a} - \left(\sqrt{\overline{\alpha}_{t_a}}\boldsymbol{\mu}_{\boldsymbol{\theta}}\left(\sqrt{\overline{\alpha}_t/\overline{\alpha}_{t_a}}\mathbf{x}_{t_a} + \sqrt{1-\overline{\alpha}_t/\overline{\alpha}_{t_a}}\boldsymbol{\epsilon},t\right) + \sqrt{1-\overline{\alpha}_{t_a}}\boldsymbol{\epsilon}'\right)\right\|^2 & \\ \quad + w(t)/g(t)\nabla_{\mathbf{x}_{t_a}}\left\langle\boldsymbol{\mu}_{\boldsymbol{\theta}}\left(\sqrt{\overline{\alpha}_t/\overline{\alpha}_{t_a}}\mathbf{x}_{t_a} + \sqrt{1-\overline{\alpha}_t/\overline{\alpha}_{t_a}}\boldsymbol{\epsilon},t\right),\boldsymbol{\epsilon}\right\rangle & t>t_a,\end{cases} \tag{54}
$$

where the coefficient $g(t)$ is omitted as it can be scaled into the step size. Here we reach the result of **Proposition 2** in the main text. $\square$

## A.4 Gradient truncation and remarks

In practice, we truncate the gradient through $\boldsymbol{\mu}_{\boldsymbol{\theta}}$, i.e., we let

$$
\mathbf{D}_{\mathbf{x}_0}\boldsymbol{\mu}_{\boldsymbol{\theta}}(\cdot) = \mathbf{D}_{\mathbf{x}_{t_a}}\boldsymbol{\mu}_{\boldsymbol{\theta}}(\cdot) = \mathbf{0}. \tag{55}
$$

Thus the approximate stochastic gradients are, respectively,

$$
\tilde{\mathbf{d}}_{\mathbf{x}_0,t} = \begin{cases}\mathbf{x}_0 - \boldsymbol{\mu}_{\boldsymbol{\theta}}\left(\sqrt{\overline{\alpha}_t}\mathbf{x}_0 + \gamma_t(\mathbf{x}_{t_a} - \sqrt{\overline{\alpha}_{t_a}}\mathbf{x}_0)/\sqrt{1-\overline{\alpha}_{t_a}} + \zeta_t\boldsymbol{\epsilon},t\right) & t\le t_a; \\ \mathbf{0} & t>t_a,\end{cases} \tag{56}
$$

and

$$
\tilde{\mathbf{d}}_{\mathbf{x}_{t_a},t} = \begin{cases}\mathbf{0} & t\le t_a; \\ \mathbf{x}_{t_a} - (\sqrt{\overline{\alpha}_{t_a}}\boldsymbol{\mu}_{\boldsymbol{\theta}}(\sqrt{\overline{\alpha}_t/\overline{\alpha}_{t_a}}\mathbf{x}_{t_a} + \sqrt{1-\overline{\alpha}_t/\overline{\alpha}_{t_a}}\boldsymbol{\epsilon},t) + \sqrt{1-\overline{\alpha}_{t_a}}\boldsymbol{\epsilon}') & t>t_a.\end{cases} \tag{57}
$$

At this point, we obtain the approximate stochastic gradients of the objective function with respect to the two variables $\mathbf{x}_0$ and $\mathbf{x}_{t_a}$. We can perform gradient descent on $\mathbf{x}_0$ and the projection gradient method on $\mathbf{x}_{t_a}$ to solve the problem.

**Remark:**

1. The idea of introducing a new noise $\boldsymbol{\epsilon}'$ is to ensure that the update direction of $\mathbf{x}_{t_a}$ remains as much as possible within the manifold at time step $t_a$. The expression $-(\mathbf{x}_{t_a} - \sqrt{\overline{\alpha}_{t_a}}\boldsymbol{\mu}_{\boldsymbol{\theta}}(\cdot) - \sqrt{1-\overline{\alpha}_{t_a}}\boldsymbol{\epsilon}'))$

can be viewed as a vector pointing from $\mathbf{x}_{t_a}$ to $\sqrt{\overline{\alpha}_{t_a}}\boldsymbol{\mu}_{\boldsymbol{\theta}}(\cdot) + \sqrt{1-\overline{\alpha}_{t_a}}\boldsymbol{\epsilon}'$. As $\boldsymbol{\mu}_{\boldsymbol{\theta}}(\cdot)$ is close to the manifold of $\mathbf{x}_0$, the introduction of $\boldsymbol{\epsilon}'$ allows the endpoint of the gradient vector to be close to the manifold at time $t_a$.

2. For a general linear observation equation, the derivation remains similar when transforming the observations into the spectral domain using SVD. Due to potentially differing noise levels for each component, the objective function will take on an element-wise form. This is based on the fact that in the diffusion model, we assume that the elements of $\mathbf{x}_{t-1}$ are independent when conditioning on $\mathbf{x}_t$.

3. In practice, we find that setting the DDIM variance $\tilde{\sigma}_t = 0$ for $t \leq t_a$ yields the best results, which results in $\gamma_t = \sqrt{1-\overline{\alpha}_t}$ and $\zeta_t = 0$. But this poses a problem as it leads to $g(t) \to \infty$. In this work, we still scale all the coefficients $g(t)$ into the step sizes as it works well in the experiments. We leave this issue for future research.

4. Note that one may find $t_a$ is not an integer in practice, as $\overline{\alpha}_{t_a}$ may not be in the discrete series $\overline{\alpha}_{0:T}$. This won't affect our algorithm, as it can be derived similarly that the approximate stochastic gradients in (56) and (57) still hold. The only difference lies in the initialization of $\mathbf{x}_0$ which will be discussed in Appendix F.

5. We use the $\boldsymbol{\mu}$-predictor in the derivation. The case of $\boldsymbol{\epsilon}$-predictor is similar, with the conversion given by $\boldsymbol{\mu}_{\boldsymbol{\theta}}(\mathbf{x}_t) = \left(\mathbf{x}_t - \sqrt{1-\overline{\alpha}_t}\boldsymbol{\epsilon}_{\boldsymbol{\theta}}(\mathbf{x}_t)\right)/\sqrt{\overline{\alpha}_t}$. We refer to [19, 32] for more details about the relation between the $\boldsymbol{\mu}$-predictor, the $\boldsymbol{\epsilon}$-predictor and the Stein score [51].

# B   ProjDiff for VE diffusion

In this section, we derive the ProjDiff algorithm for the Variance Exploding (VE) diffusion used in the source separation and partial generation tasks. The experiments using VE diffusion in this work are all noise-free, thus we only focus on the noise-free scenarios and leave the noisy scenarios of VE diffusion for future work.

Consider the forward transition probability $q(\mathbf{x}_t|\mathbf{x}_{t-1}) = \mathcal{N}(\mathbf{x}_t; \mathbf{x}_{t-1}, (\overline{\sigma}_t^2 - \overline{\sigma}_{t-1}^2)\mathbf{I})$, with $\overline{\sigma}_T > \overline{\sigma}_{T-1} > \cdots > \overline{\sigma}_1 > \overline{\sigma}_0 \to 0$. When $\overline{\sigma}_T$ is large enough, $q(\mathbf{x}_T) \approx \mathcal{N}(\mathbf{0}, \sigma_T^2\mathbf{I})$. The initial distribution of the reverse process is set to be $p_{\boldsymbol{\theta}}(\mathbf{x}_T) = \mathcal{N}(\mathbf{0}, \sigma_T^2\mathbf{I}) \approx q(\mathbf{x}_T)$ and the reverse transition in the DDPM manner is

$$p_{\boldsymbol{\theta}}(\mathbf{x}_{t-1}|\mathbf{x}_t) = \mathcal{N}\left(\mathbf{x}_{t-1}; \boldsymbol{\mu}_{\boldsymbol{\theta}}(\mathbf{x}_t, t) + \frac{\overline{\sigma}_{t-1}^2}{\overline{\sigma}_t^2}(\mathbf{x}_t - \boldsymbol{\mu}_{\boldsymbol{\theta}}(\mathbf{x}_t, t)), \frac{\overline{\sigma}_{t-1}^2(\overline{\sigma}_t^2 - \overline{\sigma}_{t-1}^2)}{\overline{\sigma}_t^2}\mathbf{I}\right), \quad (58)$$

for $t \geq 2$, which matches the condition probability

$$q(\mathbf{x}_{t-1}|\mathbf{x}_t, \mathbf{x}_0) = \mathcal{N}\left(\mathbf{x}_{t-1}; \mathbf{x}_0 + \frac{\overline{\sigma}_{t-1}^2}{\overline{\sigma}_t^2}(\mathbf{x}_t - \mathbf{x}_0), \frac{\overline{\sigma}_{t-1}^2(\overline{\sigma}_t^2 - \overline{\sigma}_{t-1}^2)}{\overline{\sigma}_t^2}\mathbf{I}\right). \quad (59)$$

And we assume the last sampling step is set to be

$$p_{\boldsymbol{\theta}}(\mathbf{x}_0|\mathbf{x}_1) = \mathcal{N}(\mathbf{x}_0; \boldsymbol{\mu}_{\boldsymbol{\theta}}(\mathbf{x}_1, 1), \delta_0^2\mathbf{I}), \quad (60)$$

for some small $\delta_0$.

Thus the ELBO is derived as the same in DDPM [2].

$$\begin{aligned}
\log p_{\boldsymbol{\theta}}(\mathbf{x}_0) &= \log \int_{t=1}^T p_{\boldsymbol{\theta}}(\mathbf{x}_{0:T})\frac{q(\mathbf{x}_{1:T}|\mathbf{x}_0)}{q(\mathbf{x}_{1:T}|\mathbf{x}_0)}\mathbf{dx}_{1:T} \\
&\geq c - D_{KL}(q(\mathbf{x}_T|\mathbf{x}_0)||p_{\boldsymbol{\theta}}(\mathbf{x}_T)) + \mathbb{E}_{q(\mathbf{x}_1|\mathbf{x}_0)}\log p_{\boldsymbol{\theta}}(\mathbf{x}_0|\mathbf{x}_1) \\
&\quad - \sum_{t=2}^T \mathbb{E}_{q(\mathbf{x}_t|\mathbf{x}_0)}D_{KL}(q(\mathbf{x}_{t-1}|\mathbf{x}_t, \mathbf{x}_0)||p_{\boldsymbol{\theta}}(\mathbf{x}_{t-1}|\mathbf{x}_t)).
\end{aligned} \quad (61)$$

$D_{KL}(q(\mathbf{x}_T|\mathbf{x}_0)||p_{\boldsymbol{\theta}}(\mathbf{x}_T)) \approx 0$, and $\log p_{\boldsymbol{\theta}}(\mathbf{x}_0|\mathbf{x}_1) = c_1 - \frac{1}{2\delta_0^2}||\mathbf{x}_0 - \boldsymbol{\mu}_{\boldsymbol{\theta}}(\mathbf{x}_1, 1)||^2$. The KL divergence between $q(\mathbf{x}_{t-1}|\mathbf{x}_t, \mathbf{x}_0)$ and $p_{\boldsymbol{\theta}}(\mathbf{x}_{t-1}|\mathbf{x}_t)$ is

$$D_{KL}(q(\mathbf{x}_{t-1}|\mathbf{x}_t, \mathbf{x}_0)|p_{\boldsymbol{\theta}}(\mathbf{x}_{t-1}|\mathbf{x}_t)) = c_2 + \frac{\overline{\sigma}_t^2 - \overline{\sigma}_{t-1}^2}{2\overline{\sigma}_{t-1}^2\overline{\sigma}_t^2}||\mathbf{x}_0 - \boldsymbol{\mu}_{\boldsymbol{\theta}}(\mathbf{x}_t, t)||^2, \quad (62)$$

Table 5: Noise-free restoration on ImageNet dataset. LPIPS metrics are multiplied by 100.

| NFEs | Method | Super-Resolution PSNR↑ SSIM↑ LPIPS↓ FID↓ | Inpainting PSNR↑ SSIM↑ LPIPS↓ FID↓ | Gaussian Deblurring PSNR↑ SSIM↑ LPIPS↓ FID↓ |
|---|---|---|---|---|
| - | $A^\dagger \mathbf{y}$ | 24.22 / 0.70 / 45.14 / 130.30 | 14.54 / 0.30 / 65.90 / 169.65 | 37.19 / 0.95 / 12.15 / 14.77 |
| 1000 | DPS | 26.14 / 0.76 / 24.17 / **28.46** | 32.97 / 0.92 / 9.74 / 14.08 | 27.21 / 0.75 / 29.59 / 39.93 |
| 100 | DDRM | 27.08 / **0.79** / 24.97 / 38.36 | 31.33 / 0.91 / 11.23 / 13.03 | 41.52 / 0.98 / 4.47 / 2.98 |
| 100 | DDNM | 27.07 / **0.79** / 23.87 / 33.45 | 31.63 / 0.91 / 9.19 / 9.57 | 43.47 / **0.99** / 2.52 / 1.59 |
| 100 | RED-diff | **27.23** / **0.79** / 26.13 / 42.23 | 9.81 / 0.17 / 83.58 / 268.93 | 28.65 / 0.83 / 26.04 / 35.76 |
| 100 | **ProjDiff** | 27.09 / **0.79** / **23.42** / 32.95 | **33.19 / 0.94 / 7.16 / 7.60** | **44.17 / 0.99 / 2.22 / 1.35** |

for $t \geq 2$. Then by reparameterization, we get the Evidence Lower Bound as

$$\text{ELBO}_{\text{VE}} = C - \sum_{t=1}^{T} s(t) \mathbb{E}_{\boldsymbol{\epsilon} \sim \mathcal{N}(\mathbf{0}, \mathbf{I})} ||\mathbf{x}_0 - \boldsymbol{\mu_\theta}(\mathbf{x}_0 + \overline{\sigma}_t \boldsymbol{\epsilon}, t)||^2, \tag{63}$$

with

$$s(t) = \begin{cases} \frac{1}{2\delta_0^2} & t = 1; \\ \frac{\overline{\sigma}_t^2 - \overline{\sigma}_{t-1}^2}{2\overline{\sigma}_{t-1}^2 \overline{\sigma}_t^2} & 2 \leq t \leq T. \end{cases} \tag{64}$$

Sampling $t \sim \mathcal{U}\{1, 2, \ldots, T\}$, $\boldsymbol{\epsilon} \sim \mathcal{N}(\mathbf{0}, \mathbf{I})$, and truncating the gradient through $\boldsymbol{\mu_\theta}$ as in the VP setting, a stochastic gradient for $-\text{ELBO}_{\text{VE}}$ (for minimization) is

$$\tilde{\mathbf{d}}_{\mathbf{x}_0, t} = \mathbf{x}_0 - \boldsymbol{\mu_\theta}(\mathbf{x}_0 + \overline{\sigma}_t \boldsymbol{\epsilon}, t). \tag{65}$$

Then the iteration step of the ProjDiff (noise-free) in the VE setting is

$$(\text{ProjDiff for VE}) \ \mathbf{x}_0 \leftarrow \mathcal{P}_{\mathbf{A}, \mathbf{y}} \left( \mathbf{x}_0 - \eta \tilde{\mathbf{d}}_{\mathbf{x}_0, t} \right), \tag{66}$$

where $\eta$ is the step size.

The RED-diff [24] algorithm in the VE setting can be derived similarly. In the source separation and partial generation tasks, we use these forms as the implementation of our ProjDiff and RED-diff [24] algorithms.

**Remark:** For VE diffusion, the Restricted Encoding method involves fixing the initial noise $\boldsymbol{\epsilon}_0 \sim \mathcal{N}(\mathbf{0}, \mathbf{I})$ and reparameterizing the sampling of $\mathbf{x}_t$ as $\mathbf{x}_t = \mathbf{x}_0 + \xi \boldsymbol{\epsilon}_0 + \sqrt{\overline{\sigma}_t^2 - \xi^2} \boldsymbol{\epsilon}$ for $\boldsymbol{\epsilon} \sim \mathcal{N}(\mathbf{0}, \mathbf{I})$. Specifically, we set $\xi = \overline{\sigma}_{t-1}$ in partial generation tasks.

## C  Additional experiments for image restoration

### C.1  Additional results

Here, we present the omitted experiments for image restoration tasks. The results for noise-free restoration tasks on ImageNet are shown in Table 5, the results for noise-free and noisy restoration tasks on CelebA are shown in Table 6 and 7, respectively, and the results for nonlinear high dynamic range (HDR) task are in Table 8. On the CelebA dataset, we further compare ProjDiff with more diffusion based rithms, including ΠGDM [22], DMPS [52], Resample [34], and DiffPIR [53]. ProjDiff demonstrates highly competitive performance on all these tasks.

### C.2  Ablation study

**Metrics balance in the super-resolution task.** An interesting observation is the trade-off between objective metrics and perceptual metrics in the super-resolution task. DPS achieves the best perceptual metrics but performs poorly on objective metrics, while RED-diff performs well in objective metrics

Table 6: Noise-free restoration on CelebA dataset. LPIPS metrics are multiplied by 100.

| NFEs | Method | Super-Resolution PSNR ↑ SSIM ↑ LPIPS ↓ FID ↓ | Inpainting PSNR ↑ SSIM ↑ LPIPS ↓ FID ↓ | Gaussian Deblurring PSNR ↑ SSIM ↑ LPIPS ↓ FID ↓ |
|---|---|---|---|---|
| - | $A^\dagger \mathbf{y}$ | 27.23 / 0.80 / 47.03 / 110.11 | 13.96 / 0.25 / 75.54 / 230.61 | 18.85 / 0.63 / 34.51 / 54.77 |
| 1000 | DPS | 30.25 / 0.86 / 16.88 / 35.65 | 36.02 / 0.95 / 8.79 / 24.73 | 35.55 / 0.94 / 11.27 / 25.45 |
| 100 | DDRM | 31.38 / 0.88 / 15.48 / 34.12 | 35.77 / 0.95 / 9.03 / 21.62 | 42.41 / 0.98 / 5.03 / 7.73 |
| 100 | DDNM | 31.39 / 0.88 / **14.45** / **26.17** | 36.32 / **0.96** / 6.86 / 12.45 | 45.36 / **0.99** / 3.01 / 2.08 |
| 100 | RED-diff | 32.44 / **0.90** / 16.58 / 30.49 | 7.94 / 0.18 / 78.23 / 192.07 | 33.07 / 0.90 / 17.02 / 20.54 |
| 100 | ΠGDM | 30.47 / 0.87 / 16.01 / 34.27 | 36.31 / **0.96** / 8.71 / 24.27 | 40.69 / 0.98 / 5.92 / 17.38 |
| 100 | DMPS | 30.94 / 0.87 / 17.25 / 31.87 | 32.30 / 0.89 / 18.58 / 31.30 | 42.84 / 0.98 / 4.27 / 3.86 |
| 100 | Resample | 31.62 / 0.88 / 20.11 / 44.18 | 35.00 / 0.93 / 13.03 / 30.87 | 33.60 / 0.91 / 17.45 / 37.70 |
| 100 | DiffPIR | 31.30 / 0.88 / 15.47 / 32.68 | 35.96 / 0.95 / 7.85 / 14.67 | 27.87 / 0.73 / 32.33 / 50.18 |
| 100 | **ProjDiff** | **32.57** / **0.90** / 16.08 / 33.74 | **36.84** / **0.96** / **6.66** / **12.27** | **45.57** / **0.99** / **2.99** / **2.05** |

Table 7: Noisy restoration on CelebA dataset with $\sigma = 0.05$. LPIPS metrics are multiplied by 100.

| NFEs | Method | Super-Resolution PSNR ↑ SSIM ↑ LPIPS ↓ FID ↓ | Inpainting PSNR ↑ SSIM ↑ LPIPS ↓ FID ↓ | Gaussian Deblurring PSNR ↑ SSIM ↑ LPIPS ↓ FID ↓ |
|---|---|---|---|---|
| - | $A^\dagger \mathbf{y}$ | 23.64 / 0.49 / 68.72 / 147.89 | 13.70 / 0.19 / 76.07 / 226.28 | 18.06 / 0.33 / 61.51 / 93.90 |
| 1000 | DPS | 27.98 / 0.78 / 23.10 / 39.91 | 32.80 / 0.90 / 16.32 / 32.80 | 29.46 / 0.81 / 21.19 / 38.54 |
| 100 | DDRM | 29.20 / 0.82 / 21.92 / 40.14 | 32.81 / 0.90 / 16.78 / 35.28 | 30.51 / 0.85 / 19.89 / 38.24 |
| 100 | DDNM+ | 29.19 / 0.82 / 21.89 / 40.20 | 32.80 / 0.90 / 16.80 / 35.13 | 30.60 / 0.85 / 19.79 / 38.23 |
| 100 | RED-diff | 24.98 / 0.55 / 50.59 / 73.89 | 7.96 / 0.18 / 78.27 / 192.32 | 26.32 / 0.56 / 41.23 / 56.59 |
| 100 | ΠGDM | 28.25 / 0.79 / 22.73 / 38.70 | 32.68 / 0.90 / 16.02 / 31.54 | 27.57 / 0.76 / 23.76 / 38.61 |
| 100 | DMPS | 29.01 / 0.81 / 22.87 / 37.80 | 32.18 / 0.87 / 18.77 / **23.88** | 30.51 / 0.84 / 20.32 / **33.63** |
| 100 | Resample | **29.58** / **0.83** / 24.18 / 45.51 | 33.11 / 0.90 / 15.97 / 31.38 | 30.89 / 0.85 / 21.91 / 38.61 |
| 100 | DiffPIR | 27.87 / 0.73 / 32.33 / 50.18 | 29.71 / 0.74 / 27.14 / 32.11 | 26.93 / 0.62 / 38.07 / 53.01 |
| 100 | **ProjDiff** | 29.49 / **0.83** / **20.86** / 36.87 | **33.43** / **0.91** / **15.33** / 31.43 | **31.41** / **0.87** / **18.12** / 34.59 |

Table 8: HDR results. For noisy observation, the standard deviation is $\sigma = 0.1$. The LPIPS metric is multiplied by 100.

| NFEs | Method | HDR $\sigma = 0$ PSNR↑ SSIM↑ LPIPS↓ FID↓ | HDR $\sigma = 0.1$ PSNR↑ SSIM↑ LPIPS↓ FID↓ |
|---|---|---|---|
| 1000 | DPS | 15.69 / 0.43 / 51.22 / 193.15 | 15.84 / 0.43 / 51.26 / 188.68 |
| 1000 | RED-diff | 26.01 / **0.87** / 23.60 / 34.03 | 22.64 / 0.66 / 38.48 / 55.31 |
| 1000 | **ProjDiff** | **28.65** / **0.87** / **16.26** / **18.44** | **25.71** / **0.84** / **25.91** / **32.87** |

Table 9: Noise-free restoration with 20 steps on ImageNet. The LPIPS metrics are multiplied by 100.

| NFEs | Method | Super-Resolution PSNR↑ SSIM↑ LPIPS↓ FID↓ | Inpainting PSNR↑ SSIM↑ LPIPS↓ FID↓ | Gaussian deblur PSNR↑ SSIM↑ LPIPS↓ FID↓ |
|---|---|---|---|---|
| 20 | DDRM | 26.54 / 0.77 / 25.88 / 40.80 | 28.63 / 0.86 / 20.85 / 30.34 | 40.46 / **0.98** / 5.68 / 4.46 |
| 20 | DDNM | 26.49 / 0.77 / **24.27** / **33.99** | 29.50 / 0.88 / 16.09 / 21.01 | 42.11 / **0.98** / **3.72** / **2.87** |
| 20 | **ProjDiff** | **26.83** / **0.78** / 25.19 / 37.59 | **30.46** / **0.90** / **15.41** / **20.09** | **42.24** / **0.98** / 3.78 / 2.99 |

Table 10: Noisy restoration with 20 steps on ImageNet ($\sigma = 0.05$). The LPIPS metrics are multiplied by 100.

| NFEs | Method | Super-Resolution PSNR↑ SSIM↑ LPIPS↓ FID↓ | Inpainting PSNR↑ SSIM↑ LPIPS↓ FID↓ | Gaussian deblur PSNR↑ SSIM↑ LPIPS↓ FID↓ |
|---|---|---|---|---|
| 20 | DDRM | 25.20 / 0.69 / 35.59 / 53.45 | 27.55 / 0.80 / 24.76 / 34.15 | 27.70 / 0.78 / 29.35 / 45.24 |
| 20 | DDNM+ | 25.23 / 0.70 / 35.06 / **52.12** | 27.56 / 0.80 / 24.71 / 34.24 | 27.70 / 0.77 / 30.78 / 50.63 |
| 20 | **ProjDiff** | **25.76** / **0.72** / **34.88** / 55.71 | **29.27** / **0.85** / **19.43** / **23.89** | **27.87** / **0.79** / **24.49** / **30.53** |

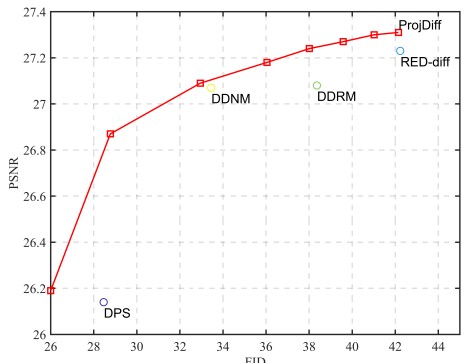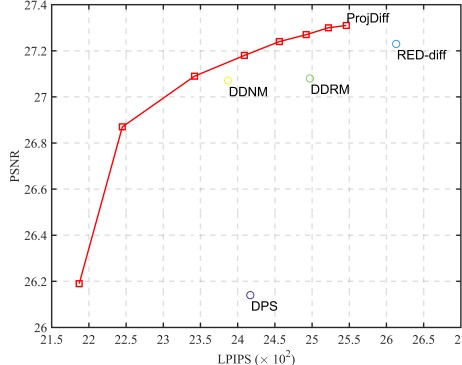

Figure 4: Noise-free super-resolution results on ImageNet. The red lines show the variation of PSNR v.s. FID and LPIPS of ProjDiff algorithm.

Table 11: Phase retrieval results. The LPIPS metrics are multiplied by 100.

| NFEs | Method | Phase Retrieval $\sigma = 0$ | Phase Retrieval $\sigma = 0.1$ |
|---|---|---|---|
| | | PSNR↑ SSIM↑ LPIPS↓ FID↓ | PSNR↑ SSIM↑ LPIPS↓ FID↓ |
| - | ER-4trials | 11.36 / 0.20 / 83.95 / 410.75 | 11.32 / 0.20 / 83.87 / 410.53 |
| - | HIO-4trials | 12.60 / 0.26 / 79.78 / 336.94 | 12.62 / 0.26 / 79.88 / 333.90 |
| - | OSS-4trials | 12.75 / 0.26 / 79.95 / 342.36 | 12.69 / 0.25 / 80.37 / 344.84 |
| 1000 | DPS-4trials | 19.36 / 0.45 / 48.48 / 104.70 | 17.25 / 0.38 / 51.83 / 82.10 |
| 1000 | **ProjDiff-4trials** | **41.58 / 0.94 / 5.49 / 7.19** | **28.11 / 0.76 / 28.43 / 53.38** |

for noise-free super-resolution but declines on perceptual metrics. We find that in the noise-free super-resolution task, the step size of ProjDiff controls the balance between objective and perceptual metrics. Figure 4 presents ProjDiff's FID-PSNR curve and LPIPS-PSNR curve in the super-resolution task on ImageNet, with step sizes ranging from $[1.2, 1.9]$. Note that lower values of LPIPS and FID and higher values of PSNR indicate better restoration results, thus the top left corner represents the ideal performance. The red lines demonstrate the variation in objective and perceptual metrics of ProjDiff by adjusting the step size. Also note that ProjDiff exhibits superior performance compared to DDRM, DDNM, DPS and RED-diff.

**Fewer steps.** DDRM [20] and DDNM [21] can both address image restoration with fewer steps (20 steps). Here we test the performance of the proposed ProjDiff with fewer steps on ImageNet. Table 9 and Table 10 show comparisons of ProjDiff with DDRM and DDNM in three linear restoration tasks under noise-free and noisy observations, respectively. ProjDiff also demonstrates highly competitive performance across multiple metrics, indicating its potential to perform image restoration with fewer steps.

**Four trials for the phase retrieval task.** It is noted in [23] that in the phase retrieval task, DPS requires sampling four times for each image and selecting the best result to achieve satisfactory performance. Therefore, we compare the performance of DPS and ProjDiff with four independent samplings. We also report the performance of the ER, HIO, and OSS algorithms with four samplings as baselines. The best image is selected as the sample with the highed PSNR metric. The results are shown in Table 11. Both DPS and ProjDiff show performance improvements with four trials, with ProjDiff in particular exhibiting much better performance, reaching a PSNR of 41.58dB in the noise-free case.

**The impact of the gradient truncation method used in ProjDiff.** We compared ProjDiff with and without the gradient truncation method on the CelebA dataset, as shown in Table 12. ProjDiff without gradient truncation is denoted as ProjDiff-FG. The results indicates that there is some performance loss when using gradient truncation compared to not using it, while the efficiency increases by nearly three times. Considering that using gradient truncation has already achieved satisfactory performance, we accept the trade-off of some performance loss for the gain in efficiency.

Table 12: Ablation study of the gradient truncation method in ProjDiff on noise-free tasks on CelebA.

| Time (s) | Method | Super-Resolution PSNR ↑ SSIM ↑ LPIPS ↓ FID ↓ | Inpainting PSNR ↑ SSIM ↑ LPIPS ↓ FID ↓ | Gaussian Deblurring PSNR ↑ SSIM ↑ LPIPS ↓ FID ↓ |
|---|---|---|---|---|
| 4.42 | ProjDiff | 32.57 / 0.90 / 16.08 / 33.74 | 36.84 / 0.96 / 6.66 / 12.27 | 45.57 / 0.99 / 2.99 / 2.05 |
| 11.13 | ProjDiff-FG | 33.05 / 0.91 / 16.87 / 36.39 | 37.30 / 0.96 / 7.12 / 11.65 | 45.98 / 0.99 / 2.72 / 1.94 |

Table 13: Using gradient descent to approximate the projection operator.

| NFEs | Method | **Phase Retrieval** $\sigma = 0$ PSNR↑ SSIM↑ LPIPS↓ FID↓ | **Phase Retrieval** $\sigma = 0.1$ PSNR↑ SSIM↑ LPIPS↓ FID↓ |
|---|---|---|---|
| 1000 | ProjDiff | 33.39 / 0.74 / 20.42 / 35.91 | 23.84 / 0.60 / 38.55 / 76.20 |
| 1000 | ProjDiff-GD | 33.55 / 0.71 / 23.04 / 45.44 | 23.08 / 0.56 / 44.86 / 97.13 |

**Using gradient descent to approximate the projection operation for nonlinear tasks.** We claim that for general nonlinear tasks, ProjDiff can be applied by using gradient descent to approximate the projection operation. Here we verify this method in the phase retrieval task (ProjDiff-GD in Table 13). The results indicate that approximating the projection operator of $\mathbf{x}_0$ with respect to the observation equation by minimizing $||\mathbf{y} - \mathcal{A}(\mathbf{x})||^2$ starting from $\mathbf{x}_0$ can also achieve satisfactory performance, which suggests that ProjDiff has generalizability for general nonlinear inverse problems.

**Sensitivity tests for the hyperparameters.** We conduct the sensitivity tests for the step size $\eta_1$ and noise level $\sigma_0$ in ProjDiff on noisy super-resolution task on CelebA, as shown in Table 14. '×$a$' denotes that we perturb the input standard deviation of ProjDiff by multiplying $a$. The results indicate that ProjDiff exhibits a certain degree of robustness to the step size and noise level. This also demonstrates that retaining the equivalent noise level as an adjustable hyperparameter is feasible when it cannot be directly calculated.

**RED-diff with random initialization.** The RED-diff algorithm is observed to accidentally fail in the 50% random inpainting task. Here we demonstrate that by adopting the random initialization method proposed in this work, RED-diff achieves more reasonable results. We compare RED-diff's performance on ImageNet for both noise-free and noisy inpainting tasks when using random initialization (denoted as 'RED-diff with random init') and using degraded images for initialization as proposed in [24]. The results are shown in Table 15. RED-diff's performance is significantly improved when using random initialization.

## D  Additional results for source separation and partial generation

### D.1  Results on MSDM model

Two models were trained in [45]: one is MSDM which models the joint distribution among the four instruments, and the other is ISDM which models the distribution of each instrument independently. ISDM performs better than MSDN in the source separation task as reported in [45]. We have presented ProjDiff's performance on ISDM in the main text, and here we supplement the experiment on MSDM. The SI-SDR$_i$ metrics of ProjDiff, RED-diff, MSDM-Gaussian, and MSDM-Dirac are shown in Table 16. ProjDiff also demonstrates the best performance on the MSDM model.

Table 14: Ablation study of the hyperparameters in ProjDiff on noisy super-resolution task on CelebA.

| $\eta_1 \backslash \sigma_0$ | PSNR ↑ ×0.7 | ×0.9 | ×1.0 | ×1.1 | ×1.3 | LPIPS ↓ ×0.7 | ×0.9 | ×1.0 | ×1.1 | ×1.3 |
|---|---|---|---|---|---|---|---|---|---|---|
| 0.1 | 29.92 | 30.16 | 30.19 | 30.18 | 30.11 | 21.96 | 21.71 | 22.25 | 22.70 | 23.29 |
| 0.2 | 29.69 | 29.94 | 29.96 | 29.94 | 29.85 | 22.48 | 20.92 | 21.23 | 21.60 | 22.12 |
| 0.4 | 29.16 | 29.47 | 29.49 | 29.47 | 29.37 | 24.25 | 21.04 | 20.86 | 21.06 | 21.51 |
| 0.8 | 28.45 | 28.85 | 28.89 | 28.87 | 28.75 | 27.67 | 22.82 | 21.85 | 21.63 | 21.74 |
| 1.6 | 27.69 | 28.10 | 28.14 | 28.09 | 27.88 | 32.23 | 26.62 | 24.88 | 24.07 | 23.69 |

Table 15: Comparison of RED-diff with and without random initialization on ImageNet inpainting task. LPIPS metrics are multiplied by 100.

| | Inpainting ($\sigma = 0$) PSNR ↑ SSIM ↑ LPIPS ↓ FID ↓ | Inpainting ($\sigma = 0.05$) PSNR ↑ SSIM ↑ LPIPS ↓ FID ↓ |
|---|---|---|
| RED-diff | 9.81 / 0.17 / 83.58 / 268.93 | 9.85 / 0.17 / 83.92 / 281.65 |
| RED-diff with random init | 29.86 / 0.89 / 14.55 / 15.68 | 25.98 / 0.66 / 29.38 / 28.02 |

Table 16: SI-SDR$_i$ metrics on MSDM (higher is better).

| Method | Bass | Drums | Guitar | Piano | Mean |
|---|---|---|---|---|---|
| MSDM-Gaussian | 13.93 | 17.92 | 14.19 | 12.11 | 14.54 |
| MSDM-Dirac | 17.12 | 18.68 | 15.38 | 14.73 | 16.48 |
| RED-diff | 16.00 | 19.98 | 16.15 | 13.67 | 16.45 |
| **ProjDiff** | **17.65** | **20.76** | **17.47** | **15.17** | **17.76** |

## D.2 Ablation study

**Repetition steps.** ProjDiff repeats $N = 5$ times on each time step as we state in Section 3.3 in the source separation task, while MSDM-Gaussian and MSDM-Dirac [45] only repeat twice using the correction steps method in [19]. A natural doubt is whether the performance improvement arises from more function evaluations. We test MSDM-Dirac algorithm with more steps, and the results are shown in Table 17. When increasing the correction steps from 2 to 5 which matches the function evaluations of ProjDiff, the performance of MSDM-Dirac declines. This verifies that ProjDiff's advantage does not simply arise from more function evaluations. Moreover, we test ProjDiff's performance with more repetitions using the MSDM model, namely 10, 25, and 50 times with the step size adjusting accordingly to stabilize convergence. The step size is set to $0.5/N$, where $N$ is the number of repetitions. The results are shown in Table 18. Further increasing the number of repetitions leads to even better metrics for ProjDiff.

Table 17: More correction steps for MSDM-Dirac (higher is better).

| | Bass | Drums | Guitar | Piano | Mean |
|---|---|---|---|---|---|
| MSDM-Dirac (2 correction steps) | 17.12 | 18.68 | 15.38 | 14.73 | 16.48 |
| MSDM-Dirac (5 correction steps) | 15.68 | 17.49 | 15.05 | 14.01 | 15.56 |

**Momentum mechanism.** We use Polyak momentum in the source separation task. Here we conduct experiments to validate the effectiveness of the momentum mechanism. The results with different momentum hyperparameter $\beta$ ranging from 0 to 0.9 are shown in Table 19. Setting $\beta$ to 0.5 yields the best performance. We infer that a certain level of momentum can accelerate convergence, while over large momentum may cause oscillations at small time steps and thus affect the results. This explains why the performance is relatively poor when $\beta = 0.9$ in the source separation task.

**Weak observation problem and Restricted Encoding.** Here we further explain why the partial generation task is a weak observation problem. The partial generation task aims to generate the tracks of other instruments based on some instruments, e.g., generating drums, piano, and guitar based on bass. The constraint provided by the observations is very weak, which is because in a musical ensemble, the correlation between different instruments exists but is not significant, and a composer can freely create the melodies of other instruments based on certain instruments' melodies. Therefore, this is more of a generation task than a restoration task, and we refer to this type of problem as the "weak observation problem" where the constraints provided by the observations are weak.

Now we verify the effectiveness of the proposed Restricted Encoding method in the partial generation task. While keeping other parameters unchanged, we switch the sampling method of $\mathbf{x}_t$ in ProjDiff from Restricted Encoding to sampling directly from the forward process $q(\mathbf{x}_t|\mathbf{x}_0)$ (denoted as 'w/o Restricted Encoding'). The results are shown in Table 20. Without Restricted Encoding, the

Table 18: More repetition steps for ProjDiff (higher is better).

|  | Bass | Drums | Guitar | Piano | Mean |
|---|---|---|---|---|---|
| ProjDiff 5 steps | 17.65 | 20.76 | 17.47 | 15.17 | 17.76 |
| ProjDiff 10 steps | 17.67 | 20.78 | 17.47 | 15.22 | 17.79 |
| ProjDiff 25 steps | 17.71 | 20.80 | 17.55 | 15.30 | 17.84 |
| ProjDiff 50 steps | **17.74** | **20.83** | **17.57** | **15.32** | **17.86** |

Table 19: Different momentum for ProjDiff (higher is better).

|  | Bass | Drums | Guitar | Piano | Mean |
|---|---|---|---|---|---|
| ProjDiff $\beta = 0$ | 17.56 | 20.75 | 17.41 | 15.06 | 17.70 |
| ProjDiff $\beta = 0.3$ | 17.47 | 20.75 | 17.44 | 15.10 | 17.69 |
| ProjDiff $\beta = 0.5$ | 17.65 | **20.76** | **17.47** | **15.17** | **17.76** |
| ProjDiff $\beta = 0.7$ | 17.64 | 20.75 | 17.33 | 15.03 | 17.69 |
| ProjDiff $\beta = 0.9$ | **17.82** | 20.74 | 17.16 | 15.00 | 17.67 |

performance of ProjDiff significantly declines. This indicates the importance of Restricted Encoding for the weak observation problem.

**Visualization of the waveforms.** In the partial generation tasks, an observation is that as the number of generated instruments decreases, the advantage of ProjDiff becomes smaller. We point out this is because the sub-FAD metric considers the combination of the generated and given tracks. Therefore, in cases where more instruments are provided, the combined track is closer to the ground truth, and the difference among different algorithms would be smaller. This can also explain why ProjDiff performs slightly worse than MSDM and RED-diff when generating only the tracks of guitar. This may be because ProjDiff tends to aggressively create new musical segments, thus resulting in a lower similarity to the ground truth set. We present some waveform samples in Figure 5 to verify this claim. The samples are randomly selected and the same row shares the same ID. ProjDiff's results have larger amplitudes while RED-diff and MSDM's results tend to remain silent.

Moreover, following the conventions in acoustics researches, we include some audio samples of ProjDiff in the supplementary materials for subjective evaluation.

# E    Experimental details

## E.1    Inverse problem settings

**Source separation.** Denote the tracks of piano, drums, guitar, and bass as $\mathbf{x}_1, \mathbf{x}_2, \mathbf{x}_3, \mathbf{x}_4 \in \mathbb{R}^n$, where n represents the length of the sequences. The observation function for source separation is $\mathbf{y} = \left((1,1,1,1)(\mathbf{x}_1, \mathbf{x}_2, \mathbf{x}_3, \mathbf{x}_4)^T\right)^T = \mathbf{x}_1 + \mathbf{x}_2 + \mathbf{x}_3 + \mathbf{x}_4$.

**Partial generation.** The observation matrix for partial generation is a diagonal matrix with elements of either zero or one, i.e., $\mathbf{A} = \mathrm{diag}(a_{11}, a_{22}, a_{33}, a_{44})$ with $a_{11}, a_{22}, a_{33}, a_{44} \in \{0, 1\}$. 0 represents that the corresponding instruments are to be generated while 1 represents that the instrument is provided as the condition. The observation is $\mathbf{y} = \left(\mathbf{A}(\mathbf{x}_1, \mathbf{x}_2, \mathbf{x}_3, \mathbf{x}_4)^T\right)^T$.

**Super-Resolution.** The super-resolution task uses $4 \times 4$ average pooling as the degradation function, which is quite similar to the source separation task. Consider a $4 \times 4$ image block, with elements

Table 20: Comparison of ProjDiff with and without Restricted Encoding (lower is better).

|  | B | D | G | P | BD | BG | BP | DG | DP | GP | BDG | BDP | BGP | DGP |
|---|---|---|---|---|---|---|---|---|---|---|---|---|---|---|
| ProjDiff with Restricted Encoding | 0.42 | 1.15 | 0.31 | 0.60 | 1.37 | 0.69 | 1.06 | 1.41 | 1.60 | 1.17 | 1.66 | 1.79 | 1.85 | 2.25 |
| ProjDiff w/o Restricted Encoding | 0.44 | 2.26 | 0.18 | 0.70 | 3.67 | 1.14 | 2.55 | 3.10 | 3.24 | 1.74 | 5.66 | 7.26 | 6.13 | 4.85 |

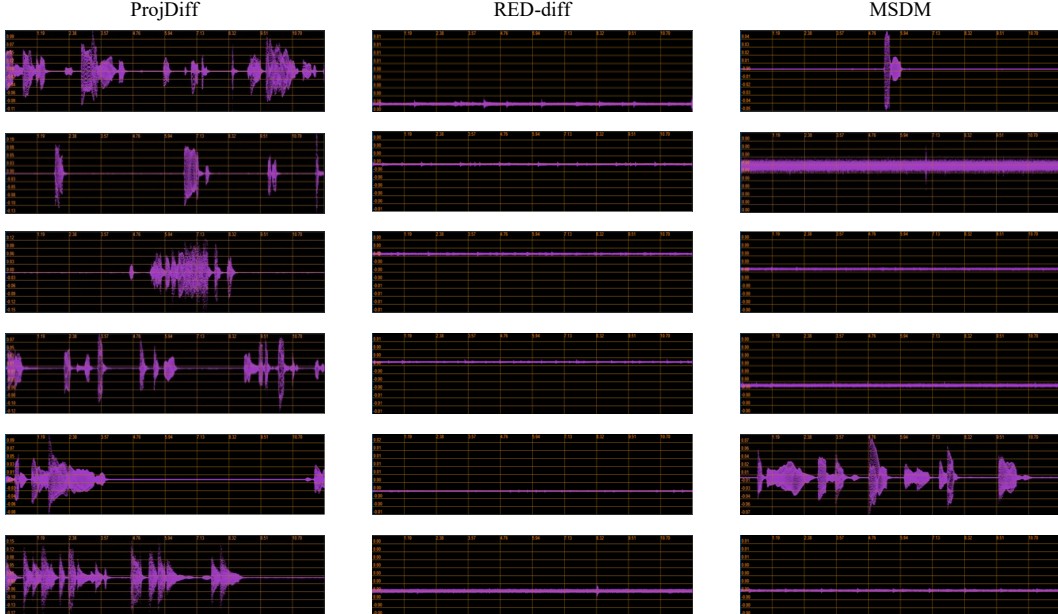

Figure 5: Samples of the partial generation results. The results from ProjDiff (left) have larger amplitude, while results from RED-diff (middle) and MSDM (right) have more silent periods.

$\mathbf{x}_{i,j} \in \mathbb{R}^3, i, j = 1, 2, 3, 4$. The dimension of 3 represents the RGB channels. The observation equation is thus given by $\mathbf{y} = \frac{1}{16} \sum_{i,j=1}^4 \mathbf{x}_{i,j}$. For the entire image, the calculation can be paralleled for all $4 \times 4$ blocks using matrix operations in PyTorch.

**Random inpainting.** The observations matrix for random inpainting is a diagonal matrix of zeros and ones, $\mathbf{A} = \mathrm{diag}\{a_{kk}\}$ for $1 \le k \le \mathbf{n}$ where $\mathbf{n}$ is the total number of pixels in an image, specifically $256 \times 256$ in this work. The diagonal elements $a_{kk}$ have a 50% chance to be 0 and a 50% chance to be 1.

**Gaussian deblurring.** We follow the settings from [20, 21] for Gaussian deblurring, with a 1D Gaussian kernel of size 5 and a standard deviation of 10. The observation equation is in the form of convolution. Efficient SVD proposed in [20] can be used for calculating the projection operator.

**Phase retrieval.** Our experimental setup for phase retrieval follows [23]. The observation is the amplitude spectrum of the original image. However, as the phase retrieval task is highly ill-posed, it is nearly impossible to recover the image directly from the amplitude spectrum. Therefore, we follow the standard practices [54, 55, 23] by first padding the image with zeros and then computing the amplitude spectrum. This increases the spectral resolution and provides more information for retrieval. Thus the observation equation for phase retrieval is $\mathbf{y} = |\mathrm{DFT}(\mathbf{Px})|$, where P represents the padding operation, which is a linear operator. The original image is $256 \times 256$, and the padding is done with size 64 in four directions, resulting in a padded image of $384 \times 384$. Note that one significant difference between the phase retrieval experiment in this paper and in [23] lies in that [23] reported the best results from 4 independent runs for each image, while we perform all the algorithms once for each image.

**High dynamic range.** The observation for the high dynamic range task can be written element-wise as

$$f(\mathbf{x}_i) = \begin{cases} -1 & 2\mathbf{x}_i \le -1; \\ 2\mathbf{x}_i & -1 < 2\mathbf{x}_i < 1; \\ 1 & 2\mathbf{x}_i \ge 1, \end{cases} \tag{67}$$

i.e., first multiplying all the pixels by 2 and then clipping them within $[-1, 1]$.

## E.2 Hyperparameters for the compared algorithms

**DDRM.** DDRM uses the recommended parameters from [20], i.e. $\eta = 0.85$ and $\eta_b = 1.0$.

**DDNM.** DDNM uses the recommended parameter $\eta = 0.85$ from [21]. For noisy tasks, we report the performance of the DDNM+ algorithm [21].

**DPS.** Following the settings in [23], the step size is set to $\xi_t = \xi / \|\mathbf{y} - \mathcal{A}(\boldsymbol{\mu_\theta}(\mathbf{x}_t))\|_2$. On ImageNet, $\xi$ is set to 1 for super-resolution and inpainting tasks, and $0.4$ for Gaussian deblurring. Since [23] did not conduct experiments on CelebA, we conducted task-by-task parameter tuning for the learning rate in DPS and reported the results, ensuring a fair comparison. For phase retrieval and high dynamic range tasks, $\xi = 0.4$.

**RED-diff.** First, in the image restoration tasks, we follow the settings in [24] using the Adam optimizer with momentum pairs of $(0.9, 0.99)$. RED-diff requires a balancing parameter $\lambda$ and a step size lr. $\lambda$ is set to 0.25 for all image restoration tasks, and the step size lr is set to 0.25 for the super-resolution and 0.5 for all other tasks. However, in the source separation and partial generation tasks, we find that the Adam optimizer yields poor results. Therefore, we run RED-diff with the same parameters used in ProjDiff and set $\lambda = 1.0$ which is tuned to the best. Thus in the source separation task, the momentum is set to 0.5 and the step size is set to 0.1 with 5 repetitions. In the partial generation task, the momentum is set to 0.9 and the step size is set to 0.05 with no repetitions.

**ΠGDM.** We carefully conducted hyperparameter searches for each task using the first eight images with the average PSNR as the metric. Except for Noisy Gaussian Deblur task, we avoided using other algorithms to initialize ΠGDM to ensure a fair comparison. Specifically, we first use the forward transition to map the degraded image to the noise level corresponding to the 500th step, and then use 100 steps of ΠGDM to solve the inverse problem. We find that this approach is much better than starting directly from white noise (i.e. 1000th step). For Noisy Gaussian Deblur task, ΠGDM without initialization from other algorithms struggled to achieve satisfactory results. Therefore, we use DDNM+ to obtain the sample at the 500th step and then switch to ΠGDM to complete the solution.

**DMPS.** The step size hyperparameter $\lambda$ was searched for each task using the first eight images with the average PSNR as the metric.

**Resample.** The weight hyperparameter $\gamma$ was searched for each task using the first eight images with the average PSNR as the metric. ReSample was originally designed for Latent Diffusion, thus in our experiments, we consider the encoder-decoder as identity mappings.

**DiffPIR.** The weight hyperparameter $\lambda$ was searched for each task using the first eight images with the average PSNR as the metric.

**MSDM/ISDM-Gaussian/Dirac for source separation.** Following [45], the algorithms use the $S_{churn}$ mechanism [56], with $S_{churn} = 20$ for the MSDM model and 40 for the ISDM model. Correction steps [19] are set to 2.

**MSDM for partial generation.** Similarly, $S_{churn} = 10$ and no correction steps are used.

## E.3 Details for metrics calculation

Here we provide the details of how we calculate the metrics for replication and fair comparison.

For image restoration, we first save the generated results in the png format. When all results are generated, we read them back in to calculate the metrics. For PSNR and SSIM, we use peak_signal_noise_ratio and structural_similarity methods from the skimage library[4]. When calculating SSIM, the parameter 'data_range' is set to 'data_range=generated_image.max()-generated_image.min()'. For LPIPS, we use the LPIPS library[5]. Note that before calculating LPIPS, pixel values are normalized to $[-1, 1]$. And for FID, we use the torch-fidelity library[6].

---

[4]https://github.com/scikit-image/scikit-image
[5]https://github.com/richzhang/PerceptualSimilarity
[6]https://github.com/toshas/torch-fidelity

For the SI-SDR$_i$ metric used in source separation, the calculation is as described in [45]:

$$\text{SI-SDR}_i = \text{SI-SDR}(\mathbf{x}_n, \hat{\mathbf{x}}_n) - \text{SI-SDR}(\mathbf{x}_n, \mathbf{y}),$$

$$\text{SI-SDR}(\mathbf{x}_n, \hat{\mathbf{x}}_n) = 10 \log_{10} \frac{||\alpha \mathbf{x}_n||^2 + \epsilon}{||\alpha \mathbf{x}_n - \hat{\mathbf{x}}_n||^2 + \epsilon}, \tag{68}$$

where $\mathbf{x}_n$ is the ground truth sequence, $\mathbf{y}$ represents the ground truth summation of the four instruments, and $\hat{\mathbf{x}}_n$ is the estimated track from the algorithm. $\alpha = \frac{\mathbf{x}_n^T \hat{\mathbf{x}}_n + \epsilon}{||\mathbf{x}_n||^2 + \epsilon}$ and $\epsilon = 10^{-8}$ to prevent numerical errors.

For the sub-FAD metric in the partial generation tasks, we first calculate the summation of the generated tracks and the corresponding conditional tracks and save the mixtures in the wav format. Then we use the frechet_audio_distance library[7] to compute the FAD metrics between the generated mixtures and the ground truth mixtures.

# F   Implemention details of ProjDiff

## F.1   Initialization for noisy observation with non-integer $t_a$

In the case of noisy observations, we aim to initialize $\mathbf{x}_0$ using the optimized $\mathbf{x}_{t_a}$ as $\boldsymbol{\mu}_{\boldsymbol{\theta}}(\mathbf{x}_{t_a}, t_a)$. However, note that $t_a$ may not correspond exactly to any discretized time step in practice. In such instance, we assume $t_0 \in [1, 2, \ldots, T]$ such that $t_0 \leq t_a \leq t_0 + 1$, and consequently we can first perform a forward transition as $\hat{\mathbf{x}}_{t_0+1} = \sqrt{\overline{\alpha}_{t_0+1}/\overline{\alpha}_{t_a}} \mathbf{x}_{t_a} + \sqrt{1 - \overline{\alpha}_{t_0+1}/\overline{\alpha}_{t_a}} \boldsymbol{\epsilon}$ for some $\boldsymbol{\epsilon} \sim \mathcal{N}(\mathbf{0}, \mathbf{I})$, and then initialize $\mathbf{x}_0 \leftarrow \boldsymbol{\mu}_{\boldsymbol{\theta}}(\hat{\mathbf{x}}_{t_0+1}, t_0 + 1)$.

## F.2   Details for linear observations

The projection operator for source separation is

$$\mathcal{P}(\mathbf{x}_1, \mathbf{x}_2, \mathbf{x}_3, \mathbf{x}_4) = (\mathbf{x}_1, \mathbf{x}_2, \mathbf{x}_3, \mathbf{x}_4) + \frac{1}{4} \left( \mathbf{y} - (\mathbf{x}_1, \mathbf{x}_2, \mathbf{x}_3, \mathbf{x}_4)(1, 1, 1, 1)^T \right) (1, 1, 1, 1). \tag{69}$$

The projection operator for partial generation is

$$\mathcal{P}(\mathbf{x}_1, \mathbf{x}_2, \mathbf{x}_3, \mathbf{x}_4) = (\mathbf{x}_1, \mathbf{x}_2, \mathbf{x}_3, \mathbf{x}_4)(\mathbf{I} - \mathbf{A}) + \mathbf{y}\mathbf{A}. \tag{70}$$

Super-resolution and random inpainting are similar to source separation and partial generation, respectively. The projection operators of these problems can be obtained without SVD. Regarding equivalent variance, for $4 \times 4$ super-resolution, based on the average of independent Gaussian variables, the equivalent variance is $16\sigma^2$ with $\sigma^2$ being the variance of the noise adding to the observation. For inpainting, the equivalent noise should enjoy the same variance as the noise on the observations.

Gaussian deblurring task is a bit more complex. The projection operator is

$$\mathcal{P}(\mathbf{x}_0) = (\mathbf{I} - \mathbf{A}^\dagger \mathbf{A})\mathbf{x}_0 + \mathbf{A}^\dagger \mathbf{y}, \tag{71}$$

where $\mathbf{A}^\dagger$ is the Moore-Penrose pseudo-inverse of $\mathbf{A}$. The element-wise equivalent noise can be handled using the SVD.

## F.3   Details for nonlinear observations

**Phase retrieval.** For the phase retrieval task, consider the observation $\mathbf{y} = |\text{DFT}(\mathbf{P}\mathbf{x})|$. Given input $\mathbf{x}_0$ and observations $\mathbf{y}$, first we calculate $\mathbf{z} = \text{DFT}(\mathbf{P}\mathbf{x}_0)$, and compute the projection operator of $\mathbf{y} = |\mathbf{z}|$ with respect to $\mathbf{z}$, which is $\tilde{\mathbf{z}} = \mathbf{y} \odot \mathbf{z}/|\mathbf{z}|$ where the division is element-wise and $\odot$ denotes the Hadamard product. Next, we map $\tilde{\mathbf{z}}$ back to the original pixel space. As DFT is invertible and the $\mathbf{P}$ is the padding matrix, we define the inverse transformation matrix corresponding to $\mathbf{P}$ as $\mathbf{P}_1$ that extract the central $256 \times 256$ pixels from a $384 \times 384$ image. Thus the inverse mapping can be written as $\tilde{\mathbf{x}} = \mathbf{P}_1 \text{DFT}^{-1}(\tilde{\mathbf{z}})$. Finally, as the resulting $\tilde{\mathbf{x}}$ is in the complex space, we project it onto

---

[7]https://github.com/gudgud96/frechet-audio-distance

Table 21: Optimal step sizes for ProjDiff.

| Dataset | ImageNet | | | CelebA | | | FFHQ | |
|---|---|---|---|---|---|---|---|---|
| Noise-free | Super-Resolution 20/100steps | Inpainting 20/100steps | Gaussian Deblurring 20/100steps | Super-Resolution 100steps | Inpainting 100steps | Gaussian Deblurring 100steps | Phase Retrieval 1000steps | HDR 100steps |
| $\eta$ | 1.5 / 1.7 | 1.1 | 0.9 | 0.7 | 1.1 | 0.8 | 1.5 | 2.0 |
| Noisy | Super-Resolution 20/100steps | Inpainting 20/100steps | Gaussian Deblurring 20/100steps | Super-Resolution 100steps | Inpainting 100steps | Gaussian Deblurring 100steps | Phase Retrieval 1000steps | HDR 100steps |
| $\eta_1$ | 0.9 / 0.5 | 4.0 / 1.0 | 0.5 / 0.1 | 0.4 | 1.1 | 0.1 | 1.9 | 1.0 |

the real space, namely $\mathbf{x}_{\mathrm{proj}} = \mathrm{real}(\tilde{\mathbf{x}})$, where the $\mathrm{real}(\cdot)$ operator takes the real part of a complex tensor.

Regarding equivalent noise, we use the principle of energy equality in the space domain and in the frequency domain. The energy of observation noise is $\mathrm{n}_1^2\sigma^2$ where $\mathrm{n}_1 = 384$ is the size of the spectrum image. Assuming the variance of the effective noise is $\sigma_{\mathrm{eff}}$, the noise energy in the space domain is $\mathrm{n}^2\sigma_{\mathrm{eff}}^2$ with $\mathrm{n} = 256$. Equating both energy yields $\sigma_{\mathrm{eff}} = \mathrm{n}_1\sigma/\mathrm{n} = 1.5\sigma$.

**High dynamic range.** The regular projection operator of the HDR task can be expressed element-wise as follows

$$\mathcal{P}(\mathbf{x}_i, \mathbf{y}_i) = \begin{cases} \mathbf{y}_i/2 & -1 < \mathbf{y}_i < 1; \\ 0.5 & \mathbf{y}_i = 1 \text{ and } \mathbf{x}_i \leq 0.5; \\ -0.5 & \mathbf{y}_i = -1 \text{ and } \mathbf{x}_i \geq -0.5; \\ \mathbf{x}_i & \text{else.} \end{cases} \tag{72}$$

This formulation applies to the noise-free case. Nonetheless, we discover an adjusted version of the projection operator that can further accommodate noisy observations, which is

$$\mathcal{P}_{\mathrm{noisy}}(\mathbf{x}_i, \mathbf{y}_i) = \begin{cases} \mathbf{y}_i/2 & -0.5 < \mathbf{y}_i < 0.5; \\ \mathbf{y}_i/2 & \mathbf{y}_i \geq 1 \text{ and } \mathbf{x}_i \leq 0.5; \\ \mathbf{y}_i/2 & \mathbf{y}_i \leq -1 \text{ and } \mathbf{x}_i \geq -0.5; \\ \mathbf{x}_i & \text{else.} \end{cases} \tag{73}$$

This adjusted form is more adept at handling uncertainty in the observation values within the intervals $[-1, -0.5]$ and $[0.5, 1]$ induced by the observation noise. We employ this modified projection operator in the noisy HDR task. The equivalent noise variance can be simply approximated as $\sigma_{\mathrm{eff}}^2 = (\sigma/2)^2$.

### F.4 Hyperparameters.

In all experiments, ProjDiff uses SGDM as the optimizer. The momentum is set to $0.5$ for the source separation task and $0.9$ for the partial generation task. The step size is set to $0.1$ for source separation with $5$ repetition steps and $0.05$ for partial generation with no repetition. For all image restoration tasks, the momentum is set to $0$, and $\eta_2$ for the noisy observations is fixed to $1.0$. The DDIM variance $\tilde{\sigma}_t$ is set to $0$ for $t \leq t_a$ and to $\tilde{\sigma}_t = \sigma_t = \sqrt{\frac{1-\overline{\alpha}_{t-1}}{1-\overline{\alpha}_t}}\sqrt{1 - \frac{\overline{\alpha}_t}{\overline{\alpha}_{t-1}}}$ for $t > t_a$. No repetitions are applied. Other tuned optimal step sizes are shown in Table 21. The time step schedule follows [45] for the source separation and partial generation tasks, and follows [20, 21] for the image restoration tasks.

## G  Discussion about the MAP framework

ProjDiff shares similarities with the Maximum A Posteriori (MAP) estimation method. Theoretically, in noisy situations, it is only necessary to use a Gaussian likelihood, i.e., $p(\mathbf{y}|\mathbf{x}_0) = \mathcal{N}(\mathbf{y}; \mathbf{x}_0, \sigma^2\mathbf{I})$, to solve the inverse problems. However, ProjDiff introduces a new auxiliary variable, which leads to better experimental outcomes than the MAP framework. Here, we present some qualitative discussion towards this experimental result.

On one hand, if we are given perfectly accurate priors and likelihoods, and have sufficient computational capability, we could obtain the exact posterior and also its gradients. Within the MAP framework, this should lead to the ideal results. However, in practice, the priors provided by diffusion

models are not entirely accurate, and the weight coefficients between the priors and the likelihood cannot always be perfectly set. Moreover, dealing with the priors in diffusion models relies on stochastic optimization methods. These factors imply that the MAP framework often fails to achieve the desired solution. In such cases, further exploring the information or capabilities within the diffusion model can provide substantial assistance in solving inverse problems and enhance performance. This is one of the reasons why ProjDiff may have a performance advantage over the original MAP framework.

On the other hand, MAP methods utilize gradient descent to leverage the information from the observation, while ProjDiff transforms this into a projection operation by introducing noisy auxiliary variables. This approach offers numerous advantages: there is no need to consider the step size of gradient descent (at least for the likelihood term); there is no need to consider the weight coefficients between the likelihood and prior terms; and it ensures consistency between noisy samples and observations (the role of this consistency has also been confirmed in the noise-free scenario). Moreover, the introduction of this auxiliary variable indicates that ProjDiff can actually be viewed as simultaneously recovering clean data and the noise added to the observations, which can yield more accurate results than merely characterizing the noise prior with a Gaussian distribution. These transformations are all thanks to ProjDiff's utilization of both the clean prior and noisy prior modeled by diffusion models (i.e., the prior modeled by the diffusion model and its denoising capability).

## H Algorithm blocks

Here, we present additional algorithm blocks for ProjDiff used in the experiments. Algorithm 2 is used for noise-free restoration tasks. Algorithm 3 and Algorithm 4 are used in the source separation task and partial generation task, respectively.

---

**Algorithm 2** ProjDiff for VP diffusion (noise-free).

---

**Require:** Observation $\mathbf{y}$, observation function $\mathbf{A}$, pre-trained diffusion model $\boldsymbol{\mu_\theta}$, step size $\eta$, total steps $T$, noise schedule $\overline{\alpha}_1 \ldots \overline{\alpha}_T$.
1: Sample $\boldsymbol{\epsilon}_T \sim \mathcal{N}(\mathbf{0}, \mathbf{I})$;
2: Initialize $\mathbf{x}_0 \leftarrow \boldsymbol{\mu_\theta}(\boldsymbol{\epsilon}_T, T)$;
3: **for** $t = T$ to 1 **do**
4:      Sample $\boldsymbol{\epsilon}_t \sim \mathcal{N}(\mathbf{0}, \mathbf{I})$;
5:      Calculate the approximate stochastic gradient: $\tilde{\mathbf{d}}_{\mathbf{x}_0,t} = \mathbf{x}_0 - \boldsymbol{\mu_\theta}(\sqrt{\overline{\alpha}_t}\mathbf{x}_0 + \sqrt{1 - \overline{\alpha}_t}\boldsymbol{\epsilon}_t, t)$;
6:      Update $\mathbf{x}_0$: $\mathbf{x}_0 \leftarrow \mathcal{P}_{\mathbf{A},\mathbf{y}}(\mathbf{x}_0 - \eta\tilde{\mathbf{d}}_{\mathbf{x}_0,t})$;
7: **end for**
8: **return** $\mathbf{x}_0$

---

---

**Algorithm 3** ProjDiff for VE diffusion (noise-free).

---

**Require:** Observation $\mathbf{y}$, observation function $\mathbf{A}$, pre-trained diffusion model $\boldsymbol{\mu_\theta}$, step size $\eta$, momentum $\beta$, total steps $T$, iterations per step $N$, noise schedule $\overline{\sigma}_1 \ldots \overline{\sigma}_T$.
1: Sample $\boldsymbol{\epsilon}_T \sim \mathcal{N}(\mathbf{0}, \mathbf{I})$;
2: Initialize $\mathbf{x}_0 \leftarrow \boldsymbol{\mu_\theta}(\overline{\sigma}_T\boldsymbol{\epsilon}_T, T)$;
3: $\mathbf{v} \leftarrow \mathbf{0}$;
4: **for** $t = T$ to 1 **do**
5:      **for** $k = 0$ to $N - 1$ **do**
6:          Sample $\boldsymbol{\epsilon}_{t,k} \sim \mathbf{N}(\mathbf{0}, \mathbf{I})$;
7:          Calculate the approximate stochastic gradient: $\tilde{\mathbf{d}}_{\mathbf{x}_0,t} = \mathbf{x}_0 - \boldsymbol{\mu_\theta}(\mathbf{x}_0 + \overline{\sigma}_t\boldsymbol{\epsilon}_{t,k}, t)$;
8:          Update momentum: $\mathbf{v} \leftarrow \beta\mathbf{v} + (1 - \beta)\tilde{\mathbf{d}}_{\mathbf{x}_0,t}$;
9:          Update $\mathbf{x}_0$: $\mathbf{x}_0 \leftarrow \mathcal{P}_{\mathbf{A},\mathbf{y}}(\mathbf{x}_0 - \eta\mathbf{v})$;
10:      **end for**
11: **end for**
12: **return** $\mathbf{x}_0$

---

---

**Algorithm 4** ProjDiff for VE diffusion with restricted encoding (noise-free).

---

**Require:** Observation $\mathbf{y}$, observation function $\mathbf{A}$, pre-trained diffusion model $\boldsymbol{\mu_\theta}$, step size $\eta$, momentum $\beta$, total steps $T$, iterations per step $N$, noise schedule $\overline{\sigma}_1 \ldots \overline{\sigma}_T$.
1: Sample $\boldsymbol{\epsilon}_T, \boldsymbol{\epsilon}_0 \sim \mathcal{N}(\mathbf{0}, \mathbf{I})$;
2: Initialize $\mathbf{x}_0 \leftarrow \boldsymbol{\mu_\theta}(\overline{\sigma}_T\boldsymbol{\epsilon}_T, T)$;
3: $\mathbf{v} \leftarrow \mathbf{0}$;
4: **for** $t = T$ to 1 **do**
5:      **for** $k = 0$ to $N - 1$ **do**
6:          Sample $\boldsymbol{\epsilon}_{t,k} \sim \mathcal{N}(\mathbf{0}, \mathbf{I})$;
7:          Calculate the approximate stochastic gradient: $\tilde{\mathbf{d}}_{\mathbf{x}_0,t} = \mathbf{x}_0 - \boldsymbol{\mu_\theta}(\mathbf{x}_0 + \overline{\sigma}_{t-1}\boldsymbol{\epsilon}_0 + \sqrt{\overline{\sigma}_t^2 - \overline{\sigma}_{t-1}^2}\boldsymbol{\epsilon}_{t,k}, t)$;
8:          Update momentum: $\mathbf{v} \leftarrow \beta\mathbf{v} + (1 - \beta)\tilde{\mathbf{d}}_{\mathbf{x}_0,t}$;
9:          Update $\mathbf{x}_0$: $\mathbf{x}_0 \leftarrow \mathcal{P}_{\mathbf{A},\mathbf{y}}(\mathbf{x}_0 - \eta\mathbf{v})$;
10:      **end for**
11: **end for**
12: **return** $\mathbf{x}_0$

---

# I   Visualization results of image restoration

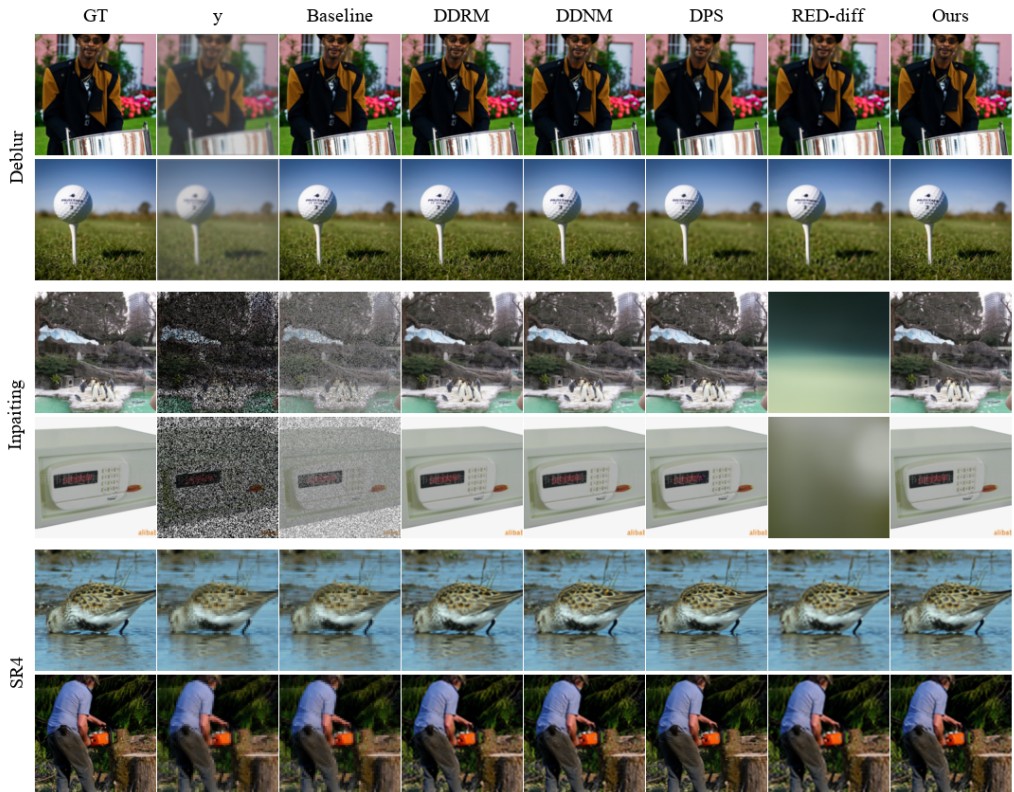

Figure 6: Noise-free results on ImageNet. Baseline means $\hat{\mathbf{x}}_0 = \mathbf{A}^\dagger \mathbf{y}$.

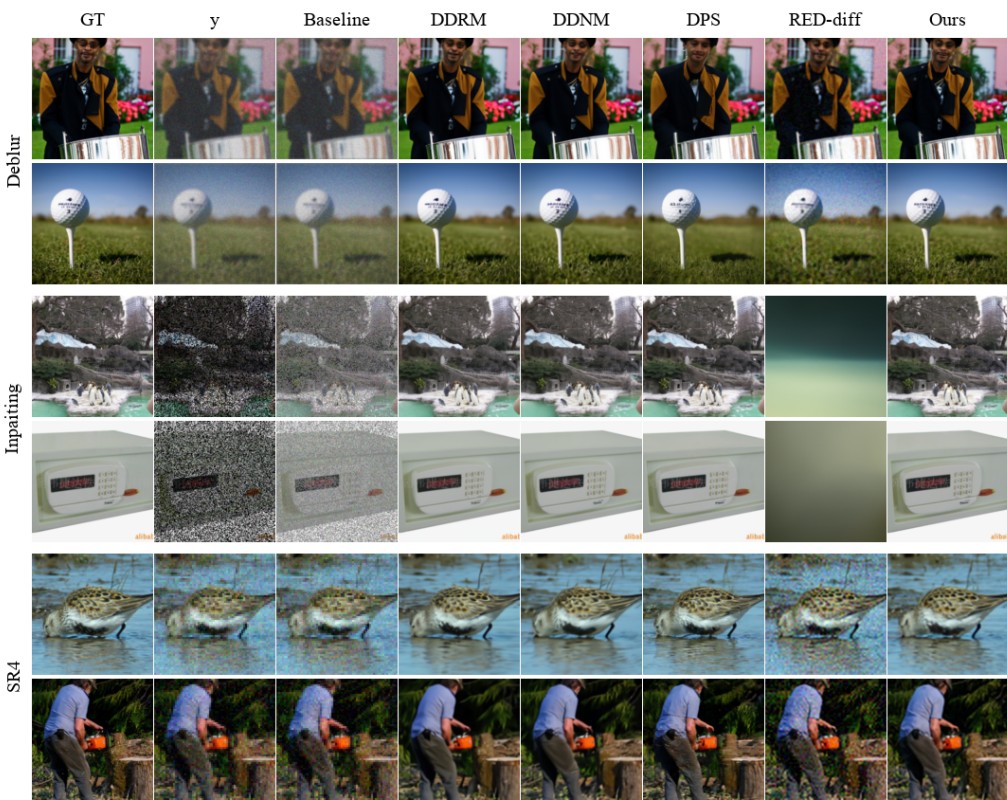

Figure 7: Noisy results on ImageNet $\sigma = 0.05$. Baseline means $\hat{\mathbf{x}}_0 = \mathbf{A}^\dagger \mathbf{y}$.

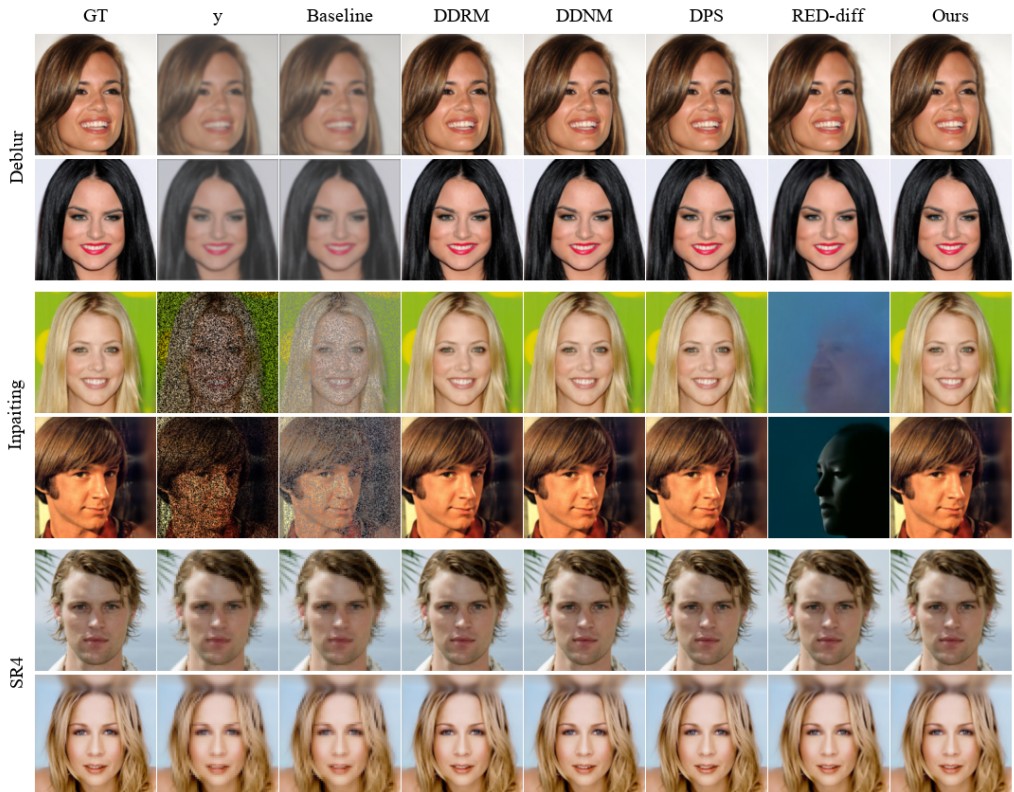

Figure 8: Noise-free results on CelebA. Baseline means $\hat{\mathbf{x}}_0 = \mathbf{A}^\dagger \mathbf{y}$.

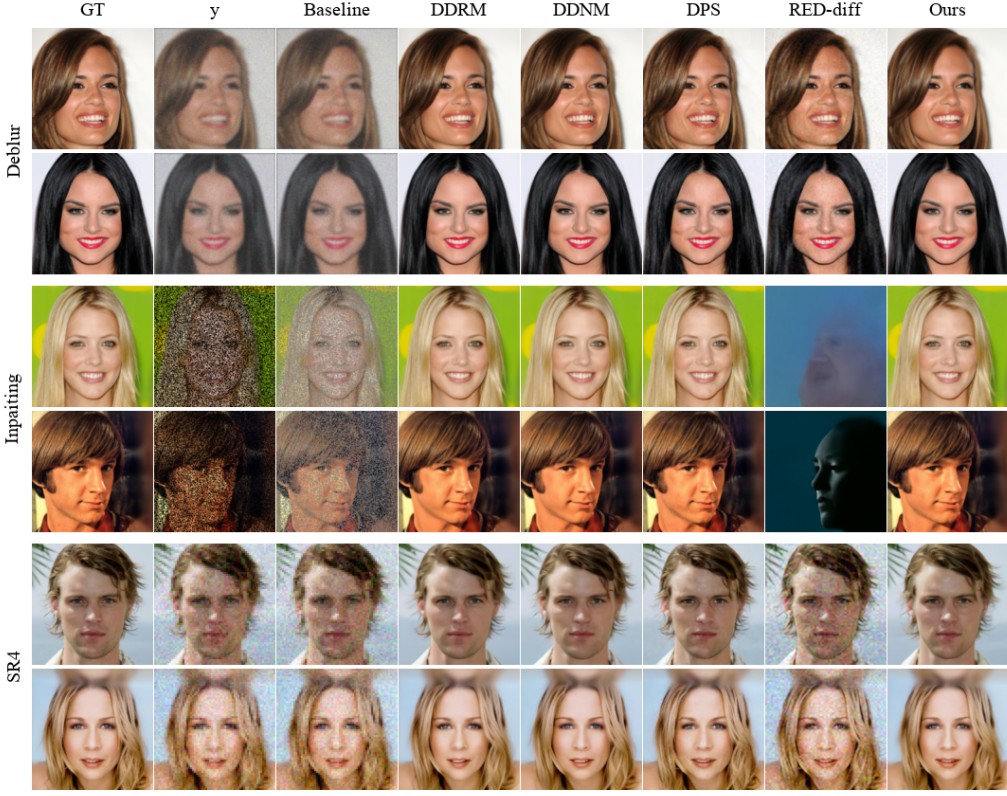

Figure 9: Noisy results on CelebA $\sigma = 0.05$. Baseline means $\hat{\mathbf{x}}_0 = \mathbf{A}^\dagger \mathbf{y}$.

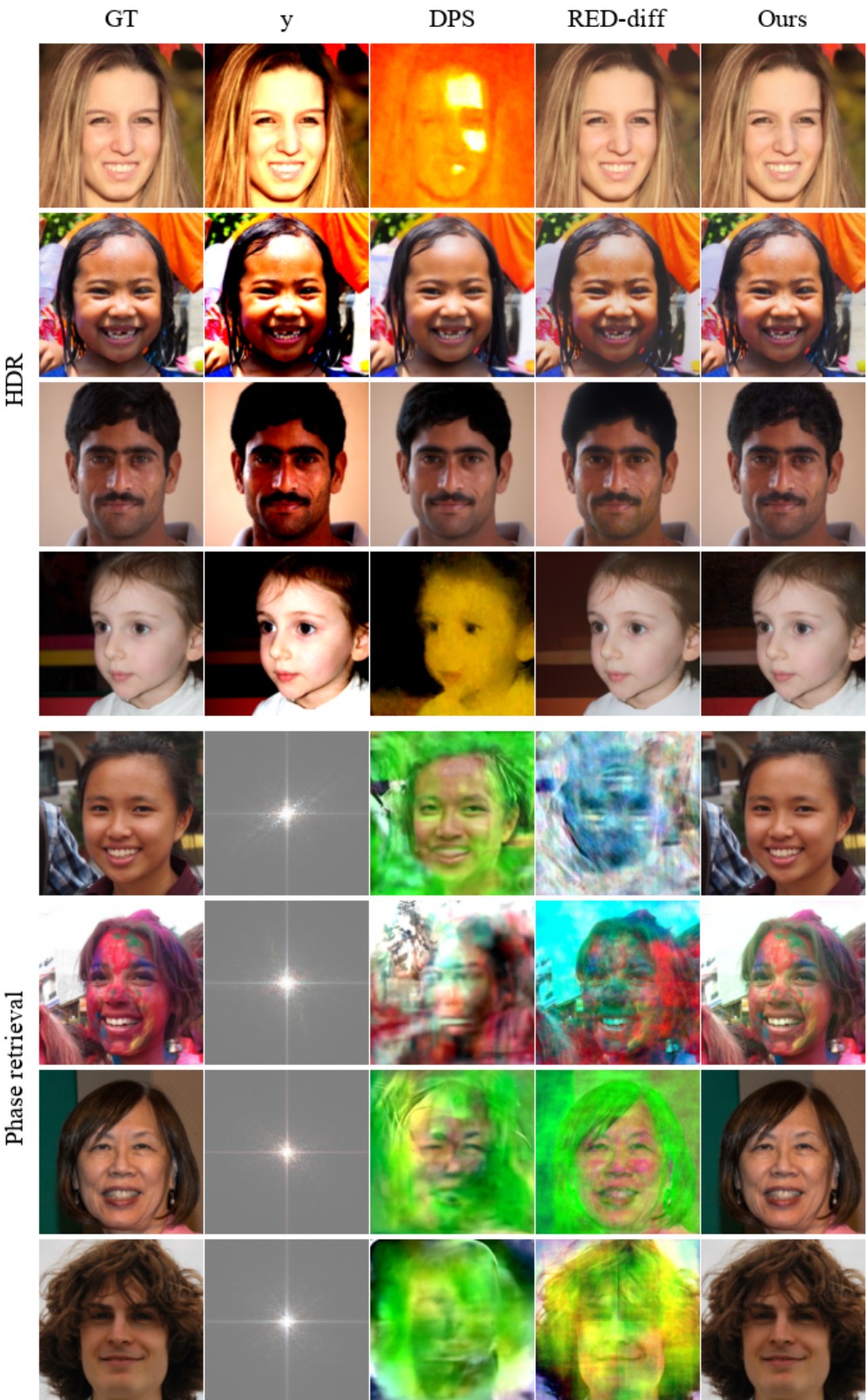

Figure 10: Noise-free nonlinear restoration on FFHQ ($\sigma = 0$).

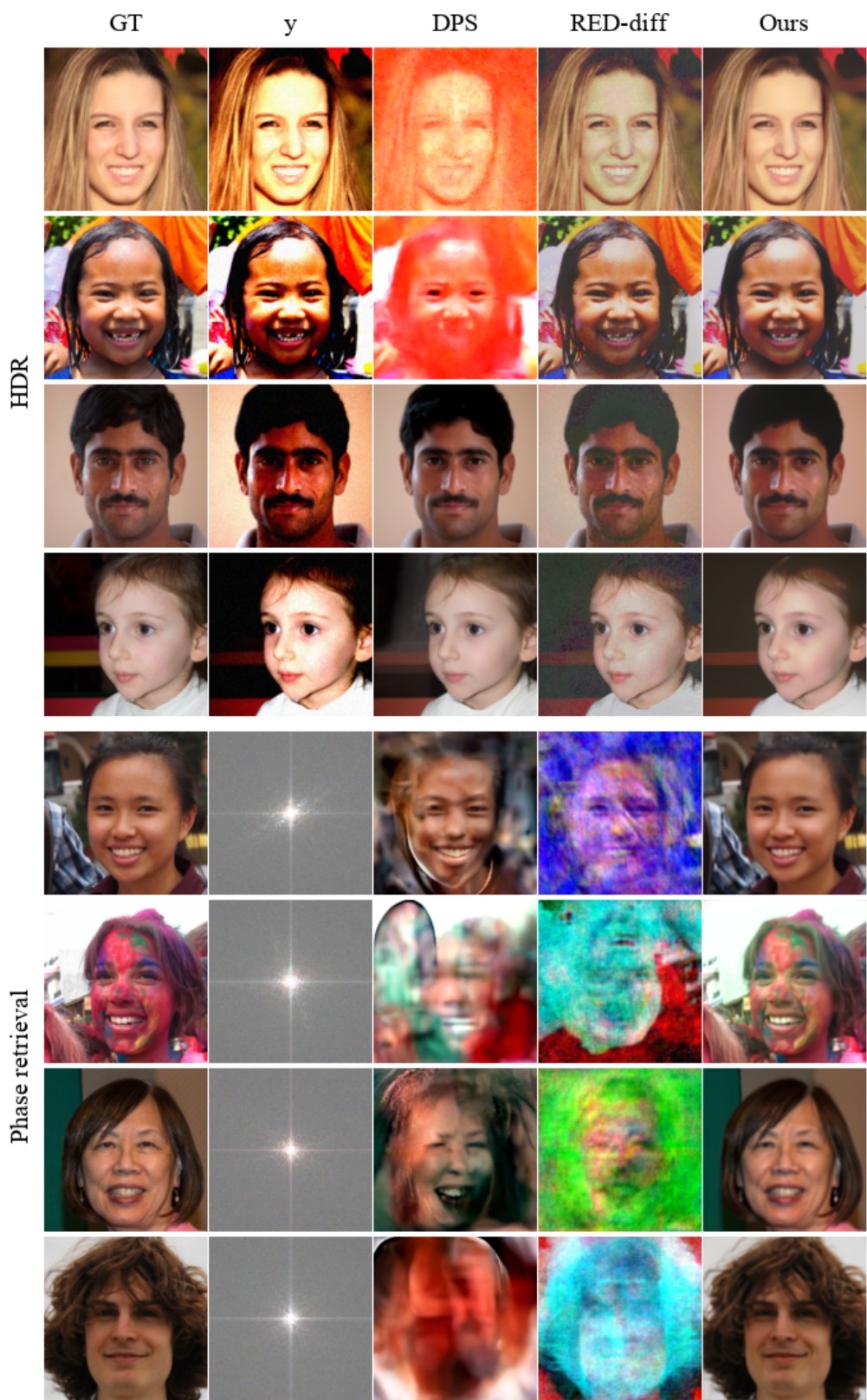

Figure 11: Noisy nonlinear restoration on FFHQ ($\sigma = 0.05$).

