# OpenReview forum: "Unleashing the Denoising Capability of Diffusion Prior for Solving Inverse Problems"
_NeurIPS.cc/2024/Conference — NeurIPS 2024 poster_

### Official Review · Reviewer_krnF · 2024-07-12

**Soundness:** 3
**Presentation:** 2
**Contribution:** 3
**Rating:** 6
**Confidence:** 4

**Summary:**

Authors propose ProjDiff which reframes noisy inverse problems with diffusion models as a two-variable constrained optimization by introducing an auxiliary optimization variable. Authors derive a two-variable ELBO as a proxy for the log-prior and solve the optimization problem via projection gradient descent. Authors conduct comprehensive experiments on several image restoration tasks (super-resolution, inpainting and deblurring), source separation, and partial generations tasks.

**Strengths:**

* Overall, paper is easy to follow and structured nicely.
* I appreciate extensive numerical results that are presented in the paper.
* Numbers for source separation look significantly better than the baselines (I have to note that I'm not an expert in that area)

**Weaknesses:**

* Prior work section is not comprehensive.
* The gains in performance compared to baseline methods (especially for noisy restoration tasks) is not convincing. Moreover, hyperparameter search for baseline models were not conducted and wrong baseline method was used (DDNM for measurements with noise).
* See the questions below.

**Questions:**

* I have several questions regarding DDNM numbers. First of all, DDNM [1] is designed for noise-free image reconstruction problems. I believe that is the reason why DDNM on Gaussian deblurring performs bad (as in Table 1: 7db PSNR, 0.03 SSIM, etc.). I would suggest the authors to switch to the DDNM+ variant as described in [1] for noisy inverse problems. If authors are already using DDNM+, could you clarify why the performance is bad on Gaussian deblurring?
* line 37-39: "However, it’s worth noting that, since diffusion models are inherently effective at denoising, considering the observation noise in the likelihood term fails to fully leverage diffusion models’ denoising capability." Could the authors clarify what it means to not fully leverage diffusion models here?
* I would recommend the authors to include $\Pi$GDM [2] in their comparisons especially since it performs much better than DDRM.
* Some other missing citations on solving inverse problems with diffusion models: CCDF [3], latent diffusion models: PSLD [4]
* line 663-665: "Since [23] did not conduct experiments on CelebA, we use the parameters on FFHQ for the CelebA dataset as both FFHQ and CelebA are datasets of faces.". I don't think this is a good practice. In my experience, even though both CelebA and FFHQ are face datasets, DPS is not robust to the choice of step size. I believe some hyper-parameter search on a small set of images is necessary for fair comparison.
* Do you think ProjDiff can be extended for using latent diffusion models as a prior?
---
References:

[1] Wang, Yinhuai, Jiwen Yu, and Jian Zhang. "Zero-shot image restoration using denoising diffusion null-space model." arXiv preprint arXiv:2212.00490 (2022).

[2] Song, Jiaming et al. “Pseudoinverse-Guided Diffusion Models for Inverse Problems.” International Conference on Learning Representations (2023).

[3] Chung, Hyungjin, Byeongsu Sim, and Jong Chul Ye. "Come-closer-diffuse-faster: Accelerating conditional diffusion models for inverse problems through stochastic contraction." Proceedings of the IEEE/CVF Conference on Computer Vision and Pattern Recognition. 2022.

[4] Rout, Litu, et al. "Solving linear inverse problems provably via posterior sampling with latent diffusion models." Advances in Neural Information Processing Systems 36 (2024).

**Limitations:**

Limitations are adequately addressed by the authors.

---

> ### Author Rebuttal · Authors · 2024-08-06
>
> We are grateful for the reviewer's affirmation of the effectiveness of our method, especially the outstanding performance in music separation and partial generation tasks. Below are our responses.
>
> 1. DDNM+ numbers for noisy Deblurring tasks
>
> We are very grateful to the reviewer for pointing out this issue. We used the official code of DDNM in the noisy linear inverse problem. We double-checked the official code and confirmed that it indeed calls the DDNM+ algorithm. However, we found that the implementation of Gaussian Deblurring in the official code seems to be incomplete, which is the reason for the abnormal results in our paper.
>
> We re-implement the part of Gaussian Deblurring and test it on both ImageNet and CelebA. The corrected results are shown in Table 5 of the attached PDF, where DDNM+ achieved reasonable results. We also checked Super-resolution and Inpainting and confirmed that their implementations are correct. We will correct the experimental results in our paper and annotate that we used the DDNM+ algorithm for noisy tasks. We would like to express our gratitude once again to the reviewer for helping us correct this error.
>
> 2. ''What it means to not fully leverage diffusion models''
>
> Our observation is that diffusion models not only model the distribution of clean data $p(x_0)$, but they also implicitly model the distribution of noisy data with different variances of Gaussian noise $p(x_t)$ within the diffusion process. This means that compared to other types of data priors, diffusion models can provide more usable information for solving inverse problems with Gaussian noisy observations. Therefore, if solved within a conventional MAP framework, only the clean data prior modeled by the diffusion model would be utilized while the noisy data prior within the diffusion model is neglected.
>
> In contrast, we introduce an auxiliary variable to transform the noisy observation into an equivalent noise-free observation of the noisy sample, thereby utilizing both the clean data prior and the noise data prior. The process of recovering clean data from the noisy auxiliary variable can be regarded as a denoising process, hence our method actually utilizes both the data prior modeled by the diffusion model and its denoising capability.
>
> 3. More comparative algorithms and related works
>
> Thanks for the reviewer's suggestion. We supplement a comparison of ProjDiff with $\\Pi$GDM, together with ReSample, DiffPIR, and DMPS on the CelebA dataset. The results are shown in Table 1 & 2 of the attached PDF. ProjDiff still demonstrates competitive performance.
>
> For $\\Pi$GDM, we carefully conducted hyperparameter searches for each task using the first eight images with the average PSNR as the metric. Except for Noisy Gaussian Deblur task, we avoided using other algorithms to initialize $\\Pi$GDM to ensure a fair comparison. Specifically, we first use the forward transition to map the degraded image to the noise level corresponding to the 500th step, and then use 100 steps of $\\Pi$GDM to solve the inverse problem. We find that this approach is much better than starting directly from white noise (i.e. 1000th step). For Noisy Gaussian Deblur task, $\\Pi$GDM without initialization from other algorithms struggled to achieve satisfactory results. Therefore, we use DDNM+ to obtain the sample at the 500th step and then switch to $\\Pi$GDM to complete the solution.
>
> We notice that $\\Pi$GDM does not always outperform the performance of our reproduced DDRM. This may be because we report the performance of DDRM using 100 steps, which is much better than the default parameters of DDRM (20 steps).
>
> We will further refine the explanation of related work according to the reviewer's suggestions.
>
> 4. DPS on CelebA
>
> Thanks for the reviewer's suggestion. We have re-adjusted the learning rate hyperparameter $\\eta$ for DPS on the CelebA dataset task by task. Using the average PSNR of the first eight images as the criterion (consistent with the method we used to adjust the hyperparameter of ProjDiff), we carefully scanned the learning rate hyperparameter with a precision of 0.1. The latest results are shown in Table 1&2 of the attached PDF (denoted as DPS-s). We will update these results in the paper and include an explanation.
>
> 5. ProjDiff for latent diffusion
>
> Yes, we believe ProjDiff can be applied in latent diffusion. The main challenge in solving inverse problems with latent diffusion priors is the high nonlinearity of the observation equation introduced by the neural encoder-decoder. ProjDiff has the capability to handle nonlinear observations. Firstly, for noise-free observations, ProjDiff can approximate the projection operation by taking $x_0$ as the initial point and minimizing $||y-\\mathcal{A}(\\mathcal{D}(x))||^2$ (where $\\mathcal{D}$ represents the neural Decoder). Secondly, for noisy observations, ProjDiff also needs to set an equivalent noise level, which can be left as an adjustable hyperparameter. Our supplemental experiments in Table 7 of the attached PDF have verified that ProjDiff is robust to some perturbations in the noise variance (i.e. equivalent noise level), making manual adjustment of the equivalent noise level feasible. We believe this is a promising direction for future work.
>
> We would like to express our gratitude once again for the reviewer's suggestions for our work. Should there be any further questions, we welcome continued discussion.

---

> ### Comment · Reviewer_krnF · 2024-08-09
>
> I would like to thank the time and effort authors put into their rebuttal. I appreciate that they have incorporated all the feedback into providing a fairer comparison (fixing DDNM+, tuning DPS step size, etc.) against baseline methods especially in a short window of time.
>
> Most of my concerns are alleviated and my questions are answered. I've also read the comments of other reviewers (and authors' response to them). I believe the results are more convincing with the updated numbers and the proposed changes. Therefore, I'm happily updating my score to $6$ and increasing soundness to $3$.

---

> > ### Author Response · Authors · 2024-08-12
> >
> > We would like to express our heartfelt gratitude for your recognition and raising the score! Thank you once again for your kind consideration!

---

### Official Review · Reviewer_dXRm · 2024-07-13

**Soundness:** 3
**Presentation:** 2
**Contribution:** 3
**Rating:** 6
**Confidence:** 3

**Summary:**

This paper proposed a new sampling strategy for solving noisy inverse problems using diffusion models. The proposed method is called ProjDiff, which is based on two-step minimization of log-posterior using gradient descent. The sampling procedure is derived by (1) Reparametrization of the noisy measurement as a noiseless measurement of an intermediate noisy sample in the diffusion process, and (2) obtaining a proxy objective from the variational lower bound.

**Strengths:**

+ The paper's approach to addressing the noisy inverse problem using the reparamterization of measurement is interesting.

+ Deriving the sampling procedure by directly minimizing the evidence lower bound is innovative.

+ The performance is competitive with the SOTA methods in solving inverse problems.

**Weaknesses:**

+ The paper is not well-written and it is hard to follow. Justification:

     1. Theoretical analysis results lack organization. Example: Lemma 1 and Lemma 2 should be separated from the main proof. The proofs for the propositions should also be separated and distinctly expressed.

     2. Theoretical results should be explicitly included. The proofs for Lemma 1 and Lemma 2 are not included in the text. If taken from a source, then the reference should be mentioned, otherwise, the proofs should be included.

    3. Abstract includes some concerning statements.
        >Since inverse problems inherently entail maximum a posteriori estimation, previous works have endeavored to integrate diffusion priors into the optimization frameworks.

        Inverse problems could be solved using MMSE estimation. It is unclear what kind of optimization algorithm the authors refer to.

        > by introducing an auxiliary optimization variable. By employing gradient truncation, the projection gradient descent method is efficiently utilized to solve the corresponding optimization problem.

        The abstract does not reflect the method used.  The statement is vague and could be inferred in many different ways.

       Revising the abstract to better reflect the proposed method can strengthen the paper.

  4. The second and third paragraph of the introduction needs to be revised. Instead of briefly mentioning what each existing method does, the authors could state how their method is different from the literature and how it contributes to the literature.



* The truncation of the gradient is based on an assumption and leads to an approximation in the solutions. Thus, the reduced computational cost should not be mentioned as a contribution of the paper, specifically when the authors have not included any experimental results regarding this.
 > 44: Through gradient truncation, we obtain an approximation of the stochastic gradient of the objective, effectively sidestepping significant computational overhead.


Overall, the weaknesses of the paper are mostly related to the presentation of the paper and the blurred message, not the proposed method and effectiveness of it.

Minor issues:

> Line 19: Their remarkable ability to capture data priors enables effective guidance for solving inverse problems [6], which are widely exploited in image restoration.

Guidance in inverse problems usually refers to enforcing data consistency with the measurement and relies on the forward model.

**Questions:**

What does the author mean by weak observations? (Line 214)

How does the algorithm look when used for nonlinear inverse problems? Can equation (8) be written for nonlinear measurements?


How do the authors set the $\eta_1$, $\eta_2$, and equivalent noise level in the algorithms?
Are the solutions sensitive to these parameters?

**Limitations:**

Partially addressed by the authors.

---

> ### Author Rebuttal · Authors · 2024-08-06
>
> We are grateful for the reviewer's recognition of the innovation and effectiveness of our work. Below are our responses.
>
> 1. Reorganizing the theoretical analysis and the proofs of the lemmas.
>
> Thanks for the reviewer's suggestion. We will move Lemma 1 and 2 ahead of the proofs of the Propositions, and we will present the proofs of the two Propositions in a clear and independent manner. The proofs of Lemma 1 and 2 are based on Bayes' theorem and mathematical induction, and we will supplement the proofs of this part.
>
> 2. Abstract includes some concerning statements.
>
> Thanks for the reviewer's feedback. We will revise the abstract accordingly.
>
> The first sentence will be revised as:
>
> ''Previous works have endeavored to integrate diffusion priors into the maximum a posteriori estimation (MAP) framework and design optimization methods to solve the inverse problem.''
>
> The second sentence will be revised as:
>
> ''... by introducing an auxiliary optimization variable that represents a 'noisy' sample at an equivalent denoising step. The projection gradient descent method is efficiently utilized to solve the corresponding optimization problem by truncating the gradient through the $\\mu$-predictor.''
>
> We will further refine the abstract to make it more fluent and reflective of our approach.
>
> 3. The authors could state how their method is different from the literature and how it contributes to the literature.
>
> Compared to previous works that apply diffusion in the MAP framework for solving inverse problems, our core observation is that diffusion models not only model the distribution of clean data but also implicitly model the distribution of noisy data. This allows us to leverage both the clean data prior and the noisy data prior to solve noisy inverse problems (in other words, both the prior modeled by the diffusion model and its denoising capabilities). Thanks for the reviewer's suggestions, and we will further improve the expression of paragraph 3 and paragraph 4.
>
> 4. The reduced computational cost
>
> Thanks for the reviewer's comments. We did not intend to present the reduction of complexity as a contribution of this paper. Line 44 will be revised to:
>
> ''We obtain a more practical approximation of the stochastic gradient of the objective through gradient truncation.''
>
> Additionally, we have added an experiment comparing the use of our gradient approximation with not using it, as shown in Table 6 of the attached PDF. It can be observed that using the approximation results in a certain loss of performance, but the efficiency is nearly tripled. Considering that the method using this approximation has already achieved satisfactory performance, we accept a certain performance loss in exchange for efficiency.
>
> 5. Line 19.
>
> Thanks for the reviewer's suggestions. Line 19 will be revised to
>
> ''Their remarkable ability to capture data priors provides promising avenues for solving inverse problems [6]...''
>
> 6. The weak observations problems.
>
> We refer to inverse problems where the constraints provided by the observation are relatively loose as ''the weak observation problem''. For instance, the partial generation task in this paper aims to generate tracks for other instruments based on certain tracks (e.g., generating tracks of drums, bass, and guitar given a track of piano). This problem is highly flexible as a composer can create various different tracks for other instruments from the same piano track while ensuring harmony. This is a case belonging to the ''the weak observation problems''. In contrast, tasks such as super-resolution, inpainting, and deblurring in image restoration are cases where the observation strongly constrains the original data. For example, given a low-resolution image, its corresponding high-resolution image is almost unique, with lower degrees of freedom, and thus it does not belong to the ''weak observation problems''.
>
> 7. Regarding nonlinear measurements
>
> The algorithm for nonlinear measurements is almost identical, except that the projection operations in equations (19) and (22) are replaced with the projection operations corresponding to nonlinear measurements, or by minimizing $||y-\\mathcal{A}(x))||^2$ from the initial point to approximate the projection operation.
>
> For general nonlinear measurements, equation (8) cannot be strictly written. However, we can still apply a similar idea to handle nonlinear measurements, which is to simply find an equivalent noise level to deal with noisy observations. This equivalent noise level can either be calculated using rules or retained as a hyperparameter to adjust. Our Phase Retrieval experiments and HDR experiments have verified the effectiveness of our method.
>
> 8. The hyperparameters
>
> All of the details of our hyperparameters are provided in Appendix E and F. Regarding the sensitivity of hyperparameters, we supplement a new experiment in Table 7 of the attached PDF. It can be observed that our algorithm exhibits robustness to the step size and the noise variance.
>
> We would like to express our gratitude once again for the reviewer's suggestions for our work. Should there be any further questions, we welcome continued discussion.

---

> > ### Comment · Reviewer_dXRm · 2024-08-12
> > **Response to the rebuttal**
> >
> > Thank you for providing the rebuttal.
> > After reading the reviews from all the reviewers and rebuttals, I would update my score to 6.

---

> > > ### Author Response · Authors · 2024-08-13
> > >
> > > We would like to extend our sincerest gratitude for raising the score! We are greatly encouraged!

---

### Official Review · Reviewer_Wp65 · 2024-07-14

**Soundness:** 3
**Presentation:** 3
**Contribution:** 3
**Rating:** 7
**Confidence:** 2

**Summary:**

The authors present a new way of solving inverse problems using a diffusion model based prior.
The key idea is that the forward process, which is to be undone, usually involves Gaussian noise.
The authors utilize this by viewing the noisy observation as a as random variable at an intermediate time step t of the diffusion forward process.
This allows them to formulate the inverse problem as a joint optimisation problem for x_0 and x_t given the noisy observation.
As a by product, their method is able to solve inverse problems which do not involve noise as well.
The authors evaluate their method and achieve superior results compared to various baselines.

**Strengths:**

* I believe this is a principled new perspective on diffusion models for inverse problems and I am sure it can be potentially useful for many practical applications.

* The method achieves remarkable results.

* I very much appreciate the fact that the method can be applied to noise-free inverse problems, without having to artificially include non-existent Gaussian noise in the model.

**Weaknesses:**

* The motivation, comparing the method to the more established MAP approach, could be a bit more clear.
The authors write:
"In this work, we introduce a novel two-variable ELBO by constructing an auxiliary variable that accounts for the observation noise, thereby utilizing both the prior information and the denoising capability in diffusion models simultaneously."
Why exactly is is describable to utilize the denoising capability of the model? Is the something wrong with the normal way of including the log of the forward model in the optimisation task? Why can we expect an improvement? Is it a problem of optimisation? Surely the way task itself is normally (e.g. [28]) formulated is correctly, or not?

* Some aspects of the method were difficult for me to follow. This may be due to my inexperience with some of the involved mathematics.
In particular, I wonder if the SVD based decoupling could be explained or illustrated differently to facilitate understanding.

**Questions:**

Regarding the SVD composition, I believe I am missing an important aspect about how the noisy observation can be considered as a step in the diffusion process when it involves A.
Let's say A corresponds to a convolution with a blur kernel. Would this not mean that we are considering a blurred noisy image to be part of the diffusion process. This would be incorrect, because the blurring would not be part of the training data statistics which would only include clean images, right?

**Limitations:**

My understanding is that the method is limited to either image degradation processes that involve Gaissian noise, or are noise-free.
Is this the case?
If yes, this would be a significant limitation with respect to practical applicability, e.g. in settings with Poisson shot noise.

---

> ### Author Rebuttal · Authors · 2024-08-06
>
> We are grateful for the reviewer's recognition of the innovation, effectiveness, and practicality of our work. Below are our responses to the reviewer's questions.
>
> 1. Clarification on the motivation and comparison to the established MAP approach.
>
> The core motivation of this paper stems from the fact that diffusion models not only model the distribution of clean data $p(x_0)$, but also simultaneously model the distribution of noisy data with varying variances of additive Gaussian noise $p(x_t)$. When observing a degraded data with Gaussian noise, the noisy data distribution can bring more usable information for solving the inverse problem. Therefore, compared to using only the prior information modeled by the diffusion model for $x_0$, we choose to use the prior information of both $x_0$ and $x_{t_a}$. Our algorithm can be viewed as two steps: first, recover the noisy sample $x_{t_a}$, and then use the consistency constraints of $x_{t_a}$ and the prior of $x_0$ to recover $x_0$ (which can be viewed as a denoising process). In other words, we utilize both the prior modeled by the diffusion model and its denoising capabilities.
>
> The conventional approach of including the logarithm of the forward model in the optimization task is also feasible, as seen in [24, 28] in the main paper, and we have compared with it in our experiments (RED-diff). The improvement of our algorithm compared to these methods can be expected because our method utilizes more information modeled by the diffusion model, i.e., it uses both the prior and denoising capabilities (or both the clean data prior and the noisy data prior). We believe this is the essential difference between our algorithm and the methods that include the log of the forward model and $\\log p(x_0)$ modeled by diffusion models in the MAP framework.
>
> 2. Further explanation on SVD-based decoupling.
>
> Here we provide additional clarification regarding the description in section 2.3. Firstly, since $V$ is orthogonal, we acknowledge that knowing the prior of $x_0$ is equivalent to knowing the prior of $\\overline x_0=V^T x_0$. Secondly, since the covariance matrix of both the forward transition and a single-step backward transition of the diffusion model is the diagonal matrix, each coordinate can be processed individually. Equation (7) presents the observation equation in a coordinate-wise form, where for each coordinate index $i$, the $i$-th coordinate of the noisy observation $\\overline y_i$ can be considered as a noise-free observation of the $i$-th coordinate of the sample $\\overline x_{t_i}$ at step $t_i$. In other words, each coordinate of the noisy observation (after spectral decomposition) can be considered as a noise-free observation of the coordinate at a certain step in the diffusion process.
>
> Regarding the reviewer's concern about a convolution with a blur kernel, it is not about viewing the noisy blurred image as a step in the diffusion process, but rather, after SVD, each coordinate is treated as the noise-free observation of the coordinate corresponding to a certain step in the diffusion model. We hope the above explanation could describe our method more clearly. Thanks for the reviewer's suggestions, and we will make further modifications and explanations regarding the SVD part of the paper.
>
> 3. Regarding the limitations.
>
> Yes, the limitation of this method is that it can only handle noise-free inverse problems or inverse problems with additive Gaussian noise. Addressing other types of noise, such as Poisson noise or multiplicative noise, is a promising direction for further research. In practical applications, noise-free inverse problems or inverse problems with additive Gaussian noise are widespread, hence there is a series of work in the field of solving inverse problems with diffusion models that assumes Gaussian noise (including but not limited to [18, 20, 21, 22, 26, 27] in the main paper). Therefore, our work holds practical significance.
>
> We would like to express our gratitude once again for the reviewer's suggestions for our work. Should there be any further questions, we welcome continued discussion.

---

> > ### Comment · Reviewer_Wp65 · 2024-08-12
> > **Thank you for the rebuttal.**
> >
> > Thank you for the clarifications!
> >
> > **Regarding the motivation:**
> > *"The conventional approach of including the logarithm of the forward model in the optimization task is also feasible, as seen in [24, 28] in the main paper, and we have compared with it in our experiments (RED-diff). The improvement of our algorithm compared to these methods can be expected because our method utilizes more information modeled by the diffusion model, i.e., it uses both the prior and denoising capabilities (or both the clean data prior and the noisy data prior). We believe this is the essential difference between our algorithm and the methods that include the log of the forward model and modeled by diffusion models in the MAP framework."*
> >
> > I am afraid I still can't completely follow here. Surely the correct prior  together with the correct forward model should provide (or be proportional to) the correct posterior. Maximising the posterior will provide the true MAP estimate. How can *"additional information"* beyond this be useful?
> > Are you saying your method is better at finding the MAP estimate (compared to the conventional approach) or is your method finding something better than the MAP estimate?

---

> > > ### Author Response · Authors · 2024-08-12
> > > **Thanks for the reply.**
> > >
> > > We greatly appreciate the reviewers' careful consideration of our rebuttal! The issue can be explained from two perspectives:
> > >
> > > On one hand, ideally, if we are given perfectly accurate priors and likelihoods, and have sufficient computational capability, we could obtain the exact posterior and also its gradients. Within the MAP framework, this should lead to the ideal results. However, in practice, the priors provided by diffusion models are not entirely accurate, and the weight coefficients between the priors and the likelihood cannot always be perfectly set. Moreover, dealing with the priors in diffusion models relies on stochastic optimization methods. These factors imply that the MAP framework often fails to achieve the desired solution. In such cases, further exploring the information or capabilities within the diffusion model can provide substantial assistance in solving inverse problems and enhance performance. This is one of the reasons why our algorithm has a performance advantage over the original MAP framework.
> > >
> > > On the other hand, MAP methods utilize gradient descent to leverage the information from the observation, while ProjDiff transforms this into a projection operation by introducing noisy auxiliary variables. This approach offers numerous advantages: there is no need to consider the step size of gradient descent (at least for the likelihood term); there is no need to consider the weight coefficients between the likelihood and prior terms; and it ensures consistency between noisy samples and observations (the role of this consistency has also been confirmed in the noise-free scenario). Moreover, the introduction of this auxiliary variable indicates that ProjDiff can actually be viewed as simultaneously recovering clean data and the noise added to the observations, which can yield more accurate results than merely characterizing the noise prior with a Gaussian distribution. These transformations are all thanks to ProjDiff's utilization of both the clean prior and noisy prior modeled by diffusion models (i.e., the prior modeled by the diffusion model and its denoising capability). This is why we claim that we "have utilized more information from the diffusion model than the original MAP framework".
> > >
> > > We would like to express our gratitude once again to the reviewer for the response! We hope that these additional answers will make our motivation and the reasons for the performance advantages of our algorithm clearer.

---

> > > > ### Comment · Reviewer_Wp65 · 2024-08-13
> > > > **Thanks**
> > > >
> > > > Thanks, that makes sense.
> > > > Would you include this line of argument in a final version of the paper?

---

> > > > > ### Author Response · Authors · 2024-08-13
> > > > >
> > > > > Absolutely! We are willing to incorporate these arguments into our final version of the paper to make our motivation and the advantages of our algorithm more clear. Thank you for the suggestion!

---

### Official Review · Reviewer_g7gD · 2024-07-18

**Soundness:** 2
**Presentation:** 3
**Contribution:** 2
**Rating:** 3
**Confidence:** 5

**Summary:**

This paper proposes ProjDiff for solving inverse problems with pre-trained diffusion models. By deriving a two-variable ELBO as a proxy for the log-prior, this paper reframes the inverse problems as constrained optimization tasks and address them via the projection gradient method.

**Strengths:**

1. The paper writing is clear and easy to follow.
2. The derivation of some formulas in this paper is solid.

**Weaknesses:**

1. The assumption of this method for the measurement noise in Section 3.1 is doubtable. (1) As the authors claim in the limitation section, assuming the Gaussian noise will limit its applicability to other noise types like Poisson or multiplicative noise; (2) The whole method design heavily relies on know the exact standard deviation of the Gaussian noise \sigma, which is impractical and can lead to robustnesses issues when facing unknown noise level. Actually, the noise estimation itself is a challenging problem, particularly on the degraded measurement A(x). Please check the related works [1,2]. In general, this method is built on an impractical scenario, so I am worried about the practical usage of this method.

2. The approximation for implementing the update rules just after Proposition 2 is not convincing. (1) One of the reasons for this approximation is that "the μ-predictor of the diffusion model should be resilient to small perturbations in the input", but this hope usually is not the truth, particularly when t is large, i.e., at the beginning of the reverse sampling procedure; (2) There is not much evidence for this important approximation, no matter theoretical or empirical evidence.

3. The proposed method is too "delicate", as shown in Algorithm 1. (1) Again, this method needs to know the measurement noise level; (2)
this method contains several hyper-parameters. In Table 17, it is clear that the hyper-parameters are heavily tuned, and there is no specific way to guide the hyper-parameter tuning.

4. No ablation studies for the hyper-parameters. For this "delicate" method, a systematical ablation study is needed. For example, it would be good to report the results for different combinations of some important hyper-parameters.

5. The extension to nonlinear inverse problems is not convincing. This paper proposes to use min_x || y - A(x) ||^2 to approximate the projection operator. However, it is almost impossible to solve nonlinear inverse problems by using this formulation because it will be easy to be stuck in local minimizers. Also, if this formulation can solve nonlinear inverse problems well, then there is no need to write this paper.

6. The experiment settings for phase retrieval are doubtable. (1) I highly suspect that this paper downplays DPS in some implicit ways. I ran DPS for phase retrieval on FFHQ before, and remembered DPS could achieve over 30dB for PSNR after trying different initializations. In the original DPS paper, the authors claim that DPS need different initializations but it seems that this paper omits this; (2) there is no comparison with the golden standard method for phase retrieval, i.e., HIO+ER.

7. Some recent SOTA methods are missing for comparison, including ReSample[3] (ICLR'24 spotlight), DiffPIR[4] (CVPR'23), DMPS[5]. Please check the recent survey for more related works [6].



[1] Liu, X., Tanaka, M. and Okutomi, M., 2013. Single-image noise level estimation for blind denoising. IEEE transactions on image processing, 22(12), pp.5226-5237.

[2] Li, F., Fang, F., Li, Z. and Zeng, T., 2023. Single image noise level estimation by artificial noise. Signal Processing, 213, p.109215.

[3] Song, B., Kwon, S.M., Zhang, Z., Hu, X., Qu, Q. and Shen, L., 2023. Solving inverse problems with latent diffusion models via hard data consistency. arXiv preprint arXiv:2307.08123.

[4] Zhu, Y., Zhang, K., Liang, J., Cao, J., Wen, B., Timofte, R. and Van Gool, L., 2023. Denoising diffusion models for plug-and-play image restoration. In Proceedings of the IEEE/CVF Conference on Computer Vision and Pattern Recognition (pp. 1219-1229).

[5] Meng, X. and Kabashima, Y., 2022. Diffusion model based posterior sampling for noisy linear inverse problems. arXiv preprint arXiv:2211.12343.

[6] Li, X., Ren, Y., Jin, X., Lan, C., Wang, X., Zeng, W., Wang, X. and Chen, Z., 2023. Diffusion Models for Image Restoration and Enhancement--A Comprehensive Survey. arXiv preprint arXiv:2308.09388.

**Questions:**

See weaknesses.

**Limitations:**

See weaknesses.

---

> ### Author Rebuttal · Authors · 2024-08-06
>
> We appreciate the reviewer's suggestions, and here are our responses.
>
> 1. ''The assumption for the measurement noise is doubtable.''
>
> We respectfully disagree. Firstly, in practical applications, noise-free inverse problems or inverse problems with Gaussian noise are ubiquitous, and the standard deviation of the Gaussian noise can be estimated using the method mentioned by the reviewer or retained as an adjustable hyperparameter. Secondly, a series of works  (including but not limited to [18, 20, 21, 22, 26, 27] in the main paper) that use diffusion models to solve the inverse problems assume additive Gaussian noise and known variance (DMPS mentioned by the reviewer also assumes additive Gaussian noise with known variance, and the DiffPIR mentioned by the reviewer also assumes the variance is known). We believe that these works, as well as ours, have practical value. Lastly, as shown in the ablation study in response to point 4, ProjDiff is robust to some perturbation of the standard deviation. Therefore, our work holds practical significance.
>
> 2. ''The approximation after Proposition 2 is not convincing.''
>
> We respectfully disagree. Firstly, this approximation is acceptable when $t$ is small, as the $\\mu$-predictor generally learns well to recover the clean image from a slightly noisy image. Secondly, for large $t$, in the case of VP diffusion, note that the coefficient $\\overline\\alpha_t$ in front of $x_0$ or $x_{t_a}$ is small, thus the gradients corresponding to $x_0$ and $x_{t_a}$ are also small. For VE diffusion, similarly, the dominant term in $\\mu$ will also be the noise term when $t$ is large, making it insensitive to the input $x_0$. Our experimental results have demonstrated the effectiveness of this approximation. Furthermore, we supplement a new experiment that uses the stochastic gradient without this approximation, as shown in Table 6 of the attached PDF (denoted as ProjDiff-FG). It can be observed that using this approximation results in a certain loss of performance, but the efficiency is nearly tripled. Considering that applying this approximation has already achieved satisfactory performance, we are willing to accept a certain performance loss in exchange for efficiency.
>
> 3. ''The proposed method is too 'delicate'.''
>
> We respectfully disagree. Once again, assuming knowledge of the variance of Gaussian noise is a default setting in many works within the field. Secondly, as we explain in Appendix F, we fix $\\eta_2=1.0$, thus only $\\eta_1$ needs to be tuned (as well as the equivalent noise level $\\overline \\alpha_{t_a}$ if we leave it as a hyperparameter to be tuned). We conduct hyperparameter tuning on the first eight images for each task. Given that the complexity of ProjDiff is not too high, tuning hyperparameters is not time-consuming. In the ablation study in response to point 4, we also demonstrate that ProjDiff is robust to some perturbation of the step size and the standard deviation (i.e. the equivalent noise level). To the best of our knowledge, many related works have at least one hyperparameter that needs to be adjusted, including the algorithms mentioned by the reviewer.
>
> 4. Ablation studies for hyper-parameters
>
> We supplement the ablation study concerning $\\eta_1$ and ${\\sigma}_0$ in Table 7 of the attached PDF. ProjDiff exhibits robustness to some perturbation of the hyperparameters of the step size and the noise standard deviation.
>
> 5. ''The extension to nonlinear inverse problems is not convincing''.
>
> The reviewer has misunderstood our approach. One of the core aspects of ProjDiff is to project a sample $x_0$ onto the manifold corresponding to $y=\\mathcal{A}(x)$. When $\\mathcal{A}$ is nonlinear whose projection operator is unavailable, one can instead find a solution that is close to $x_0$ and close to the observation manifold by solving $\\min_x||y-\\mathcal{A}(x)||^2$ starting from $x_0$. This solution can serve as an approximation for the projection operation corresponding to a nonlinear observation equation. We also supplement an experiment to validate the effectiveness of this method in the Phase Retrieval task (ProjDiff-GD in Table 4 of the attached PDF).
>
> 6. ''The experiment settings for phase retrieval are doubtable.''
>
> We respectfully disagree. Our experimental settings for phase retrieval are correct, fair, and reasonable. All of our experimental settings are detailed in Appendix E. We generate one sample per image for each algorithm. The good performance of ProjDiff under this setup further demonstrates its robustness. We also supplement the results of allowing ProjDiff and DPS to generate 4 samples for each image, as shown in Table 3 of the PDF (denoted as ProjDiff-4trials and DPS-4trials). Both DPS and ProjDiff show performance improvements, with ProjDiff in particular exhibiting better performance, reaching a PSNR of 41.58 in the noise-free case. The performance of HIO, ER, and OSS is also presented in Table 3 and the global response, where we report the results with both one repetition and four repetitions.
>
> Note that DPS does not achieve the PSNR of 30 as mentioned by the reviewer. To dispel further doubts about the correctness of our reproduction, we independently run the official code of DPS four times without any modifications (i.e., noise variance set to 0.05, with oversampling consistent with our experiments) and take the best result per sample, obtaining a PSNR of 17.45, indicating that our reproduction is also correct.
>
> 7. Comparison with the latest SOTA.
>
> We supplement comparisons of ProjDiff with ReSample, DiffPIR, DMPS, and $\\Pi$GDM on the CelebA dataset, as shown in Table 1 & 2 in the attached PDF. The hyperparameters are tuned task by task for fairness. ReSample is designed for Latent Diffusion, thus in our experiments, we consider the encoder-decoder as identity mappings. ProjDiff remains highly competitive when compared with these algorithms.

---

> > ### Author Response · Authors · 2024-08-13
> > **Looking forward to your reply!**
> >
> > Dear Reviewer g7gD:
> >
> > We would like to express our sincere gratitude once again for your valuable comments and suggestions.
> >
> > The discussion phase is due to conclude in 30 hours, and we have already received responses from the other three reviewers. We are eagerly awaiting your feedback. Should there be any further questions, we would be more than happy to engage in continued discussion if time permits.
> >
> > Best regards.

---

### Author Rebuttal · Authors · 2024-08-06

Dear Reviewers,

Please refer to the attached one-page PDF for the supplementary experimental results.

We appreciate the valuable feedback provided by all the reviewers on our paper, which has helped us to further refine our work. We are particularly encouraged by the following comments from the reviewers:

(1) ''This is a principled new perspective on diffusion models for inverse problems'' and ''it can be potentially useful for many practical applications'' (Reviewer Wp65)

(2) ''The performance is competitive'' (Reviewer dXRm)

(3) ''extensive numerical results'' and ''Numbers for source separation look significantly better than the baselines'' (Reviewer krnF)

Based on the reviewers' suggestions and advice, we have diligently conducted a substantial number of additional experiments. Please refer to the attached one-page PDF for the supplementary experimental results. Below, we provide a summary and explanation of the supplementary results.

Supplementary experiments include (all LPIPS metrics have been multiplied by 100 for clarity):

(1) New comparisons with $\\Pi$GDM, Resample, DMPS, and DiffPIR on the CelebA dataset (Tables 1 & 2), as suggested by reviewers g7gD and krnF. We have carefully adjusted the parameters task by task for the four algorithms to ensure a fair comparison. The results indicate that ProjDiff is highly competitive compared to these algorithms.

(2) Results of DPS with a task-by-task parameter tuning on the CelebA dataset (DPS-s in Tables 1 & 2). We are grateful for the suggestion from reviewer krnF. We have conducted task-by-task parameter tuning for the learning rate in DPS on the CelebA dataset and reported the results, ensuring a fair comparison. The results show that ProjDiff still holds an advantage over DPS after parameter tuning.

(3) The results for DPS and ProjDiff in the Phase Retrieval task by taking the best one from 4 independent trials for each image (DPS-4trials and ProjDiff-4trials in Table 3), in response to the question raised by reviewer g7gD. With 4 repetitions, ProjDiff shows a very significant improvement in performance, achieves a PSNR of 41.58 in the noise-free scenario, and still holds a considerable advantage over DPS.

(4) Comparisons with ER, HIO, and OSS algorithms on the Phase Retrieval task (Table 3), as suggested by reviewer g7gD. Due to space constraints, only the results with four repetitions are reported in Table 3. We have also tested the results of these three algorithms with one repetition, as shown in the following table.


|            |                Phase Retrieval ($\sigma_0=0$)                |               Phase Retrieval ($\sigma_0=0.1$)               |
| :--------: | :----------------------------------------------------------: | :----------------------------------------------------------: |
|            | PSNR $\uparrow$ SSIM $\uparrow$ LPIPS $\downarrow$ FID $\downarrow$ | PSNR $\uparrow$ SSIM $\uparrow$ LPIPS $\downarrow$ FID $\downarrow$ |
| ER-1trial  |                11.15 / 0.19 / 84.48 / 409.91                 |                11.16 / 0.19 / 84.43 / 412.59                 |
| HIO-1trial |                11.97 / 0.25 / 82.06 / 342.52                 |                11.87 / 0.24 / 82.18 / 339.13                 |
| OSS-1trial |                12.57 / 0.35 / 81.08 / 360.98                 |                12.55 / 0.25 / 81.20 / 364.52                 |

(5) The performance of ProjDiff in the Phase Retrieval task using gradient descent to approximate the projection operation (ProjDiff-GD in Table 4), in response to the question from reviewer g7gD. The results indicate that approximating the projection operator of $x_0$ with respect to the observation equation by minimizing $||y-\\mathcal{A}(x)||^2$ starting from $x_0$ can also achieve satisfactory performance, which suggests that ProjDiff has generalizability for nonlinear inverse problems.

(6) Corrected metrics for the DDNM+ algorithm on the Noisy Gaussian Deblurring task (Table 5). We appreciate the reviewer krnF for pointing out the issue with the metrics. We have re-implemented the DDNM+ algorithm on the Noisy Gaussian Deblurring task and reported the corrected performance.

(7)  Ablation study on the gradient truncation method used in ProjDiff (Table 6), in response to the questions from reviewers g7gD and dXRm. ProjDiff without gradient truncation is denoted as ProjDiff-FG. The results show that there is some performance loss when using gradient truncation compared to not using it, while the efficiency increases by nearly three times. Considering that using gradient truncation has already achieved satisfactory performance, we accept the trade-off of some performance loss for the gain in efficiency.

(8) Sensitivity tests for the hyperparameters $\\eta_1$ and noise level $\\sigma_0$ in ProjDiff (Table 7), as suggested by reviewers g7gD and dXRm. '$\\times a$' denotes that we perturb the input standard deviation of ProjDiff by multiplying $a$ (this will affect the equivalent noise level). The results indicate that ProjDiff exhibits a certain degree of robustness to the step size and noise level. This also demonstrates that retaining the equivalent noise level as an adjustable hyperparameter is feasible when it cannot be directly calculated.

We would like to express our gratitude to all the reviewers once again, and we believe that these additional experiments would significantly contribute to the improvement of our manuscript.

---

### Decision · Program_Chairs · 2024-09-25

**Decision:**

Accept (poster)

**Comment:**

The paper propose a method for solving inverse problems using diffusion models. They reformulate the log-prior using a two-variable method and use projected gradients to solve the resulting constrained optimization problem. The method is applied to super-resolution, inpainting and deblurring tasks. Three reviewers positively rated the paper, with one reviewer providing a very negative response. During the discussion period, two of the positive reviews increased their rating, while the negative reviewer didn't engage in the discussion phase. It is not clear how significant or real the negative reviewers' concerns are, and I'm inclined to give the three combined positive responses more weight. They found the quantitative results to be convincing and the method to be in line with state-of-the-art approaches and so something that could be useful to the NeurIPS audience.